SciPost Physics

Submission

# Efficiency of Dynamical Decoupling for (Almost) Any Spin–Boson Model

A. Hahn[1*], D. Burgarth[1,2], D. Lonigro[2]

**1** School of Mathematical and Physical Sciences, Macquarie University, 2109 NSW, Australia
**2** Department Physik, Friedrich-Alexander-Universität Erlangen-Nürnberg, Staudtstraße 7, 91058 Erlangen, Germany
* alexander.hahn@hdr.mq.edu.au

May 22, 2025

## Abstract

Dynamical decoupling is a technique aimed at suppressing the interaction between a quantum system and its environment by applying frequent unitary operations on the system alone. In the present paper, we analytically study the dynamical decoupling of a two-level system coupled with a structured bosonic environment initially prepared in a thermal state. We find sufficient conditions under which dynamical decoupling works for such systems, and—most importantly—we find bounds for the convergence speed of the procedure. Our analysis is based on a new Trotter theorem for multiple Hamiltonians and involves a rigorous treatment of the evolution of mixed quantum states via unbounded Hamiltonians. A comparison with numerical experiments shows that our bounds reproduce the correct scaling in various relevant system parameters. Furthermore, our analytical treatment allows for quantifying the decoupling efficiency for boson baths with infinitely many modes, in which case a numerical treatment is unavailable.

# 1   Introduction

Noise and decoherence are the main challenges in the current development of quantum technology [1]. Such phenomena are intrinsic to any quantum system, as they arise from the coupling of the system with a surrounding environment (the lab/bath). One of the most commonly used techniques to deal with noise and decoherence in practice is dynamical decoupling [2, 3, 4, 5, 6, 7, 8], an open-loop control strategy that is operated on the hardware level. More precisely, dynamical decoupling consists in averaging out the coupling between the system and the environment through strong and fast rotations on the system alone. This approach can significantly suppress errors in quantum computing [9, 10, 11, 12] and quantum sensing [13, 14], which is crucial for achieving quantum utility.

The two main advantages of dynamical decoupling can be summarized as follows: (i) it suppresses errors before they even occur, and can thus be combined with quantum error correction and mitigation [15]; (ii) it works for *all* finite-dimensional quantum systems and, under rather mild technical assumptions, even for infinite-dimensional ones [16]. The only practical requirement is to pulse faster than the system–bath interaction timescale. In fact, the repetition rate of the pulses determines the efficiency of dynamical decoupling. Consequently, it is crucial to understand how fast the driving has to be in order to achieve the desired error suppression rate—that is, to find *quantitative* bounds for it. Usually, this is done perturbatively in the language of filter functions and with the help of numerical simulations, see e.g. Refs. [17, 18, 19, 20, 21, 22]. However, analytical results are only available for simple toy models [2] or under the strong assumption of finite-energy baths [23, 24], which leaves out a wide range of actual physical systems.

Besides, an analytical treatment offers crucial advantages in the context of dynamical decoupling. On the one hand, analytical efficiency estimates establish a performance *guarantee* which is independent of the validity of perturbative approximations or numerical instabilities; on the other hand, analytical bounds give insights into the scaling of the decoupling fidelity with respect to various system parameters. This is particularly useful if one wants to determine bottlenecks in an experimental setup: for instance, one could wonder whether a faster decoupling or a lower bath temperature may have a stronger effect in reducing errors.

More fundamentally, an analytical treatment of the decoupling error even becomes a ne-

cessity in certain cases. As is well-known in the literature, there are several physical error models for which dynamical decoupling does *not* work, see in particular Refs. [25, 26] and also e.g. Refs. [16, 27, 28]. The reason for the invalidity of dynamical decoupling is exactly the breakdown of the perturbation theory, which is at the ground of the filter function approach to dynamical decoupling. The latter has been developed in Ref. [29] in 1962 and popularized for experimental applications in Ref. [30]. However, as done for instance in Ref. [16], one could equivalently take a Trotterization perspective to dynamical decoupling in which certain assumptions on the input state are made. For this case, Ref. [31] recently showed that these assumptions can be significantly relaxed to the price of a slower error scaling (also see Ref. [32]). This indicates that dynamical decoupling still works for some of these models despite the breakdown of the perturbative filter function approach. Thus, an analytical treatment of the decoupling error via Trotterization could allow for a much wider applicability to physically relevant error models than filter functions.

In this paper, we establish a general framework to analytically study the efficiency of dynamical decoupling for a finite-dimensional quantum system (hereafter, for the ease of exposition, a qubit) coupled to the quintessential example of an infinite-dimensional environment: a bosonic bath. Namely, we shall consider models of field–matter interaction described by operators in the form

$$H = H_{\mathrm{S}} + \sum_k \omega_k a_k^\dagger a_k + \sum_k \left( f_k^* B^\dagger a_k + f_k B a_k^\dagger \right), \tag{1}$$

with $H_{\mathrm{S}}$ being the Hamiltonian of the qubit alone, $(\omega_k)_k$ being the energy modes of the bath, $(f_k)_k$ being coupling constants, $a_k$, $a_k^\dagger$ being the bosonic annihilation and creation operators of mode $k$, respectively, and $B$ being an operator on the system. Finite or countably infinite modes are allowed, and an initial state in the form $\rho_{\mathrm{S}} \otimes \rho_{\mathrm{B}}$, with $\rho_{\mathrm{B}}$ being a thermal (Gibbs) state at any temperature, shall be assumed. We remark that the Hamiltonian in Eq. (1) can also describe systems of qu*d*its as long as either only a single level couples to the bath or all levels couple to the bath simultaneously via a single coupling operator $B$. An example of the first situation would be a Dicke model, while the latter could be certain Tavis–Cummings models. A generalization to arbitrary qu*d*it and multi-atom models is immediate by replacing $B$ with operators that individually couple each system level to the bath. While this would make the notation and calculations more cumbersome, it does not provide new physical insights. Therefore, we will stick to the most important and paradigmatic case of a qubit.

Operators in the form of Eq. (1) belong to the class of generalized spin–boson (GSB) models, whose mathematical properties have attracted interest in recent times [33, 34, 35, 36, 37, 38, 39, 40, 41]. At the physical level, this is a standard class of error models that naturally implements thermal noise, and includes as particular examples some of the most common toy models of quantum optics like the Jaynes–Cummings and Rabi models. See, e.g., Ref. [42, Chapter 3] or Ref. [43]. Such models find applications in a wide range of topics, ranging from quantum optics, quantum information and simulation, solid state and chemical physics. We refer to Ref. [44] for an extensive review on the subject.

Our results can be summarized as follows. For all models in this class—modulo some mild technical assumptions that are needed in the case of infinitely many boson modes—dynamical decoupling works. Furthermore, the decoupling error can be quantitatively bounded in a way that *entirely* depends on the derivatives of the grand canonical partition function $Z(\beta, \mu)$ of the boson bath, and the operators $H_{\mathrm{S}}$ and $B$. All the specific parameters of the boson field only enter our bound via the value of $Z(\beta, \mu)$. This is the content of Theorem 5.3, with a

refined bound being presented in the appendix (see Theorem C.5). In this sense, this paper offers a general recipe to bound the error on dynamical decoupling—once the grand canonical partition function is given, the bound is obtained for free. Additionally, our bounds reveal how the decoupling error can be controlled by tuning the system and experimental parameters.

Furthermore, as we will show, our results crucially rely on a new Trotter theorem for multiple Hamiltonians (Theorem 4.2) which generalizes a result first obtained in Ref. [31]. This result is of interest by itself since dynamical decoupling is usually achieved by Trotterizing between multiple Hamiltonians. Our method is generic and applicable to a large variety of typical error models.

## 1.1 Novelty and significance of this work

This paper addresses two major limitations of the existing frameworks for studying dynamical decoupling. First: the Trotterization approach [16, 26] warrants analytical convergence for unbounded baths, such as bosonic environments, and can therefore be regarded as the most rigorous mathematical framework for dynamical decoupling. However, this method only establishes the *existence* of convergence and does not provide insights into the *rate* of error suppression. In general, the convergence speed of Trotterization can be arbitrarily slow for unbounded Hamiltonians [31, 32], making it impractical for applications. To achieve meaningful results, it is crucial to identify computable conditions under which the optimal error suppression scaling of $1/N$ (with $N$ being the number of decoupling cycles) can be guaranteed.

Second: the filter function approach [17, 18, 19, 20, 22, 29, 30] is a widely used tool to assess the practical performance of dynamical decoupling. It provides a heuristic way to numerically estimate the decoupling fidelity under realistic experimental conditions. However, this approach is inherently perturbative and lacks rigorous convergence guarantees. This raises the risk of overestimating the effectiveness of dynamical decoupling, particularly in regimes where the underlying perturbation theory breaks down. Additionally, while the filter function method is a valuable heuristic method, it neither yields explicit analytical error bounds nor offers a clear connection to the scaling of the decoupling fidelity. On the other hand, filter functions are a useful tool to compare different or design optimal decoupling strategies in the perturbative regime [19].

This work is a step toward bridging the gap between these two approaches. By developing a framework that combines the analytical rigour of the Trotterization approach with the computational simplicity of the filter function formalism, we achieve the best of both worlds. Specifically, we provide explicit, easy-to-compute error bounds for dynamical decoupling in the Trotterization picture. These bounds offer not only rigorous convergence guarantees but also practical insights into the decoupling fidelity. The resulting framework unifies key aspects of the Trotterization and filter function approaches, bringing the theoretical and experimental perspectives closer together. Furthermore, they might help to identify the perturbative regime in which the filter function approach is favourable.

Beyond its immediate practical implications for dynamical decoupling, our results also represent a conceptual step towards understanding the interplay between analytical and numerical methods in quantum control. By introducing error bounds that depend on the grand canonical partition function and system-specific parameters, we demonstrate how to integrate thermodynamic considerations into the mathematical analysis of quantum error suppression. This provides a systematic way to evaluate the efficiency of dynamical decoupling across a broad class of physically relevant models.

## 1.2 Structure of the paper

The paper is structured as follows. In Section 2, as an introductory example, we discuss dynamical decoupling for a particular instance of the models considered in the paper: a spin coupled with a monochromatic boson field through a purely longitudinal interaction (pure dephasing). This model is exactly solvable [2], thus giving us the opportunity to introduce our results while keeping to a minimum the mathematical difficulties encountered in the general case. We then proceed in Section 3 by defining the wider class of models that will be considered in the remainder of the paper. In Section 4 we discuss the technical machinery required for our main result: after introducing the description of the evolution of mixed quantum states for systems with unbounded energy, we provide a new error bound for the Trotter product formula in the presence of multiple Hamiltonians, both for pure states (Theorem 4.1) and for mixed states (Corollary 4.3). In Section 5, we provide the main result of the paper (Theorem 5.3), on the efficiency of dynamical decoupling for all models in the class considered in the paper. We then discuss some other examples in Section 6, and gather some final considerations and outlooks in Section 7. All calculations involved in the proofs of our bounds are contained in the appendix.

## 2 A motivating example

In this section, we shall take a look at a simple motivating example for dynamical decoupling— namely, a spin coupled to a single-mode bosonic bath which induces pure dephasing. A multi-mode generalization of this model has been studied in detail in Ref. [2] and historically led to the advent of filter functions for dynamical decoupling [30, 21, 19]. Since this model is exactly solvable [2], here we will not be concerned about the mathematical subtleties that arise from its unbounded nature. Instead, the model will serve as a motivation and illustrative example. We will study more general models later after introducing the mathematical preliminaries necessary for a rigorous treatment.

Consider the Hamiltonian ($\hbar = 1$ and implicit tensor products)

$$
\begin{aligned}
H &= H_{\mathrm{S}} + H_{\mathrm{B}} + H_{\mathrm{SB}} \\
&= \frac{\omega_{\mathrm{S}}}{2}\sigma_z + \omega_{\mathrm{B}}a^\dagger a + f\sigma_z(a + a^\dagger),
\end{aligned}
\tag{2}
$$

where $H_{\mathrm{S}}$ is the free system Hamiltonian, $H_{\mathrm{B}}$ is the free Hamiltonian of a monochromatic boson field, and $H_{\mathrm{SB}}$ is the interaction Hamiltonian. Furthermore, $\sigma_z$ is the third Pauli matrix and $a, a^\dagger$ are the bosonic annihilation and creation operators, respectively. The system resonance frequency is $\omega_{\mathrm{S}} \in \mathbb{R}$ and the bath resonance frequency is described by $\omega_{\mathrm{B}} \in \mathbb{R}$. The coupling strength between the system and the bath is given by $f \in \mathbb{R}$.

The goal of dynamical decoupling is to effectively remove the interaction Hamiltonian $H_{\mathrm{SB}}$, which causes the system to dephase, by acting on the system alone. To this end, we can perform a Carr–Purcell dynamical decoupling sequence [45], frequently interspersing the dynamics under $H$ by instantaneous Pauli $\sigma_x$ rotations on the system. To provide a mathematical description of this situation, we need to move to the density operator picture (Liouville space) and assume that the bath is initially in a thermal (Gibbs) state,

$$
\rho_{\mathrm{B}} = \frac{\mathrm{e}^{-\beta\omega_{\mathrm{B}}a^\dagger a}}{Z(\beta)},
\tag{3}
$$

where $Z(\beta) = \mathrm{tr}(\mathrm{e}^{-\beta\omega_\mathrm{B}a^\dagger a})$ is the grand canonical partition function, and for simplicity we fix the chemical potential $\mu$ to zero. This choice of initial state is physically motivated by the fact that, in the general scenario, one does not have any information about the bath, and the Gibbs state maximizes the entropy. The global initial state of system and bath will be a product state of the form $\rho = \rho_\mathrm{S} \otimes \rho_\mathrm{B}$, where $\rho_\mathrm{S}$ is an arbitrary system input state.

In this situation, the evolution of $\rho$ induced by the total Hamiltonian $H$ is described by a unitary evolution group defined by $\mathbf{Ad}_{\mathrm{e}^{-\mathrm{i}tH}}\rho \equiv \mathrm{e}^{-\mathrm{i}tH}\rho\mathrm{e}^{+\mathrm{i}tH}$. Leaving a more precise mathematical treatment to Section 4.1, it is known [46, 47] that this group is generated by the Liouville operator (or Liouvillian) corresponding to $H$, which we denote by $\mathbf{ad}_H \equiv [H, \cdot]$ [1]. That is,

$$\mathbf{Ad}_{\mathrm{e}^{-\mathrm{i}tH}} = \mathrm{e}^{-\mathrm{i}t\,\mathbf{ad}_H}, \tag{4}$$

and we will freely switch between these two notations from now on.

In the Liouville space, a Pauli $\sigma_x$ rotation on the system is then performed by applying the map $\mathbf{X} \equiv (\sigma_x \otimes I_\mathrm{B}) \cdot (\sigma_x \otimes I_\mathrm{B})$, where $I_\mathrm{B}$ is the identity on the bath (which we will omit in the following discussion). In this notation, the Carr–Purcell dynamical decoupling is described by the following evolution:

$$\mathbf{Ad}_{U_N(t)}\rho = \left(\mathbf{X}\,\mathbf{Ad}_{\mathrm{e}^{-\mathrm{i}\frac{t}{2N}H}}\,\mathbf{X}\,\mathbf{Ad}_{\mathrm{e}^{-\mathrm{i}\frac{t}{2N}H}}\right)^N \rho. \tag{5}$$

By a direct calculation, it can be shown that $\mathbf{Ad}_{U_N(t)}\rho$ is equivalent to

$$\mathbf{Ad}_{U_N(t)}\rho = \left(\mathrm{e}^{-\mathrm{i}\frac{t}{2N}\mathbf{ad}_{\sigma_x H\sigma_x}}\mathrm{e}^{-\mathrm{i}\frac{t}{2N}\mathbf{ad}_H}\right)^N \rho, \tag{6}$$

also see Eq. (55). Eq. (6) is nothing but a Trotter product formula in the Liouville space; therefore, in the limit of large $N$, the evolution $\mathbf{Ad}_{U_N(t)}\rho$ can effectively be described by

$$\mathbf{Ad}_{U_N(t)}\rho \xrightarrow{N\to\infty} \mathbf{Ad}_{T(t)}\rho, \tag{7}$$

where $\mathbf{Ad}_{T(t)}\rho = \mathbf{Ad}_{\mathrm{e}^{-\mathrm{i}t(\sigma_x H\sigma_x + H)/2}}\rho$. Since $\sigma_x\sigma_z\sigma_x = -\sigma_z$, we have $(\sigma_x H\sigma_x + H)/2 = \omega_\mathrm{B}a^\dagger a$ and thus

$$\mathbf{Ad}_{T(t)}\rho = \rho_\mathrm{S} \otimes \mathrm{e}^{-\mathrm{i}t\omega_\mathrm{B}a^\dagger a}\rho_\mathrm{B}\mathrm{e}^{+\mathrm{i}t\omega_\mathrm{B}a^\dagger a}. \tag{8}$$

Hence, in the limit $N \to \infty$, the evolution is indeed decoupled: the initial system state $\rho_\mathrm{S}$ is retained, and only the bath evolves. In practice, already for sufficiently large $N$, all interaction terms in the Hamiltonian do not affect the evolution, whence the spin and the field are effectively decoupled.

For finite $N$, of course, the decoupling is not exact. Since $H_\mathrm{B}$ and $H_\mathrm{SB}$ do not commute, there is a non-zero Trotter error, which determines the decoupling fidelity. Determining such an error—and, in particular, determining how it scales with the number $N$ of decoupling steps—is, for all practical applications, of primary importance.

To this purpose, we first notice that the targeted decoupled evolution group $\mathbf{Ad}_{T(t)}$ is generated by the operator $\mathbf{I}_\mathrm{S} \otimes \mathbf{ad}_{\omega_\mathrm{B}a^\dagger a}$, with $\mathbf{I}_\mathrm{S}$ being the identity map on the system, i.e. $\mathbf{I}_\mathrm{S}\rho_\mathrm{S} = \rho_\mathrm{S}$. Thus, for any system input state $\rho_\mathrm{S}$, we have

$$\mathbf{I}_\mathrm{S} \otimes \mathbf{ad}_{\omega_\mathrm{B}a^\dagger a}(\rho_\mathrm{S} \otimes \rho_\mathrm{B}) = \rho_\mathrm{S} \otimes \omega_\mathrm{B}[a^\dagger a, \rho_\mathrm{B}]$$
$$= 0 \tag{9}$$

---

[1]Precisely, as briefly discussed later in Section 4.1 (also see Ref. [47] and references therein), $\mathbf{ad}_H$ is the unique self-adjoint extension of $[H, \cdot]$ on the space of Hilbert–Schmidt operators.

since the Gibbs state $\rho_B$ commutes with $a^\dagger a$. Therefore, $\rho = \rho_S \otimes \rho_B$ is an eigenstate of $\mathbf{I}_S \otimes \mathbf{ad}_{\omega_B a^\dagger a}$ with eigenvalue zero and in fact, $\mathbf{Ad}_{T(t)}\rho = \rho$.

For pure states, the Trotter error for eigenstates of the target Hamiltonian has been studied in Ref. [31]: For two Hamiltonians $H_1, H_2$ and a state $|\varphi\rangle$ with $(H_1 + H_2)|\varphi\rangle = 0$, we have

$$\left\| \left( e^{-i\frac{t}{N}H_2} e^{-i\frac{t}{N}H_1} \right)^N |\varphi\rangle - |\varphi\rangle \right\| \le \frac{t^2}{2N} \left( \|H_1^2 |\varphi\rangle\| + \|H_2^2 |\varphi\rangle\| \right), \tag{10}$$

where $\||\varphi\rangle\| = \sqrt{\langle\varphi,\varphi\rangle}$ is the standard Euclidean norm of vectors. Here, we are actually dealing with mixed states. However, a crucial point of our analysis, explained in detail in Ref. [47], is the following: the estimate (10) for the Trotter product formula on pure states, along with many similar estimates, can be immediately generalized to mixed states as long as one replaces the Hamiltonian $H$ with the corresponding Liouvillian $\mathbf{ad}_H$, the evolution group $e^{-itH}$ with the corresponding group $\mathbf{Ad}_{e^{-itH}} = e^{-it\,\mathbf{ad}_H}$, and the Euclidean norm with the Hilbert–Schmidt norm $\|\rho\|_{HS} = \sqrt{\mathrm{tr}(\rho^\dagger \rho)}$. As such, we have

$$\left\| \left( e^{-i\frac{t}{N}\mathbf{ad}_{H_2}} e^{-i\frac{t}{N}\mathbf{ad}_{H_1}} \right)^N \rho - \rho \right\|_{HS} \le \frac{t^2}{2N} \left( \|\mathbf{ad}_{H_1}^2 \rho\|_{HS} + \|\mathbf{ad}_{H_2}^2 \rho\|_{HS} \right), \tag{11}$$

where $\mathbf{ad}_{H_1+H_2}\rho = 0$. We can use Eq. (11) to bound the efficiency of Trotterization in Eq. (7), where we Trotterize between $\mathbf{ad}_{H_1} = \frac{1}{2}\mathbf{ad}_H$ and $\mathbf{ad}_{H_2} = \frac{1}{2}\mathbf{ad}_{\sigma_x H \sigma_x}$.

To make our analysis independent of the chosen system input state, we fix $\rho_S = |+\rangle\langle+|$ with $|+\rangle = \frac{1}{\sqrt{2}}(|0\rangle + |1\rangle)$. Since this state is the one with the largest decoupling error [2], an upper bound for this state will serve as an upper bound for any state. To simplify our calculation, we first notice that any unitary $U$ gives $\|\mathbf{ad}_{UHU^\dagger}^2 \rho\|_{HS} = \|\mathbf{ad}_H^2(U^\dagger \rho U)\|_{HS}$. In our case, $U = \sigma_x \otimes I$ but we also have $\sigma_x \rho_S \sigma_x = \rho_S$. Therefore, the decoupling error can be bounded by

$$\|\mathbf{Ad}_{U_N(t)}\rho - \mathbf{Ad}_{T(t)}\rho\|_{HS} \le \frac{t^2}{4N} \|\mathbf{ad}_H^2 \rho\|_{HS}, \tag{12}$$

which can be computed explicitly, see Appendix C.3. The following bound is obtained:

$$\|\mathbf{Ad}_{U_N(t)}\rho - \mathbf{Ad}_{T(t)}\rho\|_{HS} < \frac{t^2}{8N} \max\left(4, |\omega_S|^2\right) \kappa(\beta, \omega_B, f), \tag{13}$$

where

$$\kappa(\beta, \omega_B, f) = 2\left(e^{\beta\omega_B} - 1\right)^{-1/2} \left(e^{\beta\omega_B} + 1\right)^{-3/2} \left[ 4|f|^2 e^{4\beta\omega_B}\left(29|f|^2 + \omega_B^2 + 1\right) \right.$$
$$\left. - 2e^{2\beta\omega_B}\left(2|f|^2\left(\omega_B^2 + 1\right) + 1\right) + e^{4\beta\omega_B} + 1 \right]^{1/2} \tag{14}$$

In particular, this implies $\|\mathbf{Ad}_{U_N(t)}\rho - \mathbf{Ad}_{T(t)}\rho\|_{HS} = \mathcal{O}(1/N)$ as expected from Trotterization. Furthermore, our bound reveals the dependency of the decoupling efficiency on the inverse temperature $\beta$. In fact, a tighter bound than Eq. (13) can be obtained, see Eq. (175) in the appendix.

We compare these findings with a numerical simulation in Fig. 1. Fig. 1(a) shows the decoupling error as a function of the number of decoupling pulses $N$, while Fig. 1(b) shows the decoupling error as a function of the inverse temperature $\beta$. In both cases, our bound

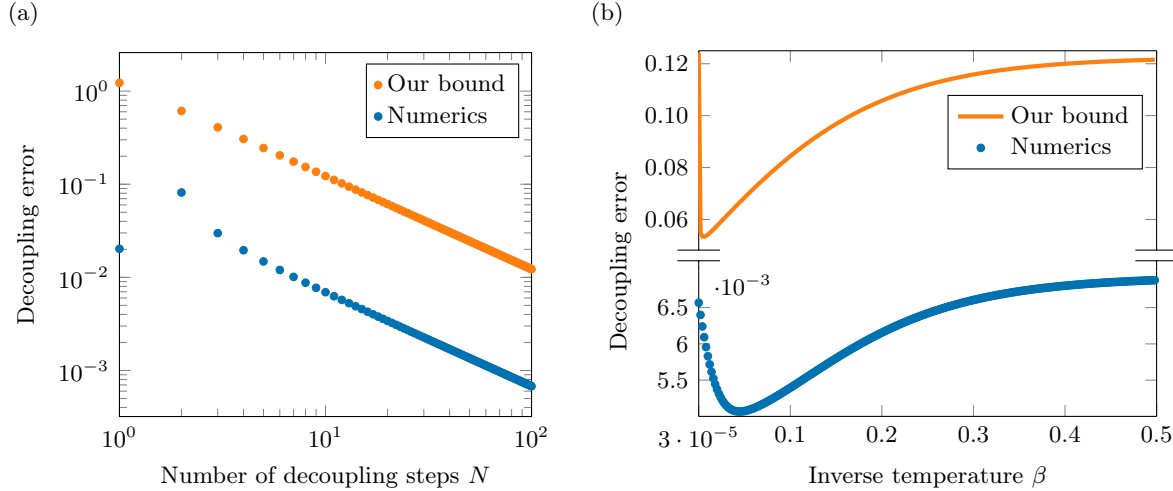

Figure 1: Carr-Purcell dynamical decoupling error for a qubit coupled to monochromatic boson field via a dephasing interaction. The Hamiltonian is given in Eq. (2), and the dynamical decoupling is performed by repetitive $\pi$–rotations (Pauli $\sigma_x$ pulses). The initial state is $|+\rangle\langle+| \otimes \rho_B$, where $\rho_B$ is the Gibbs state of the bosonic bath with inverse temperature $\beta$. We fix the total evolution time to $t = 1$ and choose $\omega_S = 1$, $\omega_B = 10$ and $f = 0.1$. The orange curve shows our analytical bound (175) and the blue curve shows a numerical simulation, where we truncated the bosonic field in Fock space at dimension $d = 10$. (a) Error as a function of the number $N$ of decoupling cycles for fixed $\beta = 1$. We see that the decoupling error decays as $1/N$. Our bound captures the asymptotic behaviour of the decoupling error. (b) Error as a function of the inverse temperature $\beta$ for fixed $N = 10$. We see that the decoupling error first decays with increasing $\beta$ and then increases as $\mathcal{O}(\sqrt{\beta})$. Finally, the error saturates at a constant due to finite $N$. The error decay for small $\beta$ is lightly visible in our bound (which scales as $\mathcal{O}(1/\sqrt{\beta})$ in this regime), and it correctly captures the $\mathcal{O}(\sqrt{\beta})$ scaling and eventual saturation for larger $\beta$.

captures the true behavior of the error. In the zero-temperature limit ($\beta \to \infty$), our bound reduces to

$$\|\mathbf{Ad}_{U_N(t)}\rho - \mathbf{Ad}_{T(t)}\rho\|_{\mathrm{HS}} \leq \frac{t^2}{8N}\Big(4\big(5 + 4\sqrt{2}\big)|f|^2 + 2\big(2 + \sqrt{2}\big)|f|\omega_B + 8\sqrt{2}|f|\omega_S + \sqrt{2}\omega_S^2\Big). \tag{15}$$

Of course, the model (2) is rather simplistic, and the treatment presented above relies on the fact that such a model is exactly solvable and can be decoupled with only two distinct decoupling operations—neither being true in the general case. Luckily, we will be able to overcome these mathematical difficulties and extend these results to general spin–boson models. This is precisely the content of the next sections.

# 3 General spin–boson models

In this section, we introduce the general class of models we consider in this paper and present the technical assumptions the models have to fulfil. Let us consider a quantum mechanical boson field with at most countably infinitely many energy modes, described in the momentum representation. A single excitation of the field is thus described by a single-particle Hilbert space equal to either $\mathbb{C}^d$, if there are $d$ energy modes, or $\ell^2$, the space of complex sequences $(\psi_k)_{k \in \mathbb{N}}$ such that $\sum_{k \in \mathbb{N}} |\psi_k|^2 < \infty$, in the case of infinitely many modes. In order to use a uniform notation, in both cases we shall denote said Hilbert space by $\mathfrak{h}$, and the corresponding momentum set shall be denoted by $K \subseteq \mathbb{N}$. The boson field is then described by the corresponding Bose–Fock space over $\mathfrak{h}$ [48, 49],

$$\mathcal{F} = \bigoplus_{n \in \mathbb{N}} S_n \mathfrak{h}^{\otimes n}, \tag{16}$$

where $S_n$ is the symmetrization operator. Physically, each element of $\mathcal{F}$ can be thought of as the superposition of completely symmetric states each with a different number $n$ of particles ranging from 0 to $\infty$. We remark that even for a finite number of boson modes (i.e. $\mathfrak{h} = \mathbb{C}^d$), this is always an infinite-dimensional space.

On this space, the free energy of the boson field is described, as usual, by the operator

$$H_{\mathrm{B}} = \sum_{k \in K} \omega_k a_k^\dagger a_k, \tag{17}$$

with $a_k, a_k^\dagger$ being the bosonic annihilation and creation operators, satisfying the usual bosonic commutation relations. $H_{\mathrm{B}}$ is a self-adjoint operator on $\mathcal{F}$. It is then known (see Appendix A) that $H_{\mathrm{B}}$ has a pure point spectrum composed by the set of all real numbers in the form $\sum_k n_k \omega_k$, where $(n_k)_{k \in K}$ is a sequence of integer numbers with finitely many of them being nonzero; correspondingly, $\mathcal{F}$ admits a complete orthonormal set of eigenvectors of $H_{\mathrm{B}}$, each of them being indexed by such sequences—the Fock states:

$$|\boldsymbol{n}\rangle = |n_{k_1}, n_{k_2}, \dots, n_{k_N}\rangle, \qquad k_1, \dots, k_N \in K. \tag{18}$$

More information can be found in Appendix A. A Fock state $|\boldsymbol{n}\rangle$ as in Eq. (18) corresponds to a configuration in which the wavenumbers $k_1, \dots, k_N$ have occupancy numbers $n_{k_1}, \dots, n_{k_N}$, and all other wavenumbers have zero occupancy number. We will use this basis to calculate all Hilbert–Schmidt norms involved in this paper; we refer to Appendix A for the details.

Throughout the paper, we will need the following two requirements:

**Assumption 3.1.** The modes $(\omega_k)_{k \in K}$ satisfy the following properties:

(i) they are bounded from below:
$$m \equiv \inf_{k \in K} \omega_k > 0; \tag{19}$$

(ii) for all $\beta > 0$, the following estimate holds:

$$\sum_{k \in K} e^{-\beta \omega_k} < \infty; \tag{20}$$

Assumption 3.1(i) is needed to avoid domain issues for the boson field Hamiltonian (17). Assumption 3.1(ii) (which, roughly speaking, entails that the modes $\omega_k$, when infinitely many, grow "sufficiently fast" as $k \to \infty$) is needed to ensure the condition

$$Z(\beta) := \operatorname{tr} e^{-\beta H_B} < \infty \tag{21}$$

for all $\beta > 0$, as recalled in Appendix B. Consequently, the Gibbs state

$$\rho_{\mathrm{B}} = \frac{e^{-\beta H_{\mathrm{B}}}}{\operatorname{tr} e^{-\beta H_{\mathrm{B}}}} \tag{22}$$

exists. As is known, $\rho_{\mathrm{B}}$ represents a thermal state with temperature $T$ obtained by $\beta = 1/(k_{\mathrm{B}}T)$, with $k_{\mathrm{B}}$ being the Boltzmann constant. Of course, both Assumptions 3.1(i)–(ii) are trivial in the case of finitely many modes; for countably many modes, the estimate (20) essentially requires that the modes $\omega_k$ diverge to infinity more quickly than logarithmically in $k$.

We remark that the Gibbs state is often presented as $\rho_{\mathrm{B}} = \frac{e^{-\beta(H_{\mathrm{B}}-\mu\mathcal{N})}}{\operatorname{tr} e^{-\beta(H_{\mathrm{B}}-\mu\mathcal{N})}}$, where $\mathcal{N} = \sum_{k \in K} a_k^\dagger a_k$ is the number operator and $\mu \in \mathbb{R}$ is the chemical potential [49]. In this case, it suffices to require $m > -\infty$ and $\mu < m$. However, as already done in the example considered in Section 2, there is no loss of generality in setting $\mu = 0$ by shifting the zero-point energy of the free boson field, thus yielding the requirement $m > 0$ that is given in Assumption 3.1(i). This will make our equations simpler; the general case can always be recovered via a shift of the modes $\omega_k$. This is because we are assuming discrete modes $\omega_k$, so that only finitely many of them can be negative. For instance, this shift has to be done for Ohmic spectra.

We shall consider the interaction between this field and a two-level system. The corresponding Hilbert space is thus $\mathcal{H} = \mathbb{C}^2 \otimes \mathcal{F}$, and the Hamiltonian shall be assumed to have the following expression (tensor product understood):

$$H = H_{\mathrm{S}} + \sum_{k \in K} \omega_k a_k^\dagger a_k + \sum_{k \in K} \left( f_k^* B^\dagger a_k + f_k B a_k^\dagger \right), \tag{23}$$

with $H_{\mathrm{S}} = H_{\mathrm{S}}^\dagger$ and $B$ being arbitrary $2 \times 2$ matrices, and $f = (f_k)_{k \in K}$ weighting the coupling between the two-level system and the $k$-th mode of the field. In fact, our results only rely on the form of the Hamiltonian in Eq. (23). Our bounds can be applied as long as the system Hilbert space is finite-dimensional and the Hamiltonian is of the form of Eq. (23). For instance, that would be the case e.g. in Dicke models if only one level of a qu$d$it is coupled to the bath, or if one considers a Tavis–Cummings-type model in which all atoms interact with the bosonic bath via a single coupling operator $B$. Notice that a generalization to arbitrary finite-dimensional systems is immediate by replacing the operator $B$ with higher-dimensional operators $B_{k,j}$, where each bath mode $k$ can couple differently to the system energy level $j$. However, since this will blow up the notation while not giving any new physical insights, we stick to the simplified qubit model in Eq. (23). For the couplings $(f_k)_{k \in K}$ we assume the following property:

**Assumption 3.2.** The couplings $f = (f_k)_{k \in K}$ satisfy

$$\sum_{k \in K} |f_k|^2 < \infty. \tag{24}$$

Again, this assumption is only nontrivial for countably many modes.

Under Assumption 3.2, it can be shown [2] that the spin–boson Hamiltonian (23) is a well-defined, self-adjoint operator on $\mathcal{H}$ with domain $\mathrm{Dom}\,H = \mathbb{C}^2 \otimes \mathrm{Dom}\,H_B$, belonging to the class of (generalized) spin–boson models introduced at the beginning of the paper [33, 34, 35, 36, 37, 38]. Some particular examples include:

- the dephasing model considered in Section 2, when $d = 1$ (single mode), $H_S = \frac{\omega_S}{2}\sigma_z$ and

$$B = \sigma_z = \begin{pmatrix} 1 & 0 \\ 0 & -1 \end{pmatrix}; \tag{25}$$

- the Jaynes–Cummings model, when $d$ and $H_S$ are again as above, and

$$B = \sigma_- = \begin{pmatrix} 0 & 0 \\ 1 & 0 \end{pmatrix}; \tag{26}$$

- the quantum Rabi model, when $d$ and $H_S$ are again as above, and

$$B = \sigma_x = \begin{pmatrix} 0 & 1 \\ 1 & 0 \end{pmatrix}, \tag{27}$$

as well as the generalizations of all these models to boson fields with at most countably many modes.

With dynamical decoupling, we aim to decouple the qubit from the degrees of freedom of the boson bath. The discussion in Section 2 shows that this is described by the Trotterization of rotated versions of the Hamiltonian (23). In the next section, we will explain how Trotterization works for more than two Hamiltonians, give bounds on the Trotter error, and translate them to the density operator picture.

# 4 The Trotter error for mixed quantum states

In this section, we provide a general Trotter error bound that can be applied to compute the efficiency of dynamical decoupling. Since the description of dynamical decoupling for thermal—thus mixed—states is to be formulated in the density operator picture, like already shown in the example in Section 2, we will need some preliminaries on mixed quantum states first.

## 4.1 Quantum mechanics in the Liouville space—from pure to mixed quantum states

For the purposes of this paper—more generally, whenever taking into account models of field–matter interaction—we will need to deal with unbounded Hamiltonians acting on infinite-dimensional spaces. We shall thus start by recalling some basic related notions. This will help us to understand how to translate results from pure to mixed quantum states.

---

[2]Assumption 3.2 may be, in fact, relaxed to accommodate a larger class of coupling functions, as shown in Refs. [39, 40, 41]; in such cases, however, the domain of $H$ acquires a nontrivial dependence on the coupling.

Let $\mathcal{H}$ be an infinite-dimensional Hilbert space. As already seen in the previous section, linear operators $H$ on $\mathcal{H}$ are then generally defined on a subspace of $\mathcal{H}$, which is referred to as the *domain* of $H$ and will be hereafter denoted by $\mathrm{Dom}\, H$. The operators for which $\sup_{\||\psi\rangle\|=1} \|H\,|\psi\rangle\| < \infty$ are called *bounded* and their domain is the whole Hilbert space. All other operators are called *unbounded*. In the following, we will denote the space of bounded linear operators on $\mathcal{H}$ by $\mathcal{B}(\mathcal{H})$. In particular, the Hamiltonian of a quantum system is the (generally unbounded) self-adjoint operator, i.e. satisfying $H = H^\dagger$, which uniquely specifies the time-evolution of the corresponding *pure* quantum system via the Schrödinger equation ($\hbar = 1$),

$$\mathrm{i}\frac{\mathrm{d}}{\mathrm{d}t}|\Psi(t)\rangle = H\,|\Psi(t)\rangle, \qquad |\Psi(t_0)\rangle = |\Psi_0\rangle \in \mathrm{Dom}\, H. \tag{28}$$

That is, there exists a (strongly continuous) unitary propagator $U(t)$, $t \in \mathbb{R}$, such that the function $|\Psi(t)\rangle = U(t)|\Psi_0\rangle$ is the unique solution of the problem (28). Furthermore, $U(t) = \exp(-\mathrm{i}tH)$, where the exponentiation is to be understood in the spectral sense.

In this paper we are instead interested in the dynamics of (possibly) *mixed* states. A mixed state of a quantum system is represented by an operator $\rho \in \mathcal{B}(\mathcal{H})$ satisfying

$$\rho = \rho^\dagger, \quad \rho \geq 0, \quad \mathrm{tr}\,\rho = 1. \tag{29}$$

In the Schrödinger picture, the evolution of such states is known to be given by $t \mapsto U(t)\rho U(t)^\dagger$. It is thus natural to look for the mixed-state counterpart of the Schrödinger equation. For bounded Hamiltonians, this would be the quantum Liouville equation:

$$\mathrm{i}\frac{\mathrm{d}}{\mathrm{d}t}\rho(t) = [H, \rho(t)]; \tag{30}$$

we refer to Ref. [46] for an extensive discussion. However, since we are dealing with unbounded Hamiltonians here, Eq. (30) is to be taken with additional care. For a rigorous mathematical discussion of the quantum Liouville equation in this case, see Ref. [47] and references therein. We informally summarize Ref. [47] as follows:

(i) The space of operators

$$\mathcal{L}(\mathcal{H}) := \left\{ S \in \mathcal{B}(\mathcal{H}) : \|S\|_{\mathrm{HS}} < \infty \right\}, \tag{31}$$

where $\|\cdot\|_{\mathrm{HS}}$ is the Hilbert–Schmidt norm,

$$\|S\|_{\mathrm{HS}}^2 := \sum_{n \in \mathbb{N}} \|S\,|e_n\rangle\|^2, \tag{32}$$

is a Hilbert space with respect to the Hilbert–Schmidt scalar product $\langle S, T\rangle_{\mathrm{HS}} := \mathrm{tr}(S^\dagger T)$. Above, $(|e_n\rangle)_{n \in \mathbb{N}} \subset \mathcal{H}$ is any complete orthonormal basis of $\mathcal{H}$. All density operators are in $\mathcal{L}(\mathcal{H})$;

(ii) The superoperator (i.e. operator acting on $\mathcal{L}(\mathcal{H})$) $\mathbf{Ad}_{U(t)} : \mathcal{L}(\mathcal{H}) \to \mathcal{L}(\mathcal{H})$ defined by

$$\mathbf{Ad}_{U(t)}\rho := U(t)\rho U(t)^\dagger \tag{33}$$

for all $t \in \mathbb{R}$, is a (strongly continuous) unitary propagator (with respect to the Hilbert–Schmidt inner product).

(iii) For suitable density operators, i.e. $\rho \in \mathrm{Dom}\,\mathbf{ad}_H$ with

$$\mathrm{Dom}\,\mathbf{ad}_H = \big\{ S \in \mathcal{L}(\mathcal{H}) : S\,\mathrm{Dom}\,H \subset \mathrm{Dom}\,H, \sum_{n\in\mathbb{N}} \|[H,S]\,|e_n\rangle\|^2 < \infty \big\}, \qquad (34)$$

$\mathbf{Ad}_{U(t)}\rho$ is the unique solution to the quantum Liouville equation (30) with initial condition $\rho(0) = \rho$, and is generated by the unique self-adjoint extension of the superoperator $[H,\cdot]$ to $\mathrm{Dom}\,\mathbf{ad}_H$, which we call the *Liouvillian* and denote by $\mathbf{ad}_H$ [3].

Combining these three statements allows us to translate results from pure quantum states to mixed quantum states, according to the following simple substitution rules:

1. the norm of state-vectors $\|\cdot\|$ is replaced by the Hilbert–Schmidt norm of density operators $\|\cdot\|_{\mathrm{HS}}$;

2. the unitary evolution group $U(t)$ on $\mathcal{H}$ is replaced by the unitary evolution group $\mathbf{Ad}_{U(t)}$ on the Liouville space $\mathcal{L}(\mathcal{H})$;

3. the Hamiltonian operator $H$ generating $U(t)$ via $U(t) = \mathrm{e}^{-\mathrm{i}tH}$, with domain $\mathrm{Dom}\,H$, is replaced by the Liouvillian superoperator $\mathbf{ad}_H$ generating $\mathbf{Ad}_{U(t)}$ via $\mathbf{Ad}_{U(t)} = \mathrm{e}^{-\mathrm{i}t\mathbf{ad}_H}$, with domain $\mathrm{Dom}\,\mathbf{ad}_H$ as per Eq. (34) [4].

Similar rules are explicitly proven in Ref. [47, Section 3.3] for the square of the Liouvillian.

In the next subsection, we will see a direct application of these rules in the context of the Trotter product formula. As we have seen in Section 2, the process of dynamical decoupling can be understood in terms of Trotterization. In fact, through the lens of Trotterization, dynamical decoupling has some very favourable properties. Here, the only infinite-dimensional degrees of freedom come from the bath, while the system is usually assumed to be finite-dimensional so that Trotter always converges with a low error on the system degrees of freedom [50]. The bath is assumed to be initially in a Gibbs state, which acts as a regularizer of the bath infinities: under suitable conditions, it ensures the finiteness of high moments of the Liouvillian. This is remarkable because the Liouvillian might be a doubly unbounded operator even if its Hamiltonian is only semi-unbounded.

## 4.2 The Trotter product formula for multiple Hamiltonians

In their different incarnations, Trotter product formulas allow us to express the evolution generated by the sum of two or more operators as the limit of the iterated Trotter evolution $(\mathrm{e}^{-\mathrm{i}tH_1/N}\mathrm{e}^{-\mathrm{i}tH_2/N})^N$ as $N \to \infty$. Let us recall the following result about the existence of this limit [16]:

**Theorem 4.1.** *Let $H_j$ $(j = 0, \ldots, L-1)$ be self-adjoint operators on a Hilbert space $\mathcal{H}$, with domains $\mathrm{Dom}\,H_j$, and assume that their sum $\sum_{j=0}^{L-1} H_j$ is essentially self-adjoint on the domain $\bigcap_j \mathrm{Dom}\,H_j$. Then the Trotter product formula converges in the strong sense, that is:*

---

[3] We remark that the operator $\mathbf{ad}_H$ is denoted by $\mathbf{H}$ in Ref. [47].

[4] We remark that for the spin–boson models defined in Section 3, the Gibbs state $\rho_\mathrm{B}$ is in the core of $\mathbf{ad}_H$, where it acts as the commutator, see Ref. [47, Section 4]. Since we only consider the Gibbs state as the bath input state, we can safely write $\mathbf{ad}_H\rho_\mathrm{S} \otimes \rho_\mathrm{B} = [H, \rho_\mathrm{S} \otimes \rho_\mathrm{B}]$ as has been done in Section 3.

*for all* $|\Psi\rangle \in \mathcal{H}$ [5],

$$\left(\prod_{j=0}^{L-1} e^{-i\frac{t}{N}H_j}\right)^N |\Psi\rangle \xrightarrow{N\to\infty} e^{-it\sum_{j=0}^{L-1}H_j} |\Psi\rangle. \tag{35}$$

*Furthermore, the limit above is uniform in t on compact time intervals.*

*Proof.* This was first proven by Kato for two operators [51] and later in Ref. [16, Theorem 3.1] for multiple operators (as stated here). $\qquad\square$

For practical purposes, knowing that the limit above holds is not enough—one needs to evaluate the rate of convergence of the aforementioned limit in the number $N$ of Trotter steps. The bounds commonly used in the literature [50] usually depend on the operator norm of the commutator between $H_1$ and $H_2$, thus clearly not being adaptable to unbounded Hamiltonians [52] and, in any case, possibly overestimating the error, as extensively discussed in Ref. [31] (also see Refs. [32, 53]). In the latter paper, a *state-dependent* bound for the Trotter product formula was found, cf. [31, Theorem 1].

The following result consists of a generalization of said result to the case of more than two Hamiltonians. This is crucial for dynamical decoupling, where the number of Hamiltonians in the Trotter product coincides with the size $L$ of the decoupling group: in fact, already for a single qubit—save from particular examples as the one considered in Section 2—we generally have $L = 4$, and this number increases exponentially with the number of qubits.

**Theorem 4.2.** *Let $H_j$ ($j = 0, \ldots, L-1$) be self-adjoint operators on a Hilbert space $\mathcal{H}$, with domains $\operatorname{Dom} H_j$, and assume that their sum $\sum_{j=0}^{L-1} H_j$, with domain $\bigcap_j \operatorname{Dom} H_j$, admits an eigenvalue $h$ with corresponding eigenstate $|\varphi\rangle$. Also assume $|\varphi\rangle \in \bigcap_{j=0}^{L-1} \operatorname{Dom} H_j^2 \cap \operatorname{Dom} H_j \sum_{i=0}^{j-1} H_i$. Then the state-dependent Trotter error*

$$\xi_N(t; |\varphi\rangle) = \left\| \left(\prod_{j=0}^{L-1} e^{-i\frac{t}{N}H_j}\right)^N |\varphi\rangle - e^{-ith} |\varphi\rangle \right\| \tag{36}$$

*is bounded by*

$$\xi_N(t; |\varphi\rangle) \le \frac{t^2}{N} \sum_{k=0}^{L-1} \left( \frac{1}{2} \left\| H_k(g_k)^2 |\varphi\rangle \right\| + \left\| H_k(g_k) \sum_{i=0}^{k-1} H_i(g_i) |\varphi\rangle \right\| \right). \tag{37}$$

*Here, $H_k(g_k) = H_k - hg_k$ with $g_k \in \mathbb{R}$, such that $\sum_{k=0}^{L-1} g_k = 1$.*

*Proof.* The proof follows the same steps as the proof of Ref. [31, Theorem 1]. Let us first consider the case $h = 0$, so that the target evolution $e^{-ith} |\varphi\rangle$ becomes the identity on $|\varphi\rangle$. We notice that the Trotter unitary for $L$ Hamiltonians,

$$U_N(Lt) = \left( e^{-i\frac{t}{N}H_{L-1}} e^{-i\frac{t}{N}H_{L-2}} \ldots e^{-i\frac{t}{N}H_0} \right)^N, \tag{38}$$

---

[5]Here we are making a slight abuse of notation to avoid cumbersome equations: the right-hand side of Eq. (35) should have the topological closure $\overline{\Sigma_j H_j}$ at the exponent.

is generated by a piecewise constant, time-dependent Hamiltonian

$$
\tilde{H}(s) = \begin{cases} H_0, & s \in \left[0, \frac{t}{N}\right), \\ H_1, & s \in \left[\frac{t}{N}, \frac{2t}{N}\right), \\ \vdots & \vdots \\ H_{L-1}, & s \in \left[\frac{(L-1)t}{N}, \frac{Lt}{N}\right), \end{cases} \tag{39}
$$

which can be extended periodically, $\tilde{H}\left(s + \frac{Lt}{N}\right) = \tilde{H}(s)$. Thus, $\tilde{H}(s)$ is a family of self-adjoint and locally integrable operators. For this reason, Ref. [54, Lemma 1] applies and we can write

$$
[U_N(s) - I] \ket{\varphi} = -\mathrm{i}S(s) \ket{\varphi} - \int_0^s \mathrm{d}u \, U_N(s) U_N^\dagger(u) \tilde{H}(u) S(u) \ket{\varphi}, \tag{40}
$$

where $S(s)$, the integral action, at the time step $j$ reads

$$
\begin{aligned} S(s) \ket{\varphi} &= \int_0^s \mathrm{d}u \, \tilde{H}(u) \ket{\varphi} \\ &= \left(s - \frac{jt}{N}\right) H_{j \bmod L} \ket{\varphi} + \frac{t}{N} \sum_{i=0}^{j-1} H_{i \bmod L} \ket{\varphi}. \end{aligned} \tag{41}
$$

Also see Ref. [31, Lemma 12]. In explicit terms, $S(s)$ is written as

$$
S(s) \ket{\varphi} = \begin{cases} sH_0 \ket{\varphi}, & s \in \left[0, \frac{t}{N}\right), \\ \left[\left(s - \frac{t}{N}\right) H_1 + \frac{t}{N} H_0\right] \ket{\varphi}, & s \in \left[\frac{t}{N}, \frac{2t}{N}\right), \\ \left[\left(s - \frac{2t}{N}\right) H_2 + \frac{t}{N} \left(H_0 + H_1\right)\right] \ket{\varphi}, & s \in \left[\frac{2t}{N}, \frac{3t}{N}\right) \\ \vdots & \vdots \\ \left[\left(s - \frac{(L-1)t}{N}\right) H_{L-1} + \frac{t}{N} \sum_{i=1}^{L-2} H_i\right] \ket{\varphi}, & s \in \left[\frac{(L-1)t}{N}, \frac{Lt}{N}\right), \end{cases} \tag{42}
$$

which is again extended periodically. Therefore, at the boundary of each Trotter cycle ($j = ML$ for $M = 1, \ldots, N$),

$$
S\left(\frac{MLt}{N}\right) \ket{\varphi} = \frac{Mt}{N} h \ket{\varphi} = 0. \tag{43}
$$

By evaluating Eq. (40) for a single Trotter cycle, i.e. $U_{N=1}$ at $s = Lt/N$ and inserting Eq. (43),

$$
\left\| [U_1(Lt/N) - I] \ket{\varphi} \right\| \leq \int_0^{Lt/N} \mathrm{d}u \, \| \tilde{H}(u) S(u) \ket{\varphi} \|, \tag{44}
$$

where we also used the triangle inequality to move the norm inside the integral, and the fact that the norm is unitarily invariant. By a standard telescoping sum,

$$
\left[U_1(Lt/N)\right]^N - I = \sum_{k=0}^{N-1} \left[U_1(Lt/N)\right]^k \left[U_1(Lt/N) - I\right] \tag{45}
$$

we obtain

$$
\xi_N(t; \ket{\varphi}) \leq N \left\| [U_1(Lt/N) - I] \ket{\varphi} \right\|. \tag{46}
$$

By inserting there Eq. (44) and the explicit form of the integral action $S(s)$, we get

$$
\begin{aligned}
\xi_N(t; |\varphi\rangle) &\leq N \sum_{k=0}^{L-1} \int_{k\frac{t}{N}}^{(k+1)\frac{t}{N}} \mathrm{d}s \, \|H(s)S(s) |\varphi\rangle\| \\
&= N \sum_{k=0}^{L-1} \int_{k\frac{t}{N}}^{(k+1)\frac{t}{N}} \mathrm{d}s \left\| \left[ \left(s - \frac{kt}{N}\right) H_k^2 + \frac{t}{N} H_k \sum_{i=0}^{k-1} H_i \right] |\varphi\rangle \right\| \\
&\leq N \sum_{k=0}^{L-1} \left( \int_{k\frac{t}{N}}^{(k+1)\frac{t}{N}} \left\| \left(s - \frac{kt}{N}\right) H_k^2 |\varphi\rangle \right\| \mathrm{d}s + \int_{k\frac{t}{N}}^{(k+1)\frac{t}{N}} \left\| \frac{t}{N} H_k \sum_{i=0}^{k-1} H_i |\varphi\rangle \right\| \mathrm{d}s \right) \\
&= \frac{t^2}{N} \sum_{k=0}^{L-1} \left( \frac{1}{2} \|H_k^2 |\varphi\rangle\| + \left\| H_k \sum_{i=0}^{k-1} H_i |\varphi\rangle \right\| \right).
\end{aligned}
\tag{47}
$$

This concludes the proof in the case $h = 0$. The case of a general eigenvalue $h$ follows by simply performing the replacement $H_k \to H_k(g_k) \equiv H_k - hg_k$, where $\sum_{k=0}^{L-1} g_k = 1$.  □

While apparently only valid for pure quantum states, Theorem 4.2 and its proof only employ concepts that refer to the underlying Hilbert space structure (inner product, norm, unitarity, unitary norm equivalence, self-adjointness). Therefore, by directly applying the three substitution rules listed at the end of Section 4.1 (more details can be found in Ref. [47]) that Theorem 4.2 can be directly extended to the Liouville space. We state explicitly this fact in the case $h = 0$, which will suffice for our purposes:

**Corollary 4.3.** *Let $\mathbf{ad}_{H_j}$ ($j = 0, \dots, L-1$) be self-adjoint operators on the Hilbert space $\mathcal{L}(\mathcal{H})$ with domains $\mathrm{Dom}\,\mathbf{ad}_{H_j}$. Furthermore, let the sum $\sum_{j=0}^{L-1} \mathbf{ad}_{H_j}$ with domain $\bigcap_j \mathrm{Dom}\,\mathbf{ad}_{H_j}$ admit an eigenvalue $h = 0$ with corresponding eigen-density operator $\rho$. If $\rho \in \bigcap_{j=0}^{L-1} \mathrm{Dom}\,\mathbf{ad}_{H_j}^2 \cap \mathrm{Dom}\,\mathbf{ad}_{H_j} \sum_{i=0}^{j-1} \mathbf{ad}_{H_i}$, we can bound the Trotter error*

$$
\xi_N(t; \rho) = \left\| \left( \prod_{j=0}^{L-1} \mathrm{e}^{-\mathrm{i}\frac{t}{N}\mathbf{ad}_{H_j}} \right)^N \rho - \rho \right\|_{\mathrm{HS}}
\tag{48}
$$

*by means of Theorem 4.2 as*

$$
\xi_N(t; \rho) \leq \frac{t^2}{N} \sum_{k=0}^{L-1} \left( \frac{1}{2} \|\mathbf{ad}_{H_k}^2 \rho\|_{\mathrm{HS}} + \left\| \mathbf{ad}_{H_k} \sum_{i=0}^{k-1} \mathbf{ad}_{H_i} \rho \right\|_{\mathrm{HS}} \right).
\tag{49}
$$

Notice that $\mathbf{ad}_H^2$ is to be understood as the unique self-adjoint extension of the superoperator $[H, [H, \cdot]]$. See Ref. [47, Section 3.3] for details.

As we will see in the next section, Corollary 4.3 will enable us to compute the error on dynamical decoupling for the general class of models considered in Section 3.

*Remark* 4.4. From a physical perspective, it seems more natural to consider the trace norm $\|X\|_{\mathrm{tr}} = \mathrm{tr}(\sqrt{X^\dagger X})$ instead of the Hilbert–Schmidt norm distance to quantify the decoupling error as it corresponds to physically measurable quantities. Furthermore, one would mostly be interested in the distance of the *reduced system* dynamics under dynamical decoupling to the free decoupled *system* evolution. We remark that such a result can indeed be obtained as a simple corollary from our Theorem 4.2: It follows from Ref. [55, Lemma 5.1] that the result in

Corollary 4.3 also holds when replacing the Hilbert–Schmidt norm with the trace norm. One can then use the fact that the partial trace is a linear contraction to obtain, under the same assumptions as in Corollary 4.3,

$$\left\| \mathrm{tr}_{\mathrm{B}} \left[ \left( \prod_{j=0}^{L-1} \mathrm{e}^{-\mathrm{i}\frac{t}{N}\mathbf{ad}_{H_j}} \right)^N \rho \right] - \rho_{\mathrm{S}} \right\|_{\mathrm{tr}} \leq \frac{t^2}{N} \sum_{k=0}^{L-1} \left( \frac{1}{2} \left\| \mathbf{ad}_{H_k}^2 \rho \right\|_{\mathrm{tr}} + \left\| \mathbf{ad}_{H_k} \sum_{i=0}^{k-1} \mathbf{ad}_{H_i} \rho \right\|_{\mathrm{tr}} \right). \quad (50)$$

Here, $\rho = \rho_{\mathrm{S}} \otimes \rho_{\mathrm{B}}$ is a product state acting on the bipartite Hilbert space $\mathcal{H} = \mathcal{H}_{\mathrm{S}} \otimes \mathcal{H}_{\mathrm{B}}$. While this result is valuable from an abstract perspective, the bound in Eq. (50) is not computable in practice as the operators $\mathbf{ad}_{H_k}\mathbf{ad}_{H_i}\rho$ are not necessarily positive, thus making their corresponding trace norms not explicit. This is why we use the Hilbert–Schmidt norm to quantify the decoupling error.                                                                      △

# 5   Efficiency of dynamical decoupling for spin–boson models

In this section we discuss the procedure of dynamical decoupling for the general spin–boson models introduced before. We then present our main results on the dynamical decoupling of these models: a general-purpose recipe for obtaining bounds for the decoupling error.

## 5.1   Dynamical decoupling for spin–boson models

Physically, the model (23) describes a qubit system that is coupled to a bosonic bath, which introduces noise to the system. We shall assume $\mathrm{tr}(H_{\mathrm{S}}) = 0 = \mathrm{tr}(B)$, which is a standard assumption in the context of dynamical decoupling. Both conditions, however, are to ease the presentation and could be relaxed:

- if $\mathrm{tr}(H_{\mathrm{S}}) \neq 0$, we would simply get an additional global phase in the target dynamics which would not affect our results;

- if $\mathrm{tr}(B) \neq 0$, one can always rewrite the Hamiltonian (23) in the form $H = H_{\mathrm{S}} + \tilde{H}_{\mathrm{B}} + \sum_{k \in K} \left( f_k^* \tilde{B}^\dagger a_k + f_k \tilde{B} a_k^\dagger \right)$, where now $\mathrm{tr}(\tilde{B}) = 0$ and the the transformed bath Hamiltonian $\tilde{H}_{\mathrm{B}} = \sum_{k \in K} \left( \tilde{a}_k^\dagger \tilde{a}_k - \omega_k^{-1}|f_k|^2|\mathrm{tr}(B)|^2 \right)$ is given in terms of shifted bosonic operators defined by $\tilde{a}_k = \omega_k^{1/2} a_k + \omega_k^{-1/2} f_k \mathrm{tr}(B)$. The additional term $\omega_k^{-1}|f_k|^2|\mathrm{tr}(B)|^2$ in $\tilde{H}_{\mathrm{B}}$ commutes with everything and does not affect our results.

The goal of dynamical decoupling (DD) is then to effectively remove the interaction Hamiltonian $H_{\mathrm{I}} = \sum_{k \in K} \left( f_k^* B^\dagger a_k + f_k B a_k^\dagger \right)$ and the system Hamiltonian $H_{\mathrm{S}}$ by means of strong and fast controls on the system alone. That is, by only acting on $\mathcal{H}_{\mathrm{S}}$, we want to achieve the following:

$$\mathrm{e}^{-\mathrm{i}tH} \xrightarrow{DD} \mathrm{e}^{-\mathrm{i}tI_{\mathrm{S}} \otimes H_{\mathrm{B}}}, \quad (51)$$

where $I_{\mathrm{S}}$ denotes the identity on $\mathcal{H}_{\mathrm{S}}$. To describe the system controls, we define a set $\mathscr{V} = \{\mathbf{V}_j\}_{j=0}^{L-1}$ called the "decoupling set". Its elements are called "pulses" or "unitary kicks" and are unitary operations. They are of the form $\mathbf{V}_j = (v_j \otimes I_{\mathrm{B}}) \cdot (v_j^\dagger \otimes I_{\mathrm{B}})$, where the $v_j$'s are unitary matrices and $I_{\mathrm{B}}$ denotes the identity on the bath Hilbert space $\mathcal{H}_{\mathrm{B}}$. Furthermore, we require the so-called "decoupling condition": $\{v_j\}_{j=0}^{L}$ generates a unitary group that acts

irreducibly, i.e. only the identity $v_0 = I_S$ commutes with the entire group. By Schur's lemma, this is equivalent to

$$\frac{1}{L} \sum_{j=0}^{L-1} v_j A v_j^{\dagger} = \frac{\mathrm{tr}(A)}{\dim(\mathcal{H}_S)} I_S, \quad \text{for all } A \in \mathcal{B}(\mathcal{H}_S), \tag{52}$$

where $\mathcal{B}(\mathcal{H}_S)$ is the set of linear operators acting on $\mathcal{H}_S$. We would like to remark that Eq. (52) is sometimes also referred to as a "twirl over a unitary 1-design" in the case in which $\mathscr{V}$ does not admit a group structure.

To describe the process of dynamical decoupling, we will have to go to the density operator picture and thus to the Liouville space, as described in Section 4.1. Here, dynamical decoupling means interspersing the dynamics under the Liouville operator associated with $H$ by the decoupling pulses, i.e.

$$\mathbf{Ad}_{U_N(t)} \rho_S \otimes \rho_B = \left( \prod_{j=0}^{L-1} \mathbf{V}_j e^{-\mathrm{i}\frac{t}{LN} \mathbf{ad}_H} \mathbf{V}_j^{\dagger} \right)^N (\rho_S \otimes \rho_B)$$

$$= \left( \prod_{j=0}^{L-1} e^{-\mathrm{i}\frac{t}{LN} \mathbf{V}_j \mathbf{ad}_H \mathbf{V}_j^{\dagger}} \right)^N (\rho_S \otimes \rho_B) \tag{53}$$

where $N \in \mathbb{N}$, $\rho_S$ is a density operator on $\mathcal{H}_S$, and $\rho_B$ is assumed to be the Gibbs state, see Eq. (22). This is the commonly assumed initial state for dynamically decoupling bosonic baths; physically, as the Gibbs state maximizes the entropy, it reflects the fact that we do not have any information about the bath.

Under the assumption (52), the procedure in Eq. (53) removes the interaction Hamiltonian, and an initial product state $\rho = \rho_S \otimes \rho_B$ stays a product state *approximately* after the evolution under $\mathbf{Ad}_{U_N(t)}$. To see this, notice that Eq. (53) is essentially a Trotter product, where one Trotterizes between operators $\frac{1}{L}\mathbf{ad}_{H_j} = \frac{1}{L}\mathbf{V}_j \mathbf{ad}_H \mathbf{V}_j^{\dagger}$. Therefore, we know that, if $\frac{1}{L} \sum_{j=0}^{L-1} \mathbf{ad}_{H_j}$ is essentially self-adjoint (with respect to the Hilbert–Schmidt product) on the domain $\bigcap_{j=0}^{L-1} \mathrm{Dom}\,\mathbf{ad}_{H_j}$, Trotter converges on all input states, see Theorem 4.1. Then,

$$\lim_{N \to \infty} \mathbf{Ad}_{U_N(t)} \rho = e^{-\mathrm{i}t\frac{1}{L} \sum_{j=0}^{L-1} \mathbf{ad}_{H_j}} \rho. \tag{54}$$

This can be computed explicitly: If we define $U_j = v_j \otimes I$, we have

$$\mathbf{ad}_{H_j}\rho = \mathbf{Ad}_{U_j} \mathbf{ad}_H \mathbf{Ad}_{U_j^{\dagger}} \rho$$

$$= \mathbf{Ad}_{U_j}[H, U_j^{\dagger}\rho U_j]$$

$$= U_j H U_j^{\dagger} \rho U_j U_j^{\dagger} - U_j U_j^{\dagger} \rho U_j H U_j^{\dagger}$$

$$= \mathbf{ad}_{U_j H U_j^{\dagger}} \rho \tag{55}$$

and thus

$$\frac{1}{L} \sum_{j=0}^{L-1} \mathbf{ad}_{H_j} \rho = \frac{1}{L} \sum_{j=0}^{L-1} [U_j H U_j^{\dagger}, \rho_S \otimes \rho_B]$$

$$= \frac{1}{L} \sum_{j=0}^{L-1} \Big( [v_j H_S v_j^\dagger, \rho_S] \otimes \rho_B$$
$$+ \sum_{k \in K} f_k^* \big( [v_j B^\dagger v_j^\dagger, \rho_S] \otimes a_k \rho_B + \rho_S v_j B^\dagger v_j^\dagger \otimes [a_k, \rho_B] \big)$$
$$+ f_k \big( [v_j B v_j^\dagger, \rho_S] \otimes a_k^\dagger \rho_B + \rho_S v_j B v_j^\dagger \otimes [a_k^\dagger, \rho_B] \big) \Big)$$
$$= 0, \tag{56}$$

where the second step uses that the Gibbs state $\rho_B$ commutes with $H_B$, and the last step follows from Eq. (52) and $\mathrm{tr}(H_S) = 0 = \mathrm{tr}(B)$.

A general treatment of dynamical decoupling for unbounded operators has been developed in Ref. [16] for the case of pure states. Here, it has been shown that the convergence of Trotter, together with the condition (52), suffices for dynamical decoupling to work. However, Ref. [16] does not explicitly cover the mixed state case; furthermore, it only considers the limiting evolution $N \to \infty$ and does not provide quantitative bounds to the convergence rate of dynamical decoupling for finite $N$. The solutions to both of these problems are presented in the next subsection.

## 5.2 Error bound for the dynamical decoupling of spin–boson models

This section combines the results and discussions from the previous sections and presents a generic scheme to quantify the efficiency of dynamical decoupling for the class of models introduced in Section 3.

We begin by stating two additional regularity requirements for spin–boson models other than Assumptions 3.1–3.2. They are needed to get an explicit bound on the decoupling error in Theorem 5.3.

**Assumption 5.1.** The modes $\omega = (\omega_k)_{k \in K}$ and the couplings $f = (f_k)_{k \in K}$ between the qubit and the field modes satisfy the following property:

$$\sum_{k \in K} \omega_k^2 |f_k|^2 < \infty. \tag{57}$$

Physically, Assumption 5.1 might require the existence of a UV-cutoff in the noise spectrum (e.g. for Lorentzian resonance, where dynamical decoupling is expected to work). However, it is mathematically unclear whether it is possible to relax this assumption, while still obtaining the same $1/N$ error scaling in dynamical decoupling. We expect that combining our method with a result similar to [31, Theorem 3] might enable the computation of decoupling error bounds under a weaker condition than Assumption 5.1 at the cost of a slower convergence speed. We also comment on this in the concluding remarks (see Section 7, point (iv) of the avenues for generalization).

**Assumption 5.2.** For $p \in \{1, 2\}$ and all $\beta > 0$, the following estimate holds for the modes $\omega = (\omega_k)_{k \in K}$:

$$\sum_{k \in K} \omega_k^p e^{-2\beta \omega_k} < \infty. \tag{58}$$

For a full list of all assumptions made in this paper and their purpose, we refer to Table 1. If all these assumptions are satisfied, we can apply our Trotter bound from Corollary 4.3 to

| | Assumption | Purpose |
|---|---|---|
| 3.1(i) | $m = \inf_{k \in K} \omega_k > -\infty$ | Self-adjointness of $H_\mathrm{B}$ (17) |
| 3.1(ii) | $\sum_{k \in K} \mathrm{e}^{-\beta \omega_k} < \infty$ | Well-definedness of the Gibbs state (22) |
| 3.2 | $\sum_{k \in K} |f_k|^2 < \infty$ | Self-adjointness of $H$ (23) |
| 5.1 | $\sum_{k \in K} \omega_k^2 |f_k|^2 < \infty$ | $1/N$ error scaling for dynamical decoupling |
| 5.2 | $\sum_{k \in K} \omega_k^p \mathrm{e}^{-2\beta \omega_k} < \infty$ for $p \in \{1, 2\}$ | Partition function $Z(2\beta)$ twice differentiable in $(2\beta)$ |

Table 1: Assumptions made in this paper and their purpose. In particular, these assumptions are made in Theorem 5.3. This table aims to help navigate the paper and clarify the reason for each assumption. We remark that all these assumptions are trivially satisfied whenever the boson field has finitely many modes. Also notice that Assumption 3.1(i) is stated in a slightly different form here and in the main text, where we assume $m > 0$. Both assumptions are equivalent after a shift in the zero-point energy $\omega_0$, which can always be done w.l.o.g. To simplify the calculations, we assumed $m > 0$ in the derivation of Theorem 5.3 but an analogous result for the case $m > -\infty$ is directly obtained by replacing $m \to m - \omega_0$ and each $\omega_k \to \omega_k - \omega_0$.

the setting of dynamical decoupling. This allows us to finally state our main result about the efficiency of dynamical decoupling.

**Theorem 5.3.** *Consider a spin–boson Hamiltonian $H$ on $\mathcal{H} = \mathbb{C}^2 \otimes \mathcal{F}$ as in Eq. (23) satisfying Assumptions 3.1 and 3.2. Let the initial state be of the form $\rho_\mathrm{S} \otimes \rho_\mathrm{B}$, where $\rho_\mathrm{S}$ is an arbitrary density matrix on $\mathbb{C}^2$ and $\rho_\mathrm{B} = \mathrm{e}^{-\beta H_\mathrm{B}}/Z(\beta)$ is the Gibbs state of the boson field at inverse temperature $\beta \in \mathbb{R}_+$. Here, $Z(\beta) = \mathrm{tr}\, \mathrm{e}^{-\beta H_\mathrm{B}}$ is the grand canonical partition function. Then, for a decoupling set $\mathcal{V} = \{v_0, \dots, v_{L-1}\}$ satisfying the decoupling condition (52), the following properties hold:*

- *dynamical decoupling works for $H$;*

- *if, in addition, both assumptions 5.1 and 5.2 hold, then $Z(2\beta)$ is twice differentiable in $(2\beta)$, $\rho_\mathrm{S} \otimes \rho_\mathrm{B} \in \bigcap_{j=0}^{L-1} \mathrm{Dom}\, \mathbf{ad}^2_{U_j H U_j^\dagger} \cap \mathrm{Dom}\, \mathbf{ad}_{U_j H U_j^\dagger} \sum_{i=0}^{j-1} \mathbf{ad}_{U_i H U_i^\dagger}$, and the error of dynamical decoupling can be bounded through Corollary 4.3 by*

$$\xi_N(t; \rho) < \frac{t^2}{N} \frac{L+1}{2L} C, \tag{59}$$

*where $|\mathcal{V}| = L \in \mathbb{N}$ is the number of decoupling operations, $U_j = v_j \otimes I_\mathrm{B}$, and $C \in \mathbb{R}_+$ is a constant that entirely depends on the Hamiltonian $H$. More precisely,*

$$
\begin{aligned}
C = \frac{4}{Z(\beta)} \max\{\|H_\mathrm{S}\|_\mathrm{HS}^2, \|B\|_\mathrm{HS}^2\} \times \Bigg( & Z(2\beta) \\
& + 4\big(\|f\|^2 + \|\omega f\|^2\big) \left[ -\frac{1}{m} \frac{\mathrm{d}}{\mathrm{d}(2\beta)} + 1 \right] Z(2\beta) \\
& + 58\|f\|^4 \left[ \frac{1}{m^2} \frac{\mathrm{d}^2}{\mathrm{d}(2\beta)^2} - \frac{3}{m} \frac{\mathrm{d}}{\mathrm{d}(2\beta)} + 2 \right] Z(2\beta) \Bigg)^{\frac{1}{2}},
\end{aligned}
\tag{60}
$$

*where again $m = \inf_j \omega_j$.*

*Proof.* The first statement follows directly from Thm. 4.1 noticing that $\bigcap_{j=0}^{L-1} \mathrm{Dom}\, \mathbf{ad}_{H_j} = \mathrm{Dom}\, \mathbf{ad}_{I_S \otimes H_B}$. The second statement is proven in the Appendix. In particular, Proposition B.1 shows that $Z(2\beta)$ is twice differentiable in $(2\beta)$ under our assumptions. Then, Corollary C.6 proves the bound presented here. It is a consequence of the Trotter bound in Liouville space (Corollary 4.3) applied to the context of dynamical decoupling. After applying a triangle inequality to Corollary 4.3, one obtains the following error bound for dynamical decoupling

$$\xi_N(t; \rho_S \otimes \rho_B) \leq \frac{1}{L^2} \sum_{k=0}^{L-1} \left( \frac{1}{2} \left\| \mathbf{ad}^2_{U_k H U_k^\dagger} \rho_S \otimes \rho_B \right\|_{\mathrm{HS}} + \sum_{i=0}^{k-1} \left\| \mathbf{ad}_{U_k H U_k^\dagger} \mathbf{ad}_{U_i H U_i^\dagger} \rho_S \otimes \rho_B \right\|_{\mathrm{HS}} \right). \quad (61)$$

Then, the bound follows from Theorem C.5, which summarizes the explicit computations of all Hilbert–Schmidt norms appearing in Eq. (61). These computations are performed in Lemmas C.2–C.4. The constant $C$ is then the maximum over all these norms. The pre-factor $(L+1)/(2L)$ comes from noticing that the sum over $k$ in Eq. (61) has $L(L+1)/2$ terms (the $L$-th triangular number). $\qquad \square$

The quantity $C$ in Eq. (60), in fact, constitutes a loose version of a tighter (but with a much more cumbersome expression) bound presented in the Appendix, see Theorem C.5. This tighter bound, unlike the one presented here, carries an explicit dependence on the density matrix of the system $\rho_S$ as well as the specific unitary matrices $v_j$ employed in the decoupling process.

Remarkably, in both versions (loose and tight) of our bound, the dependence of the error on the specifics of the boson field is entirely encoded in the grand canonical partition function $Z(\beta)$ of the boson field and its first two derivatives. All intricate domain conditions that would have to be checked in the general case, Corollary 4.3, are automatically taken care of in this case, as the Gibbs state regularizes the bath infinities. Therefore, this bound constitutes a ready-to-use recipe for computing dynamical decoupling efficiencies for spin–boson models— once the partition function of the boson field is known, the bound can be computed.

We conclude this section with some physical remarks. At first glance, it might seem counter-intuitive that the decoupling error increases when the lowest bath frequency $m$ is small: one would expect low-frequency noise to be easy to decouple. However, this effect actually makes sense from a physical perspective, if we recall that we consider the bath to be in a thermal state $\rho_B$. For small $m$, the "cost" of creating a bath excitation is low: thus, for a fixed bath temperature $T$, the occupancy number of the Gibbs state increases when decreasing $m$. In turn, $\rho_B$ becomes less tracial (less regularizing). Instead, if one starts in the ground state of the bath, the occupancy number is fixed and one would indeed expect a smaller decoupling error when $m$ is small. The same low-error behaviour with low $m$ is also expected for finite-dimensional baths, in which the noise frequency completely determines the required decoupling speed. This shows an important qualitative feature of our bound that—to the best of our knowledge—has not been observed before: On the one hand, if our qubit is coupled to other two-level bath systems, low-frequency noise is favourable for dynamical decoupling. On the other hand, if our qubit is coupled to a bath oscillator in a thermal state, low-frequency noise becomes particularly hard to decouple.

# 6   Examples

In this section, we apply the bound to some relevant examples and compare it with numerical simulations. In the case of the qubit coupled to a single bosonic mode, we are able to obtain even tighter estimates than the one given in Theorem 5.3. These are explicitly given in Appendix C.2 and are used for the single–mode examples in the following.

As we will see in this section, our bounds correctly characterize the error scaling in various system parameters, most importantly in the number of decoupling cycles $N$ and the inverse temperature $\beta$. Nevertheless, a comparison with numerical simulations shows that they are quite loose. This is not surprising as their calculation relies upon several loose estimates so there might be room for improvement. Most importantly, we use several instances of the triangle inequality as well as some other loose estimates to simplify the technical derivations (in particular, see Equations (117)–(119) in Appendix C).

## 6.1   Jaynes–Cummings model

The Jaynes–Cummings model is recovered from the general spin–boson Hamiltonian (23) by setting the number of bath modes to $d = 1$, $H_{\mathrm{S}} = \frac{\omega_{\mathrm{S}}}{2}\sigma_z$ and $B = \sigma_-$. Furthermore, we will set $f \in \mathbb{R}$. Then, the Hamiltonian reads

$$H = \frac{\omega_{\mathrm{S}}}{2}\sigma_z + \omega_{\mathrm{B}} a^\dagger a + f\left(\sigma^+ a + \sigma^- a^\dagger\right). \tag{62}$$

This Hamiltonian describes a spin, which interacts with a monochromatic bosonic environment via a flip–flop interaction. Differently from the dephasing model analyzed in Section 2, here we have to use the full qubit decoupling set $\mathcal{V} = \{I, \sigma_x, \sigma_y, \sigma_z\}$, and thus rely on the novel results presented in the previous sections, to eliminate the system Hamiltonian and interaction components through dynamical decoupling. We find that the decoupling error can be upper bounded through Eq. (60) by

$$\xi_N(t; \rho) < \frac{5t^2}{16N}\max\left(2, |\omega_{\mathrm{S}}|^2\right)\kappa(\beta, \omega_{\mathrm{B}}, f), \tag{63}$$

where $\kappa$ is given in Eq. (14). A more refined bound is given in Appendix C.3, see Eqs. (177)–(178). In particular, we again have $\xi_N(t; \rho) = \mathcal{O}(1/N)$. This is confirmed numerically in Fig. 2(a). The refined bounds take into account the explicit dependency on the system input state on the error. For example, consider the qubit states $\rho_{\mathrm{S}} = |1\rangle\langle 1|$ and $\rho_{\mathrm{S}} = |+\rangle\langle +|$. If we take the zero-temperature limit ($\beta \to \infty$) of the refined bound, we find that the decoupling errors become

$$\xi_N(t; |1\rangle\langle 1| \otimes \rho_{\mathrm{B}}) \leq \frac{|f|t^2}{4N}\left((3 + 6\sqrt{2})|f| + 4(\omega_{\mathrm{S}} + \omega_{\mathrm{B}})\right) \tag{64}$$

and

$$\xi_N(t; |+\rangle\langle +| \otimes \rho_{\mathrm{B}}) \leq \frac{t^2}{8N}\left((14 + 9\sqrt{2} + 2\sqrt{3} + \sqrt{6})|f|^2 + 6\sqrt{2}|f|(\omega_{\mathrm{S}} + \omega_{\mathrm{B}}) + 2\sqrt{2}\omega_{\mathrm{S}}^2\right). \tag{65}$$

The dependency of the refined bounds on the inverse temperature $\beta$ is shown in Fig. 2(b). To the best of our knowledge, this is the first completely analytical treatment of the dynamical decoupling efficiency for such a flip–flop interaction.

(a)           (b)

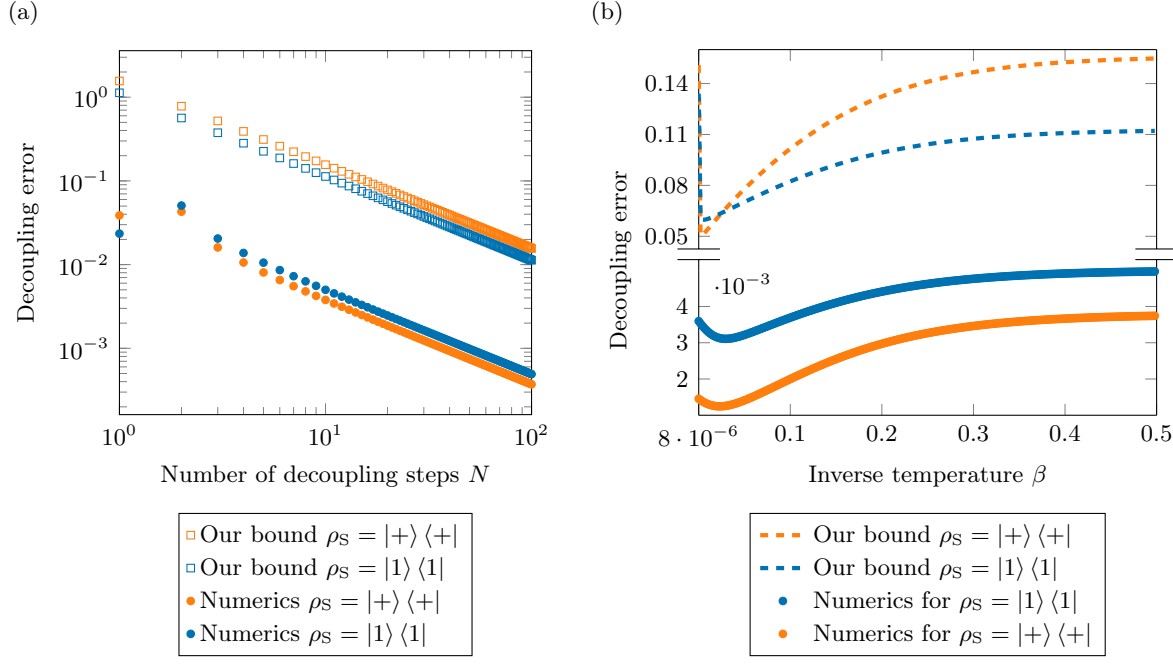

Figure 2: Dynamical decoupling error for a qubit coupled to monochromatic boson field via a flip–flop (Jaynes–Cummings) interaction. The Hamiltonian is given in Eq. (62) and dynamical decoupling is performed by repetitive pulse cycles through the Pauli group $\mathcal{V} = \{I, \sigma_x, \sigma_y, \sigma_z\}$. The initial state is $\rho_S \otimes \rho_B$, where $\rho_B$ is the Gibbs state of the bosonic bath with inverse temperature $\beta$ and $\rho_S$ is either $|+\rangle\langle+|$ (orange) or $|1\rangle\langle1|$ (blue). We fix the total evolution time to $t = 1$ and choose $\omega_S = 1$, $\omega_B = 10$ and $f = 0.1$. (a) Error as a function of the number $N$ of decoupling cycles for fixed $\beta = 1$. The empty squares show our analytical bound (Eq. (177) for $\rho_S = |0\rangle\langle0|$ and Eq. (178) for $\rho_S = |+\rangle\langle+|$) and the filled dots show a numerical simulation, where we truncated the bosonic field in Fock space at dimension $d = 10$. We see that the decoupling error decays as $1/N$. Our bound captures this asymptotic behaviour of the decoupling error. (b) Error as a function of the inverse temperature $\beta$ for fixed $N = 10$. The dashed lines are our bound (Eq. (177) for $\rho_S = |0\rangle\langle0|$ and Eq. (178) for $\rho_S = |+\rangle\langle+|$) and the dots are a numerical simulation, where we truncated the bosonic field in Fock space at dimension $d = 10$. We see that the decoupling error first decays with increasing $\beta$ and then increases as $\mathcal{O}(\sqrt{\beta})$. Finally, the error saturates at a constant due to finite $N$. The error decay for small $\beta$ is lightly visible in our bound (which scales as $\mathcal{O}(1/\sqrt{\beta})$ in this regime), and it correctly captures the $\mathcal{O}(\sqrt{\beta})$ scaling and eventual saturation for larger $\beta$.

## 6.2 Quantum Rabi model

The quantum Rabi model is a general model to describe the interaction between light and matter; the Jaynes–Cummings model presented before actually corresponds to the rotating-wave approximation of this model, as rigorously proven in Ref. [56]. We can obtain it as a special case of the class of spin–boson Hamiltonians (23) with the choice of $d = 1$ bath modes, $H_S = \frac{\omega_S}{2}\sigma_z$ and $B = \sigma_x$. In addition, we will fix $f \in \mathbb{R}$, so that the Hamiltonian becomes

$$H = \frac{\omega_S}{2}\sigma_z + \omega_B a^\dagger a + f\sigma_x(a + a^\dagger). \tag{66}$$

Again, the decoupling group will be $\mathscr{V} = \{I, \sigma_x, \sigma_y, \sigma_z\}$, and the decoupling error is bounded via Eq. (60) as

$$\xi_N(t; \rho) < \frac{5t^2}{16N}\max\left(4, |\omega_S|^2\right)\kappa(\beta, \omega_B, f), \tag{67}$$

where again $\kappa(\beta, \omega_B, f)$ is given in Eq. (14). A tighter bound can be found in Appendix C.3, see Eq. (180). Importantly, our bound proves that the decoupling error admits a $\mathcal{O}(1/N)$ scaling, which is confirmed numerically in Fig. 3(a). Furthermore, our (tighter) bound captures the dependency of the error on the inverse temperature $\beta$ genuinely, see Fig. 3(b) for a comparison to a numerical simulation, when the initial state of the system is $\rho_S = |0\rangle\langle0|$. In this case, the zero temperature limit ($\beta \to \infty$) of the refined error bound becomes

$$\xi_N(t; |0\rangle\langle0| \otimes \rho_B) \leq \frac{|f|t^2}{2N}\left(2(5 + 4\sqrt{2})|f| + (2 + \sqrt{2})(\omega_S + \omega_B)\right). \tag{68}$$

As in the case of the Jaynes–Cummings model (62), the decoupling dynamics for the quantum Rabi model (66) is not exactly solvable and we are not aware of any fully analytical treatment of its decoupling error.

## 6.3 Qubit coupled to infinitely many modes

The last example we consider is a qubit isotropically coupled to infinitely many bosonic modes. This model can be recovered from the general case (23) by setting $K = \mathbb{N}$ (thus with the dimension of the single-particle space being $d = \infty$) and $H_S = \frac{\omega_S}{2}\sigma_z$. Leaving the parameters $\omega_k$, $f_k$, and the operator $B$ arbitrary for the moment, the Hamiltonian reads:

$$H = \frac{\omega_S}{2}\sigma_z + \sum_{k=1}^{\infty}\omega_k a_k^\dagger a_k + \sum_{k=1}^{\infty}\left(f_k^* B^\dagger a_k + f_k B a_k^\dagger\right). \tag{69}$$

The decoupling set is again $\mathscr{V} = \{I, \sigma_x, \sigma_y, \sigma_z\}$. Under the assumptions of Theorem 5.3, we can bound the decoupling error for this model. On the one hand, this model involves an infinite number of field modes: therefore, numerical simulations become unfeasible and we must entirely rely on analytical estimates for the decoupling efficiency. On the other hand, this model describes noise more realistically than the previous models, since it incorporates a qubit coupling to arbitrary bath frequencies. This highlights the importance of our analytical procedure.

To compute the bounds, we recall that the partition function for $H$ (69) reads

$$Z(\beta) = e^{-\sum_{i=1}^{\infty}\ln[1-\exp(-\beta\omega_i)]}. \tag{70}$$

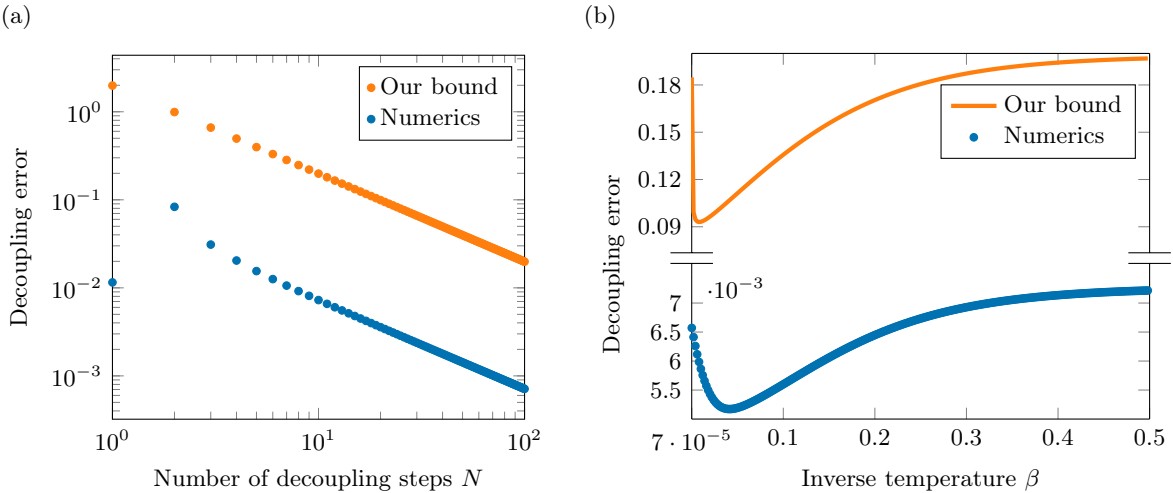

Figure 3: Dynamical decoupling error for a qubit coupled to monochromatic boson field via a quantum Rabi interaction. The Hamiltonian is given in Eq. (66) and dynamical decoupling is performed by repetitive pulse cycles through the Pauli group $\mathscr{V} = \{I, \sigma_x, \sigma_y, \sigma_z\}$. The initial state is $|0\rangle\langle 0|\otimes\rho_{\mathrm{B}}$, where $\rho_{\mathrm{B}}$ is the Gibbs state of the bosonic bath with inverse temperature $\beta$. We fix the total evolution time to $t = 1$ and choose $\omega_{\mathrm{S}} = 1$, $\omega_{\mathrm{B}} = 10$ and $f = 0.1$. The orange dots show our analytical bound (180) and the blue dots show a numerical simulation, where we truncated the bosonic field in Fock space at dimension $d = 10$. (a) Error as a function of the number $n$ of decoupling cycles for fixed $\beta = 1$. We see that the decoupling error decays as $1/N$. Our bound captures this asymptotic behaviour of the decoupling error. (b) Error as a function of the inverse temperature $\beta$ for fixed $N = 10$. We see that the decoupling error first decays with increasing $\beta$ and then increases as $\mathcal{O}(\sqrt{\beta})$. Finally, the error saturates at a constant due to finite $N$. The error decay for small $\beta$ is lightly visible in our bound (which scales as $\mathcal{O}(1/\sqrt{\beta})$ in this regime), and it correctly captures the $\mathcal{O}(\sqrt{\beta})$ scaling and eventual saturation for larger $\beta$.

Therefore, the first derivative reads

$$-\frac{\mathrm{d}}{\mathrm{d}(2\beta)} Z(2\beta) = \sum_{i=1}^{\infty} \frac{\mathrm{e}^{-2\beta\omega_i}\omega_i}{\left(1 - \mathrm{e}^{-2\beta\omega_i}\right)\prod_{j=1}^{\infty}\left(1 - \mathrm{e}^{-2\beta\omega_j}\right)} \qquad (71)$$

and the second derivative computes to

$$\frac{\mathrm{d}^2}{\mathrm{d}(2\beta)^2} Z(2\beta) = \sum_{i=1}^{\infty}\left(4\prod_{k=1}^{\infty}\left(1 - \mathrm{e}^{-2\beta\omega_k}\right)\right)^{-1}\left[\omega_i^2\mathrm{csch}^2\left(\beta\omega_i\right)\right.$$
$$\left.+ \omega_i\left(\coth\left(\beta\omega_i\right) - 1\right)\sum_{j=1}^{\infty}\omega_j\left(\coth\left(\beta\omega_j\right) - 1\right)\right]. \qquad (72)$$

As long as Assumption 5.2 is satisfied, the series above converge and can be computed for a given choice of $(\omega_k)_{k\in K}$. An explicit derivation of $\frac{\mathrm{d}^2}{\mathrm{d}(2\beta)^2}Z(2\beta)$ in the general case can be found in Appendix C.3, in particular, see Eq. (184). A closed-form expression for the bound on the dynamical decoupling error can then directly obtained by inserting Eqs. (70)–(72) into Theorem 5.3. When particular values for $\omega_{\mathrm{S}}$, $(f_k)_{k\in\mathbb{N}}$, and $(\omega_k)_{k\in\mathbb{N}}$, as well as the coupling operator $B$ are specified for the model at hand, this bound can be computed explicitly.

For concreteness, we shall consider a toy model corresponding to the following choices of parameters: $\omega_k = k$ (linearly increasing modes) and $f_k = f/k^2$ for $k = 1, 2, \ldots$, with $f \in \mathbb{R}$ being a coupling constant. Then, all of our assumptions are satisfied for all $\beta > 0$. Indeed, we have

$$m = \inf_{k\in\mathbb{N}} k = 1 > 0 \qquad (73)$$

$$\sum_{k=1}^{\infty}\mathrm{e}^{-\beta k} = \frac{1}{\mathrm{e}^{\beta} - 1} < \infty, \qquad (74)$$

so that Assumption 3.1 is satisfied. Furthermore,

$$\sum_{k=1}^{\infty}\frac{f^2}{k^4} = \frac{f^2\pi^4}{90} < \infty, \qquad (75)$$

thus Assumption 3.2 also holds. Lastly,

$$\sum_{k=1}^{\infty}k^2\frac{f^2}{k^4} = \sum_{k=1}^{\infty}\frac{f^2}{k^2} = \frac{f^2\pi^2}{6} < \infty; \qquad (76)$$

$$\sum_{k=1}^{\infty}k\mathrm{e}^{-2\beta k} = \frac{\mathrm{e}^{2\beta}}{\left(\mathrm{e}^{2\beta} - 1\right)^2} < \infty; \qquad (77)$$

$$\sum_{k=1}^{\infty}k^2\mathrm{e}^{-2\beta k} = \frac{\mathrm{e}^{2\beta} + \mathrm{e}^{4\beta}}{\left(\mathrm{e}^{2\beta} - 1\right)^3} < \infty, \qquad (78)$$

which also shows that Assumptions 5.1 and 5.2 are satisfied. Thus, Theorem 5.3 applies. For this choice of parameters, we can compute the partition function by evaluating the series

$$\sum_{k=1}^{\infty}\ln\left[1 - q^k\right] = \ln\left[(q; q)_{\infty}\right], \qquad (79)$$

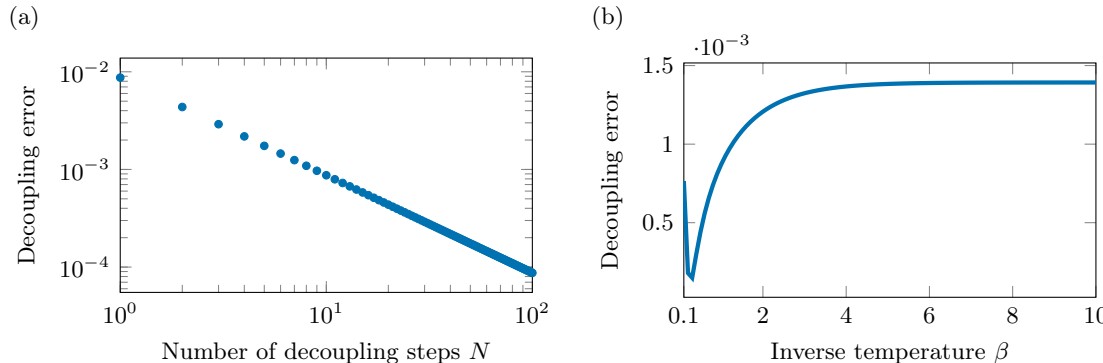

Figure 4: Dynamical decoupling error for a qubit coupled to an infinite number of bosonic modes via a Jaynes–Cummings interaction. The Hamiltonian is given in Eq. (69) for $B = \sigma_-$, $(\omega_k)_{k\in\mathbb{N}} = (k)_{k\in\mathbb{N}}$ and $(f_k)_{k\in\mathbb{N}} = \left(\frac{f}{k^2}\right)_{k\in\mathbb{N}}$ with $f = 0.1$. Dynamical decoupling is performed by repetitive pulse cycles through the Pauli group $\mathcal{V} = \{I, \sigma_x, \sigma_y, \sigma_z\}$. The initial state is $|0\rangle\langle 0| \otimes \rho_B$, where $\rho_B$ is the Gibbs state of the bosonic bath with inverse temperature $\beta$. We fix the total evolution time to $t = 1$ and choose $\omega_S = 1$. The blue dots show our analytical bound for the decoupling error. (a) Error as a function of the number of decoupling cycles $N$ for fixed $\beta = 1$ (This bound is obtained by inserting Eqs. (185)–(187) into Eq. (147)). We see that the decoupling error decays as $1/N$. (b) Error as a function of $\beta$ for fixed $N = 10$ (This bound is obtained by inserting Eqs. (185)–(187) into Eq. (147)). We see that the decoupling error increases as $\mathcal{O}(\sqrt{\beta})$ for small $\beta$ and eventually saturates at a constant due to finite $N$.

where $q = \mathrm{e}^{-\beta}$ and $(a; q)_\infty$ is the $q$-Pochhammer symbol [57, Chapter 2], which is easily evaluated numerically. Thus, the partition function reads

$$Z(\beta) = \frac{1}{(q; q)_\infty}. \tag{80}$$

The derivatives of $Z(2\beta)$ with respect to $(2\beta)$ are given in Appendix C.3. In particular, see Eqs. (185)–(187).

We now fix $B = \sigma_-$, i.e. a flip–flop interaction analogous to the one for the Jaynes–Cummings model, and compute our bound numerically. The results are shown in Fig. 4(a) for the error as a function of decoupling cycles $N$ and in Fig. 4(b) for the error as a function of the inverse temperature $\beta$. The decoupling error scales as $\mathcal{O}(N^{-1})$; furthermore, for fixed $N$, the error increases as $\mathcal{O}(\sqrt{\beta})$ for small $\beta$ and eventually saturates.

We stress again that, despite the simplicity of the toy model studied here, for an infinite number of modes numerical simulations of the true dynamics become unavailable so that one has to rely on error bounds. This highlights the importance of our analytical results.

# 7 Concluding remarks

We have presented a general framework to analytically compute quantitative bounds for the efficiency of dynamical decoupling in a vast class of models describing the interaction between a two-level system and a structured boson bath, the latter being in a thermal state with arbitrary temperature. Our results are gathered in Theorem 5.3 and can be interpreted as

a general recipe to test the convergence speed of dynamical decoupling for all models in the class, no matter the structure of the qubit–field coupling—be it longitudinal as in the Jaynes–Cummings and Rabi model, purely transversal as in the dephasing-type model described in the motivating example, or anything in the middle. Our results neither have restrictions on the structure of the boson modes (as long as there are at most countably many) nor the coupling constants between the qubit and each mode. This makes our results potentially adaptable to a vast range of experimental scenarios.

In all such cases, our analytic results show that (i) the Trotter error corresponding to a finite number $N$ of steps decreases as $\mathcal{O}(N^{-1})$, a result which (as pointed out in Refs. [31, 32]) is not obvious a priori because of the unbounded nature of our models; (ii) while our bounds are not sharp, they fully capture the asymptotic behavior of the decoupling error, both in relation with the number of steps as well as the temperature of the bath. Model-specific and/or state-specific bounds obtained through our approach might improve the matching of our results with the simulations while retaining the correct description of the asymptotic behavior.

Besides being applicable more broadly, our analytical approach offers several advantages over a numerical or perturbative treatment of dynamical decoupling via filter functions. The latter two methods rely on an arbitrarily chosen cut-off in the bath dimension that neither has any physical interpretation nor predictive power. Thus, the cut-off dimension has to be tuned *a posteriori* to match the experimental data. On the other hand, our analytical approach shows that the temperature of the bath takes the role of a *physical* cut-off parameter. In this sense, our analytical treatment is fully *ab initio* and only depends on real physical parameters. Nevertheless, doing analytics requires additional knowledge about the physical error model; in particular, about the bath resonance frequencies and coupling strengths between the system and the bath. Since these quantities are usually not known in real experiments, analytical decoupling bounds can serve as a first step towards a more rigorous analytical treatment of *bath spectroscopy* [58, 59, 60, 61, 62, 63] in the presence of bosonic baths, where the common qu*d*it assumption does not hold (for a review on bath spectroscopy for dephasing noise using dynamical decoupling, see Ref. [64]). This topic has attracted a lot of attention recently as it is crucial for the technological development of robust quantum devices [65, 66, 67, 68, 69].

From a physical perspective, our bounds revealed a surprising qualitative behavior of dynamical decoupling: One would expect that low-frequency noise is easier to decouple than high-frequency noise. That is because the system-bath interaction time scale is slower for low-frequency noise and therefore slower decoupling suffices to filter out the noise. While this intuition holds true for a system qubit coupled to two-level baths, it turns out to be the opposite for an oscillator bath in a thermal state. For the latter, a low minimal frequency implies a small energy barrier for the creation of a photon. Therefore, low-frequency noise leads to a high occupancy number of the thermal state, which makes it less regular. In turn, dynamical decoupling becomes harder to achieve. This behavior can be observed from our bounds, where it enters via the factors $1/m$ and $1/m^2$.

Furthermore, our results about spin–boson models crucially rely on a new Trotter convergence theorem for multiple Hamiltonians (Theorem 4.1), which extends a similar result already reported in Ref. [31] on the Trotterization between two Hamiltonians. Other than constituting the mathematical backbone of our results about spin–boson models, this theorem is *per se* of major interest for dynamical decoupling. Indeed, save from specific examples like the model described in the motivating example or spin–spin interactions [70], dynamical decoupling usually requires a decoupling set of cardinality larger than two. As such, other than being applicable to the spin–boson models considered in this paper, this theorem could foster

a plethora of new quantitative estimates for the convergence speed of dynamical decoupling in many other systems of theoretical and practical interest.

We conclude by commenting on five directions in which the results of the present paper could be further extended or improved: (i) obtaining estimates on the decoupling error in the trace norm, instead of the Hilbert–Schmidt norm, of the system components; (ii) replacing the qubit with a system of finite, but arbitrarily large, dimension (e.g. a qu*d*it or a family of multiple qubits); (iii) investigating the continuum limit for the boson bath; (iv) relaxing the assumption $\sum_k \omega_k^2 |f_k|^2 < \infty$ that was used to obtain our estimates; (v) extending the bounds to other decoupling schemes with a faster convergence rate in the number of decoupling steps.

About point (i): a decoupling bound that depends on trace-norm expectation values of the system quantities would be more natural than bounding the error globally in the Hilbert–Schmidt norm, as the former choice of norm would yield a bound depending on actual errors in physical measurement results. However, this approach would come along with many technical difficulties. Since the overall Hilbert space in our setting is infinite-dimensional, norms are no longer equivalent. Due to this, tools like Rastegin's inequality [71] become unavailable. Furthermore, it becomes unclear how to bound the Trotter error in terms of trace norms, as the space of operators with finite trace norm is only a Banach space and not a Hilbert space. Even though both spaces are structurally similar (see Ref. [55, Lemma 5.1]), proof techniques relying on the Hilbert space structure of the underlying state space, like the proof of Theorem 4.2, are not directly applicable. A third technical challenge would be the practical difficulty to explicitly compute trace norms if the operator of interest is not positive (like e.g. the bosonic annihilation and creation operators). This makes trace-norm estimates less favourable from a practical perspective than bounds in the Hilbert–Schmidt norm, which one can easily compute explicitly.

Point (ii), as already remarked in the text, does not exhibit particular technical challenges regarding the estimate procedure. In this case, the bosonic bath could involve different coupling strengths (form factors) to each qu*d*it-level or qubit. All calculations would remain the same otherwise. Point (iii) is more delicate, as the existence of the Gibbs state in such a case is compromised. Here, thermal states are implicitly defined via the KMS condition [72], and their existence and uniqueness must be carefully examined. Besides, the occupancy number basis, which plays a key role in computing all Hilbert–Schmidt norms involved in our bounds, ceases to be well-defined in the continuum limit.

About point (iv), we can distinguish two cases. If $\sum_k \omega_k^2 |f_k|^2 = \infty$ but still $\sum_k |f_k|^2 < \infty$ (normalizable form factor), then dynamical decoupling is still expected to work, but, as Theorem 5.3 does not apply, one could conjecture that a scenario analogous to the one studied in Refs. [31] is unveiled: decoupling works, but with a lower convergence speed in the number $N$ of steps, say, $\mathcal{O}(N^{-\delta})$ for some $0 \leq \delta < 1$. Such an expectation seems to be backed up by the recent results in Ref. [32] on the convergence of the Trotter product formula for two Hamiltonians and will be the object of future research. If, instead, $\sum_k |f_k|^2 = \infty$ (non-normalizable form factor), then, as discussed in Refs. [39, 40, 41], the self-adjointness domain of the models considered in this paper acquires a nontrivial dependence on the coupling itself, potentially invalidating the validity itself of dynamical decoupling. In fact, negative counterexamples were presented in Refs. [27, 28], where two spin–boson models with continuous bath and flat (thus, non-normalizable) form factors were shown to exactly satisfy the quantum regression theorem—thus necessarily disproving the possibility of achieving decoupling. Incidentally, comparing these negative results with the positive ones shown in the present paper clearly shows that the presence of ultraviolet divergences in models of matter–field interaction can

have practical, experimentally relevant consequences.

Finally, about point (v), there are several possible ways to improve the convergence speed of dynamical decoupling to $1/N^2$ or even better. An obvious route would be to employ higher-order Trotterization schemes. Obtaining bounds for this approach would be relatively straightforward by generalizing the higher-order Trotter bounds from Ref. [31] to multiple Hamiltonian components and applying them to dynamical decoupling following the recipe that is established in this paper. Another route would be to use non-equidistant pulses as in Uhrig dynamical decoupling [9, 73]. We expect that decoupling error bounds for such methods are obtainable by using the framework established in Ref. [31, Appendix C.1] and again following the recipe from this paper to convert pure state into mixed state error bounds.

## Acknowledgments

A.H. was partially supported by the Sydney Quantum Academy. D.L. acknowledges financial support by Friedrich-Alexander-Universität Erlangen-Nürnberg through the funding program "Emerging Talent Initiative" (ETI).

# A  The occupancy number basis

Let us consider a boson bath Hamiltonian $H_{\mathrm{B}} = \sum_{k \in K} \omega_k a_k^\dagger a_k$ acting on the Bose–Fock space $\mathcal{F}$ with single-particle space $\mathfrak{h}$, as defined in Section 3 of the main text. Recall that $\mathfrak{h} = \mathbb{C}^d$ and $K = \{1, \ldots, d\}$ for a boson field with a finite number $d$ of modes, while we have $\mathfrak{h} = \ell^2$ and $K = \mathbb{N}$ for countably many modes. We shall start our analysis by introducing the orthonormal basis that will be used in the computation of all trace and Hilbert–Schmidt norms involved in our bounds—the occupancy number basis.

To begin with, the single-particle Hilbert space $\mathfrak{h}$ admits a complete orthonormal set $(e_k)_{k \in K}$ of eigenvectors of $H_{\mathrm{B}}$, each being defined as the sequence

$$e_k := (\delta_{kh})_{h \in K} = \begin{cases} 1, & h = k \\ 0, & h \neq k, \end{cases} \tag{81}$$

whence immediately satisfying $H_{\mathrm{B}} e_k = \omega_k e_k$. This complete orthonormal set in $\mathfrak{h}$ serves as the primary building block of a complete orthonormal set of the whole Bose–Fock space $\mathcal{F}$, defined as follows. Given a finite subset $\{k_1, \ldots, k_N\}$ of the momentum set $K$, and $n_{k_1}, \ldots, n_{k_N} \in \mathbb{N}$ nonzero integers, define

$$|n_{k_1}, n_{k_2}, \ldots, n_{k_N}\rangle := S\left(e_{k_1}^{\otimes n_{k_1}} \otimes e_{k_2}^{\otimes n_{k_2}} \otimes \cdots e_{k_N}^{\otimes n_{k_N}}\right), \tag{82}$$

where $S$ is the symmetrization operator. The action of the annihilation and creation operators on these states can be directly computed: given $k_j \in \{k_1, \ldots, k_N\}$, one has

$$a_{k_j} |n_{k_1}, \ldots, n_{k_j}, \ldots, n_{k_N}\rangle = \sqrt{n_{k_j}} |n_{k_1}, \ldots, n_{k_j} - 1, \ldots, n_{k_N}\rangle; \tag{83}$$

$$a_{k_j}^\dagger |n_{k_1}, \ldots, n_{k_j}, \ldots, n_{k_N}\rangle = \sqrt{n_{k_j} + 1} |n_{k_1}, \ldots, n_{k_j} + 1, \ldots, n_{k_N}\rangle, \tag{84}$$

while, given $k \in K \setminus \{k_1, \ldots, k_N\}$,

$$a_k |n_{k_1}, \ldots, n_{k_N}\rangle = 0; \tag{85}$$

$$a_k^\dagger |n_{k_1}, \ldots, n_{k_N}\rangle = |n_{k_1}, \ldots, n_{k_j}, 1_k\rangle, \tag{86}$$

the latter notation signaling that the new state corresponds to a family of integers $n_{k_1}, \ldots, n_{k_N}, n_k$ with $n_k = 1$. As such, the operator $\mathcal{N}_k := a_k^\dagger a_k$ serves as the *number operator* corresponding to the $k$-th mode of the boson field, since, in all cases,

$$\mathcal{N}_k |n_{k_1}, \ldots, n_{k_j}, \ldots, n_{k_N}\rangle = n_k |n_{k_1}, \ldots, n_{k_j}, \ldots, n_{k_N}\rangle, \tag{87}$$

where of course $n_k = 0$ unless $k \in \{k_1, \ldots, k_N\}$.

Equivalently, and more conveniently, we can index all these states as $\{|\boldsymbol{n}\rangle\}_{\boldsymbol{n} \in \mathbb{N}_0^K}$, where $\mathbb{N}_0^K$ is the space of all sequences of integers $(n_k)_{k \in K}$ such that $n_k = 0$ for all but finitely many $k \in K$. Note that, even if $K = \mathbb{N}$, this is a countable set since it can be obtained as the countable union of countable sets (the set of all sequences with 1 nonzero integer, the set of all sequences with 2 nonzero integers, etc.). With this notation, for every $k \in K$ we have

$$a_k |\boldsymbol{n}\rangle = \sqrt{n_k} |n_1, n_2, \ldots, n_k - 1, \ldots\rangle; \tag{88}$$

$$a_k^\dagger |\boldsymbol{n}\rangle = \sqrt{n_k + 1} |n_1, n_2, \ldots, n_k + 1, \ldots\rangle; \tag{89}$$

$$\mathcal{N}_k |\boldsymbol{n}\rangle = n_k |\boldsymbol{n}\rangle, \tag{90}$$

with the convention that, whenever one of the entries is $-1$ (which happens at the right-hand side of Eq. (88) whenever $n_k = 0$), the vector equals zero. $\{|\boldsymbol{n}\rangle\}_{\boldsymbol{n}\in\mathbb{N}_0^K}$ is a complete orthonormal set of $\mathcal{F}$, which is usually referred to as the *occupancy number* basis. This set is particularly useful since it is invariant under the action of $a_k$ and $a_k^\dagger$. Furthermore, since $H_\mathrm{B} = \sum_{k\in K} \omega_k a_k^\dagger a_k = \sum_{k\in K} \omega_k \mathcal{N}_k$, Eq. (90) readily implies that any vector $|\boldsymbol{n}\rangle$ is also an eigenvector of $H_\mathrm{B}$,

$$H_\mathrm{B} |\boldsymbol{n}\rangle = \left( \sum_{k\in K} n_k \omega_k \right) |\boldsymbol{n}\rangle, \tag{91}$$

the sum being finite since $n_k = 0$ for all but finitely many values of $k$. As such, this is in fact a complete orthonormal set of eigenvectors of $H_\mathrm{B}$. They are also eigenvectors of the total number operator $\mathcal{N} = \sum_{k\in K} \mathcal{N}_k$, with

$$\mathcal{N} |\boldsymbol{n}\rangle = \left( \sum_{k\in K} n_k \right) |\boldsymbol{n}\rangle. \tag{92}$$

## B   Grand-canonical partition function and the Gibbs state

Now let us assume that the boson field described above satisfies Assumption 3.1. Let $\beta > 0$, and $\mu$ be a real parameter strictly smaller than $\min_{k\in K} \omega_k$, that is, $m := \inf_{k\in K}(\omega_k - \mu) > 0$; physically, $\beta$ is a thermodynamic quantity related to the temperature $T$ by $\beta = 1/(k_\mathrm{B}T)$, and $\mu$ is the chemical potential of the field. Notice that $H_\mathrm{B} - \mu\mathcal{N}$, defined on the domain of $\mathcal{N}$, is a nonnegative operator since

$$H_\mathrm{B} - \mu\mathcal{N} = \sum_{k\in K} (\omega_k - \mu) a_k^\dagger a_k, \tag{93}$$

and $\omega_k - \mu > 0$ for all $k \in K$. Correspondingly,

$$\rho_\mathrm{B} = \frac{\mathrm{e}^{-\beta(H_\mathrm{B}-\mu\mathcal{N})}}{\mathrm{tr}\,\mathrm{e}^{-\beta(H_\mathrm{B}-\mu\mathcal{N})}} \tag{94}$$

is the grand canonical Gibbs state of the field at temperature $T$ and chemical potential $\mu$ [72]. As is clear from Eq. (93), and already remarked in the main text, there is no loss of generality in assuming $\mu = 0$ (and thus $m = \inf_{k\in K} \omega_k$) since we can always redefine the field modes accordingly; as such, we will hereafter work under this simplifying choice. The Gibbs state is formally given by $\rho_\mathrm{B} := Z(\beta)^{-1}\mathrm{e}^{-\beta H_\mathrm{B}}$, where we introduced the grand canonical partition function,

$$Z(\beta) := \mathrm{tr}\,\mathrm{e}^{-\beta H_\mathrm{B}}. \tag{95}$$

At this stage, we still do not know whether $Z(\beta)$ is actually a finite quantity. However, it is a known fact (see e.g. [72, Proposition 5.2.27]) that Assumption 3.1(ii) for some $\beta$ is equivalent to the finiteness of $Z(\beta)$. For our purposes, it will be useful to recall the explicit calculation of $Z(\beta)$, together with some useful properties of it.

Before starting, let us recall the following definition. Given a family $(c_j)_{j\in\mathbb{N}}$ of positive real numbers, their infinite product is defined via

$$\prod_{j\in\mathbb{N}} c_j = \exp\left( \sum_{j\in\mathbb{N}} \log c_j \right), \tag{96}$$

and, as such, it converges if and only if the series $\sum_j \log c_j$ converges. Instead, when the series $\sum_j \log c_j$ diverges to infinity, we say that $\prod_j c_j$ diverges to zero.

**Proposition B.1.** *Let the boson field satisfy Assumption 3.1. Then, for all $\beta > 0$, $Z(\beta)$ is finite and given by*

$$Z(\beta) = \prod_{k \in K} \frac{1}{1 - \mathrm{e}^{-\beta\omega_k}}. \tag{97}$$

*If, in addition, Assumption 5.2 is satisfied, the function $\mathbb{R}_+ \ni \beta \to Z(\beta)$ is twice differentiable.*

*Proof.* We consider $K = \mathbb{N}$, i.e. countably many modes; the finite case is easily recovered as a particular case. Let us start by showing that the function

$$\mathbb{R}_+ \ni \beta \mapsto \prod_{k \in \mathbb{N}} \frac{1}{1 - \mathrm{e}^{-\beta\omega_k}} = \exp\left( \sum_{k \in \mathbb{N}} \log \frac{1}{1 - \mathrm{e}^{-\beta\omega_k}} \right) \tag{98}$$

is well-defined and twice differentiable. For the first one, we need to show that the series

$$z(\beta) := \sum_{k \in \mathbb{N}} \log \frac{1}{1 - \mathrm{e}^{-\beta\omega_k}} \tag{99}$$

converges. For this purpose, it suffices to note that, by Assumption 3.1(ii), the series $\sum_{k \in K} \mathrm{e}^{-\beta\omega_k}$ converges, which necessarily implies $\lim_{k \to \infty} \mathrm{e}^{-\beta\omega_k} = 0$. But then

$$\log \frac{1}{1 - \mathrm{e}^{-\beta\omega_k}} \sim \mathrm{e}^{-\beta\omega_k} \qquad (k \to \infty), \tag{100}$$

whence the series above, and thus the corresponding infinite product, converges by the limit comparison test. Thus $\beta \mapsto z(\beta)$ is well-defined and so is its exponential. To show it has the desired regularity, we notice that, by Eq. (100), for every $p \in \mathbb{N}$ we also have

$$\frac{\mathrm{d}^p}{\mathrm{d}\beta^p} \log \frac{1}{1 - \mathrm{e}^{-\beta\omega_k}} \sim \frac{\mathrm{d}^p}{\mathrm{d}\beta^p} \mathrm{e}^{-\beta\omega_k} = (-1)^p \, \omega_k^p \mathrm{e}^{-\beta\omega_k} \qquad (k \to \infty). \tag{101}$$

By our assumptions, $\sum_k \omega_k^p \mathrm{e}^{-\beta\omega_k} < \infty$ for all $\beta \in \mathbb{R}$ and $p \in \{0, 1, 2\}$. Therefore, we can differentiate under the series sign:

$$\frac{\mathrm{d}^p}{\mathrm{d}\beta^p} z(\beta) = \sum_{k \in K} \frac{\mathrm{d}^p}{\mathrm{d}\beta^p} \log \frac{1}{1 - \mathrm{e}^{-\beta\omega_k}} < \infty, \tag{102}$$

where again we used the limit comparison test. Thus $\beta \mapsto z(\beta)$ is twice differentiable and so is $\exp(z(\beta))$.

As such, we only need to show that $\mathrm{tr}\, \mathrm{e}^{-\beta H_{\mathrm{B}}}$ is actually equal to the quantity in Eq. (96). To this end, we compute the trace on the occupancy number basis $\{|n\rangle\}_{n \in \mathbb{N}_0^K}$ defined in the

previous section. We have

$$
\begin{aligned}
\operatorname{tr} e^{-\beta H_{\mathrm{B}}} &= \sum_{\boldsymbol{n} \in \mathbb{N}_0^K} \langle \boldsymbol{n} | e^{-\beta H_{\mathrm{B}}} | \boldsymbol{n} \rangle \\
&= \sum_{\boldsymbol{n} \in \mathbb{N}_0^K} \exp\left(-\beta \sum_{k \in K} n_k \omega_k\right) \\
&= \sum_{\boldsymbol{n} \in \mathbb{N}_0^K} \prod_{k \in K} \left(e^{-\beta \omega_k}\right)^{n_k} \\
&= \prod_{k \in K} \sum_{n \in \mathbb{N}} \left(e^{-\beta \omega_k}\right)^n \\
&= \prod_{k \in K} \frac{1}{1 - e^{-\beta \omega_k}},
\end{aligned}
\tag{103}
$$

where in the last step we computed a geometric series and used the fact that $\omega_k > 0$ for all $k \in K$. This completes the proof. $\qquad\square$

We remark that, as a result, the grand canonical partition function of a boson field with modes $(\omega_k)_{k \in K}$ equals the product of the partition functions of single-mode boson fields with each mode being $\omega_1, \omega_2, \ldots$ Physically speaking, this could be regarded as a manifestation of the fact that $H_{\mathrm{B}}$ describes a family of *noninteracting* bosons.

## C Calculating the Hilbert–Schmidt norms

We will now prove Theorem C.5. That is, we will prove that, under our assumptions, the state

$$
\rho_{\mathrm{S}} \otimes \rho_{\mathrm{B}} \in \bigcap_{j=0}^{L-1} \operatorname{Dom} \mathbf{ad}_{U_j H U_j^\dagger}^2 \cap \operatorname{Dom} \mathbf{ad}_{U_j H U_j^\dagger} \sum_{i=0}^{j-1} \mathbf{ad}_{U_i H U_i^\dagger},
\tag{104}
$$

i.e. satisfies the domain condition to apply Coroallary 4.3. Here, $\rho_{\mathrm{S}}$ is an arbitrary density matrix of the system, $\rho_{\mathrm{B}}$ is the Gibbs state (94) (with $\mu = 0$), $H$ being the Hamiltonian for the spin–boson model, and the $U_j = v_j \otimes I_{\mathrm{B}}$ are unitaries that act on the finite-dimensional system component alone. Again, without loss of generality, we will put ourselves in the countable case $K = \mathbb{N}$ (and thus $\mathfrak{h} = \ell^2$, and $\boldsymbol{n} \in \mathbb{N}_0^{\mathbb{N}}$); the cases of finitely many modes follow identically. Our proof is done by explicitly calculating all Hilbert–Schmidt norms of the summands appearing in $\mathbf{ad}_{U_j H U_j^\dagger} \mathbf{ad}_{U_i H U_i^\dagger} \rho_{\mathrm{S}} \otimes \rho_{\mathrm{B}}$ and showing that they are finite for all pairs $i, j$ ($j = 0, \ldots, L-1$; $i = 0, \ldots, j-1$). These computations guarantee the domain condition (104) and immediately lead to a bound of the error for dynamical decoupling via Corollary 4.3. A loose form of this bound is presented in the main text in Theorem 5.3.

We shall start by introducing the following compact notation. For any $f \in \ell^2$, we set

$$
a(f) = \sum_{k \in \mathbb{N}} f_k^* a_k, \qquad a^\dagger(f) = \sum_{k \in \mathbb{N}} f_k a_k^\dagger.
\tag{105}
$$

With this notation, the model in Eq. (23) reads, more compactly,

$$
H = H_{\mathrm{S}} + H_{\mathrm{B}} + B \, a^\dagger(f) + B^\dagger a(f),
\tag{106}
$$

where again tensor products are understood.

## C.1 The general case

Throughout this section, given $\boldsymbol{n} \in \mathbb{N}_0^{\mathbb{N}}$, we will use the notation $|\boldsymbol{n}(n_k \to n_k \pm 1)\rangle$ for $|n_1, \ldots, n_k \pm 1, \ldots, \rangle$.

**Lemma C.1.** *Let the boson field satisfy Assumption 3.1. Furthermore, let $\boldsymbol{n} \in \mathbb{N}_0^{\mathbb{N}}$, and $f \in \ell^2$ (Assumption 3.2). Then the following properties hold:*

(i) $[a(f), a^\dagger(f)] |\boldsymbol{n}\rangle = \|f\|^2 |\boldsymbol{n}\rangle$;

(ii) $\left[a(f), \mathrm{e}^{-\beta H_{\mathrm{B}}}\right] |\boldsymbol{n}\rangle = \sum_{k\in\mathbb{N}} f_k^* \sqrt{n_k} \mathrm{e}^{-\beta \sum_i \omega_i n_i} \left(1 - \mathrm{e}^{+\beta\omega_k}\right) |\boldsymbol{n}(n_k \to n_k - 1)\rangle$;

(iii) $\left[a^\dagger(f), \mathrm{e}^{-\beta H_{\mathrm{B}}}\right] |\boldsymbol{n}\rangle = \sum_{k\in\mathbb{N}} f_k \sqrt{n_k + 1} \mathrm{e}^{-\beta \sum_i \omega_i n_i} \left(1 - \mathrm{e}^{-\beta\omega_k}\right) |\boldsymbol{n}(n_k \to n_k + 1)\rangle$.

*In addition, if $\omega f \in \ell^2$ (Assumption 5.1), then*

(iv) $[a(f), H_{\mathrm{B}}] |\boldsymbol{n}\rangle = a(\omega f) |\boldsymbol{n}\rangle$;

(v) $[a^\dagger(f), H_{\mathrm{B}}] |\boldsymbol{n}\rangle = -a^\dagger(\omega f) |\boldsymbol{n}\rangle$.

*Proof.* All calculations that follow are legitimate since, as previously pointed out, the set $\{|\boldsymbol{n}\rangle\}_{\boldsymbol{n}\in\mathbb{N}_0^{\mathbb{N}}}$ is mapped into itself by any polynomial in $a_k$ and $a_k^\dagger$ (and thus by $H_{\mathrm{B}}$ as well), whence no domain issues arise. We start by noticing the following relations:

$$
\begin{aligned}
[a_k, H_{\mathrm{B}}] |\boldsymbol{n}\rangle &= \sum_{k'\in\mathbb{N}} \omega_{k'} [a_k, a_{k'}^\dagger a_{k'}] |\boldsymbol{n}\rangle \\
&= \sum_{k'\in\mathbb{N}} \omega_{k'} \left([a_k, a_{k'}^\dagger] a_{k'} + a_{k'}^\dagger [a_k, a_{k'}]\right) |\boldsymbol{n}\rangle \\
&= \omega_k a_k |\boldsymbol{n}\rangle,
\end{aligned}
\tag{107}
$$

and similarly

$$
\begin{aligned}
[a_k^\dagger, H_{\mathrm{B}}] |\boldsymbol{n}\rangle &= \sum_{k'\in\mathbb{N}} \omega_{k'} [a_k^\dagger, a_{k'}^\dagger a_{k'}] |\boldsymbol{n}\rangle \\
&= \sum_{k'\in\mathbb{N}} \omega_{k'} \left([a_k^\dagger, a_{k'}^\dagger] a_{k'} + a_{k'}^\dagger [a_k^\dagger, a_{k'}]\right) |\boldsymbol{n}\rangle \\
&= -\omega_k a_k^\dagger |\boldsymbol{n}\rangle.
\end{aligned}
\tag{108}
$$

Then:

$$[a(f), a^\dagger(f)] |\boldsymbol{n}\rangle = \sum_{k,k'\in\mathbb{N}} f_k^* f_{k'} [a_k, a_{k'}^\dagger] |\boldsymbol{n}\rangle = \|f\|^2 |\boldsymbol{n}\rangle \, ; \tag{109}$$

$$[a(f), H_{\mathrm{B}}] |\boldsymbol{n}\rangle = \sum_{k\in\mathbb{N}} f_k^* [a_k, H_{\mathrm{B}}] |\boldsymbol{n}\rangle$$

$$= \sum_{k\in\mathbb{N}} \omega_k f_k a_k^\dagger |\boldsymbol{n}\rangle$$

$$= a(\omega f) |\boldsymbol{n}\rangle \, ; \tag{110}$$

$$[a^\dagger(f), H_{\mathrm{B}}] |\boldsymbol{n}\rangle = \sum_{k\in\mathbb{N}} f_k [a_k, H_{\mathrm{B}}] |\boldsymbol{n}\rangle$$

$$= -\sum_{k\in\mathbb{N}} \omega_k f_k a_k^\dagger |\boldsymbol{n}\rangle$$

$$= -a^\dagger(\omega f) |\boldsymbol{n}\rangle \, . \tag{111}$$

Furthermore, we have

$$\left[a^\dagger(f), \mathrm{e}^{-\beta H_{\mathrm{B}}}\right] |\boldsymbol{n}\rangle = \mathrm{e}^{-\beta \sum_i \omega_i n_i} \sum_{k\in\mathbb{N}} f_k a_k^\dagger |\boldsymbol{n}\rangle$$

$$- \sum_{k\in\mathbb{N}} f_k \sqrt{n_k+1} \prod_i \mathrm{e}^{-\beta \omega_i a_i^\dagger a_i} |n_1, \ldots, n_{k-1}, n_k+1, \ldots\rangle$$

$$= \mathrm{e}^{-\beta \sum_i \omega_i n_i} \sum_{k\in\mathbb{N}} f_k \sqrt{n_k+1} \, |\boldsymbol{n}(n_k \to n_k+1)\rangle$$

$$- \sum_{k\in\mathbb{N}} f_k \sqrt{n_k+1} \left(\prod_{i\neq k} \mathrm{e}^{-\beta \omega_i n_i}\right) \mathrm{e}^{-\beta \omega_k (n_k+1)} |\boldsymbol{n}(n_j \to n_k+1)\rangle$$

$$= \sum_{k\in\mathbb{N}} f_k \sqrt{n_k+1} \left(\mathrm{e}^{-\beta \sum_i \omega_i n_i} - \mathrm{e}^{-\beta\left(\omega_k(n_k+1)+\sum_{i\neq k}\omega_i n_i\right)}\right)$$

$$|\boldsymbol{n}(n_k \to n_k+1)\rangle$$

$$= \sum_{k\in\mathbb{N}} f_k \sqrt{n_k+1} \, \mathrm{e}^{-\beta \sum_i \omega_i n_i} \left(1 - \mathrm{e}^{-\beta \omega_k}\right) |\boldsymbol{n}(n_k \to n_k+1)\rangle \tag{112}$$

and

$$\left[a(f), \mathrm{e}^{-\beta H_{\mathrm{B}}}\right] |\boldsymbol{n}\rangle = \mathrm{e}^{-\beta \sum_i \omega_i n_i} \sum_{k \in \mathbb{N}} f_k^* a_k |\boldsymbol{n}\rangle$$

$$- \sum_{k \in \mathbb{N}} f_k^* \sqrt{n_k} \prod_i \mathrm{e}^{-\beta \omega_i a_i^\dagger a_i} |n_1, \dots, n_{k-1}, n_k - 1, \dots\rangle$$

$$= \mathrm{e}^{-\beta \sum_i \omega_i n_i} \sum_{k \in \mathbb{N}} f_k^* \sqrt{n_k} |\boldsymbol{n}(n_k \to n_k - 1)\rangle$$

$$- \sum_{k \in \mathbb{N}} f_k^* \sqrt{n_k} \left(\prod_{i \neq k} \mathrm{e}^{-\beta \omega_i n_i}\right) \mathrm{e}^{-\beta \omega_k (n_k - 1)} |\boldsymbol{n}(n_k \to n_k - 1)\rangle$$

$$= \sum_{k \in \mathbb{N}} f_k^* \sqrt{n_k} \left(\mathrm{e}^{-\beta \sum_i \omega_i n_i} - \mathrm{e}^{-\beta \left(\omega_k (n_k - 1) + \sum_{i \neq k} \omega_i n_i\right)}\right)$$

$$|\boldsymbol{n}(n_k \to n_k - 1)\rangle$$

$$= \sum_{k \in \mathbb{N}} f_k^* \sqrt{n_k} \mathrm{e}^{-\beta \sum_i \omega_i n_i} \left(1 - \mathrm{e}^{+\beta \omega_k}\right) |\boldsymbol{n}(n_k \to n_k - 1)\rangle , \qquad (113)$$

thus completing the proof. $\qquad \square$

Next, we compute $\mathbf{ad}_{U_j H U_j^\dagger} \mathbf{ad}_{U_i H U_i^\dagger}(\rho_S \otimes \rho_B)$, which we need for the explicit bounds on the error of dynamical decoupling (Recall that $U_j = v_j \otimes I$, where $v_j \in \mathcal{B}(\mathcal{H}_{\mathrm{S}})$ is unitary). In particular, by doing so, we will retrieve $\mathbf{ad}_H^2(\rho_{\mathrm{S}} \otimes \rho_{\mathrm{B}})$ by setting $U_i = U_j = I \otimes I$; notice, however, that for the condition $\rho_{\mathrm{S}} \otimes \rho_{\mathrm{B}} \in \mathrm{Dom}\, \mathbf{ad}_{U_j H U_j^\dagger} \mathbf{ad}_{U_i H U_i^\dagger}$ the $U_k$ do not play any role, as they only act on the finite-dimensional system component.

We start by formally computing $\mathbf{ad}_{U_j H U_j^\dagger}(\rho_{\mathrm{S}} \otimes \rho_{\mathrm{B}})$:

$$\mathbf{ad}_{U_j H U_j^\dagger}(\rho_{\mathrm{S}} \otimes \rho_{\mathrm{B}}) = \left[v_j H_{\mathrm{S}} v_j^\dagger, \rho_{\mathrm{S}}\right] \otimes \rho_{\mathrm{B}} + \left[v_j B v_j^\dagger, \rho_{\mathrm{S}}\right] \otimes a^\dagger(f) \rho_{\mathrm{B}} + \rho_{\mathrm{S}} v_j B v_j^\dagger \otimes \left[a^\dagger(f), \rho_{\mathrm{B}}\right]$$

$$+ \left[v_j B^\dagger v_j^\dagger, \rho_{\mathrm{S}}\right] \otimes a(f) \rho_{\mathrm{B}} + \rho_{\mathrm{S}} v_j B^\dagger v_j^\dagger \otimes [a(f), \rho_{\mathrm{B}}] , \qquad (114)$$

where we used the formal relation $[A \otimes B, C \otimes D] = [A, C] \otimes BD + CA \otimes [B, D]$ and the fact that the Gibbs state $\rho_{\mathrm{B}}$ commutes with $H_{\mathrm{B}}$. From this, we can formally compute

$$\mathbf{ad}_{U_j H U_j^\dagger} \mathbf{ad}_{U_i H U_i^\dagger}(\rho_S \otimes \rho_B) = \left[U_j H U_j^\dagger, \left[v_i H_{\mathrm{S}} v_i^\dagger, \rho_{\mathrm{S}}\right] \otimes \rho_{\mathrm{B}}\right]$$

$$+ \left[U_j H U_j^\dagger, \left[v_i B v_i^\dagger, \rho_{\mathrm{S}}\right] \otimes a^\dagger(f) \rho_{\mathrm{B}}\right]$$

$$+ \left[U_j H U_j^\dagger, \rho_{\mathrm{S}} v_i B v_i^\dagger \otimes \left[a^\dagger(f), \rho_{\mathrm{B}}\right]\right]$$

$$+ \left[U_j H U_j^\dagger, \left[v_i B^\dagger v_i^\dagger, \rho_{\mathrm{S}}\right] \otimes a(f) \rho_{\mathrm{B}}\right]$$

$$+ \left[U_j H U_j^\dagger, \rho_{\mathrm{S}} v_i B^\dagger v_i^\dagger \otimes [a(f), \rho_{\mathrm{B}}]\right]$$

$$= \left[v_j H_{\mathrm{S}} v_j^\dagger, \left[v_i H_{\mathrm{S}} v_i^\dagger, \rho_{\mathrm{S}}\right]\right] \otimes \rho_{\mathrm{B}}$$

$$+ \left[v_j B v_j^\dagger, \left[v_i H_{\mathrm{S}} v_i^\dagger, \rho_{\mathrm{S}}\right]\right] \otimes a^\dagger(f) \rho_{\mathrm{B}}$$

$$+ \left[v_i H_{\mathrm{S}} v_i^\dagger, \rho_{\mathrm{S}}\right] v_j B v_j^\dagger \otimes \left[a^\dagger(f), \rho_{\mathrm{B}}\right]$$

$$+ \left[ v_j B^\dagger v_j^\dagger, \left[ v_i H_{\mathrm{S}} v_i^\dagger, \rho_{\mathrm{S}} \right] \right] \otimes a(f) \rho_{\mathrm{B}}$$

$$+ \left[ v_i H_{\mathrm{S}} v_i^\dagger, \rho_{\mathrm{S}} \right] v_j B^\dagger v_j^\dagger \otimes [a(f), \rho_{\mathrm{B}}]$$

$$+ \left[ v_j H_{\mathrm{S}} v_j^\dagger, \left[ v_i B v_i^\dagger, \rho_{\mathrm{S}} \right] \right] \otimes a^\dagger(f) \rho_{\mathrm{B}}$$

$$+ \left[ v_i B v_i^\dagger, \rho_{\mathrm{S}} \right] \otimes \left[ H_{\mathrm{B}}, a^\dagger(f) \rho_{\mathrm{B}} \right]$$

$$+ \left[ v_j B v_j^\dagger, \left[ v_i B v_i^\dagger, \rho_{\mathrm{S}} \right] \right] \otimes a^\dagger(f) a^\dagger(f) \rho_{\mathrm{B}}$$

$$+ \left[ v_i B v_i^\dagger, \rho_{\mathrm{S}} \right] v_j B v_j^\dagger \otimes \left[ a^\dagger(f), a^\dagger(f) \rho_{\mathrm{B}} \right]$$

$$+ \left[ v_j B^\dagger v_j^\dagger, \left[ v_i B v_i^\dagger, \rho_{\mathrm{S}} \right] \right] \otimes a(f) a^\dagger(f) \rho_{\mathrm{B}}$$

$$+ \left[ v_i B v_i^\dagger, \rho_{\mathrm{S}} \right] v_j B^\dagger v_j^\dagger \otimes \left[ a(f), a^\dagger(f) \rho_{\mathrm{B}} \right]$$

$$+ \left[ v_j H_{\mathrm{S}} v_j^\dagger, \rho_{\mathrm{S}} v_i B v_i^\dagger \right] \otimes \left[ a^\dagger(f), \rho_{\mathrm{B}} \right]$$

$$+ \rho_{\mathrm{S}} v_i B v_i^\dagger \otimes \left[ H_{\mathrm{B}}, \left[ a^\dagger(f), \rho_{\mathrm{B}} \right] \right]$$

$$+ \left[ v_j B v_j^\dagger, \rho_{\mathrm{S}} v_i B v_i^\dagger \right] \otimes a^\dagger(f) \left[ a^\dagger(f), \rho_{\mathrm{B}} \right]$$

$$+ \rho_{\mathrm{S}} v_i B v_i^\dagger v_j B v_j^\dagger \otimes \left[ a^\dagger(f), \left[ a^\dagger(f), \rho_{\mathrm{B}} \right] \right]$$

$$+ \left[ v_j B^\dagger v_j^\dagger, \rho_{\mathrm{S}} v_i B v_i^\dagger \right] \otimes a(f) \left[ a^\dagger(f), \rho_{\mathrm{B}} \right]$$

$$+ \rho_{\mathrm{S}} v_i B v_i^\dagger v_j B^\dagger v_j^\dagger \otimes \left[ a(f), \left[ a^\dagger(f), \rho_{\mathrm{B}} \right] \right]$$

$$+ \left[ v_j H_{\mathrm{S}} v_j^\dagger, \left[ v_i B^\dagger v_i^\dagger, \rho_{\mathrm{S}} \right] \right] \otimes a(f) \rho_{\mathrm{B}}$$

$$+ \left[ v_i B^\dagger v_i^\dagger, \rho_{\mathrm{S}} \right] \otimes [H_{\mathrm{B}}, a(f) \rho_{\mathrm{B}}]$$

$$+ \left[ v_j B v_j^\dagger, \left[ v_i B^\dagger v_i^\dagger, \rho_{\mathrm{S}} \right] \right] \otimes a^\dagger(f) a(f) \rho_{\mathrm{B}}$$

$$+ \left[ v_i B^\dagger v_i^\dagger, \rho_{\mathrm{S}} \right] v_j B v_j^\dagger \otimes \left[ a^\dagger(f), a(f) \rho_{\mathrm{B}} \right]$$

$$+ \left[ v_j B^\dagger v_j^\dagger, \left[ v_i B^\dagger v_i^\dagger, \rho_{\mathrm{S}} \right] \right] \otimes a(f) a(f) \rho_{\mathrm{B}}$$

$$+ \left[ v_i B^\dagger v_i^\dagger, \rho_{\mathrm{S}} \right] v_j B^\dagger v_j^\dagger \otimes [a(f), a(f) \rho_{\mathrm{B}}]$$

$$+ \left[ v_j H_{\mathrm{S}} v_j^\dagger, \rho_{\mathrm{S}} v_i B^\dagger v_i^\dagger \right] \otimes [a(f), \rho_{\mathrm{B}}]$$

$$+ \rho_{\mathrm{S}} v_i B^\dagger v_i^\dagger \otimes [H_{\mathrm{B}}, [a(f), \rho_{\mathrm{B}}]]$$

$$+ \left[ v_j B v_j^\dagger, \rho_{\mathrm{S}} v_i B^\dagger v_i^\dagger \right] \otimes a^\dagger(g) [a(f), \rho_{\mathrm{B}}]$$

$$+ \rho_{\mathrm{S}} v_i B^\dagger v_i^\dagger v_j B v_j^\dagger \otimes \left[ a^\dagger(f), [a(f), \rho_{\mathrm{B}}] \right]$$

$$+ \left[ v_j B^\dagger v_j^\dagger, \rho_{\mathrm{S}} v_i B^\dagger v_i^\dagger \right] \otimes a(f) [a(f), \rho_{\mathrm{B}}]$$

$$+ \rho_{\mathrm{S}} v_i B^\dagger v_i^\dagger v_j B^\dagger v_j^\dagger \otimes [a(f), [a(f), \rho_{\mathrm{B}}]]$$

$$= \left[ v_j H_{\mathrm{S}} v_j^\dagger, \left[ v_i H_{\mathrm{S}} v_i^\dagger, \rho_{\mathrm{S}} \right] \right] \otimes \rho_{\mathrm{B}}$$

$$+ \left( \left[ v_j B v_j^\dagger, \left[ v_i H_S v_i^\dagger, \rho_S \right] \right] + \left[ v_j H_S v_j^\dagger, \left[ v_i B v_i^\dagger, \rho_S \right] \right] \right) \otimes a^\dagger(f) \rho_B$$

$$+ \left( \left[ v_j B^\dagger v_j^\dagger, \left[ v_i H_S v_i^\dagger, \rho_S \right] \right] + \left[ v_j H_S v_j^\dagger, \left[ v_i B^\dagger v_i^\dagger, \rho_S \right] \right] \right) \otimes a(f) \rho_B$$

$$+ \left( \left[ v_i H_S v_i^\dagger, \rho_S \right] v_j B v_j^\dagger + \left[ v_j H_S v_j^\dagger, \rho_S v_i B v_i^\dagger \right] \right) \otimes \left[ a^\dagger(f), \rho_B \right]$$

$$+ \left( \left[ v_i H_S v_i^\dagger, \rho_S \right] v_j B^\dagger v_j^\dagger + \left[ v_j H_S v_j^\dagger, \rho_S v_i B^\dagger v_i^\dagger \right] \right) \otimes [a(f), \rho_B]$$

$$+ \left[ v_j B v_j^\dagger, \left[ v_i B v_i^\dagger, \rho_S \right] \right] \otimes a^\dagger(f) a^\dagger(f) \rho_B$$

$$+ \left[ v_j B^\dagger v_j^\dagger, \left[ v_i B^\dagger v_i^\dagger, \rho_S \right] \right] \otimes a(f) a(f) \rho_B$$

$$+ \left[ v_j B^\dagger v_j^\dagger, \left[ v_i B v_i^\dagger, \rho_S \right] \right] \otimes a(f) a^\dagger(f) \rho_B$$

$$+ \left[ v_j B v_j^\dagger, \left[ v_i B^\dagger v_i^\dagger, \rho_S \right] \right] \otimes a^\dagger(f) a(f) \rho_B$$

$$+ \left[ v_i B v_i^\dagger, \rho_S \right] v_j B v_j^\dagger \otimes \left[ a^\dagger(f), a^\dagger(f) \rho_B \right]$$

$$+ \left[ v_i B^\dagger v_i^\dagger, \rho_S \right] v_j B^\dagger v_j^\dagger \otimes [a(f), a(f) \rho_B]$$

$$+ \left[ v_i B v_i^\dagger, \rho_S \right] v_j B^\dagger v_j^\dagger \otimes \left[ a(f), a^\dagger(f) \rho_B \right]$$

$$+ \left[ v_i B^\dagger v_i^\dagger, \rho_S \right] v_j B v_j^\dagger \otimes \left[ a^\dagger(f), a(f) \rho_B \right]$$

$$+ \left[ v_j B v_j^\dagger, \rho_S v_i B v_i^\dagger \right] \otimes a^\dagger(f) \left[ a^\dagger(f), \rho_B \right]$$

$$+ \left[ v_j B^\dagger v_j^\dagger, \rho_S v_i B^\dagger v_i^\dagger \right] \otimes a(f) [a(f), \rho_B]$$

$$+ \left[ v_j B^\dagger v_j^\dagger, \rho_S v_i B v_i^\dagger \right] \otimes a(f) \left[ a^\dagger(f), \rho_B \right]$$

$$+ \left[ v_j B v_j^\dagger, \rho_S v_i B^\dagger v_i^\dagger \right] \otimes a^\dagger(f) [a(f), \rho_B]$$

$$+ \rho_S v_i B v_i^\dagger v_j B v_j^\dagger \otimes \left[ a^\dagger(f), \left[ a^\dagger(f), \rho_B \right] \right]$$

$$+ \rho_S v_i B^\dagger v_i^\dagger v_j B^\dagger v_j^\dagger \otimes [a(f), [a(f), \rho_B]]$$

$$+ \rho_S v_i B v_i^\dagger v_j B^\dagger v_j^\dagger \otimes \left[ a(f), \left[ a^\dagger(f), \rho_B \right] \right]$$

$$+ \rho_S v_i B^\dagger v_i^\dagger v_j B v_j^\dagger \otimes \left[ a^\dagger(f), [a(f), \rho_B] \right]$$

$$+ \left[ v_i B v_i^\dagger, \rho_S \right] \otimes \left[ H_B, a^\dagger(f) \rho_B \right]$$

$$+ \left[ v_i B^\dagger v_i^\dagger, \rho_S \right] \otimes [H_B, a(f) \rho_B]$$

$$+ \rho_S v_i B v_i^\dagger \otimes \left[ H_B, \left[ a^\dagger(f), \rho_B \right] \right]$$

$$+ \rho_S v_i B^\dagger v_i^\dagger \otimes [H_B, [a(f), \rho_B]] . \tag{115}$$

Our goal is to show that $\| \mathbf{ad}_{U_j H U_j^\dagger} \mathbf{ad}_{U_i H U_i^\dagger} (\rho_S \otimes \rho_B) \|_{\mathrm{HS}}$ is finite. By taking the Hilbert–Schmidt norm of the above expression and using the triangle inequality as well as the identity $\| A \otimes B \|_{\mathrm{HS}} = \| A \|_{\mathrm{HS}} \| B \|_{\mathrm{HS}}$, we can reduce this task to ensuring that the Hilbert–Schmidt norm of every individual bath term stays finite. For this, we collect the values of certain norms in the following lemma.

**Lemma C.2.** *Let the boson field satisfy Assumption 3.1. In particular, define $m := \inf_j \omega_j > 0$. Let $f \in \ell^2$ (Assumption 3.2) and $\omega f \in \ell^2$ (Assumption 5.1). Furthermore, let Assumption 5.2 be fulfilled. Then the following relations hold:*

(i) $\left\| e^{-\beta H_B} \right\|_{\mathrm{HS}}^2 = Z(2\beta);$

(ii) $\left\| a(f) e^{-\beta H_B} \right\|_{\mathrm{HS}}^2 \leq -\frac{\|f\|^2}{m} \frac{\mathrm{d}}{\mathrm{d}(2\beta)} Z(2\beta);$

(iii) $\left\| a^\dagger(f) e^{-\beta H_B} \right\|_{\mathrm{HS}}^2 \leq \|f\|^2 \left[ -\frac{1}{m} \frac{\mathrm{d}}{\mathrm{d}(2\beta)} + 1 \right] Z(2\beta);$

(iv) $\left\| a(f)^2 e^{-\beta H_B} \right\|_{\mathrm{HS}}^2 < \frac{\|f\|^4}{m^2} \frac{\mathrm{d}^2}{\mathrm{d}(2\beta)^2} Z(2\beta);$

(v) $\left\| a^\dagger(f)^2 e^{-\beta H_B} \right\|_{\mathrm{HS}}^2 < \|f\|^4 \left[ \frac{1}{m^2} \frac{\mathrm{d}^2}{\mathrm{d}(2\beta)^2} - \frac{3}{m} \frac{\mathrm{d}}{\mathrm{d}(2\beta)} + 2 \right] Z(2\beta);$

(vi) $\left\| a^\dagger(f) a(f) e^{-\beta H_B} \right\|_{\mathrm{HS}}^2 \leq \|f\|^4 \left[ \frac{1}{m^2} \frac{\mathrm{d}^2}{\mathrm{d}(2\beta)^2} - \frac{1}{m} \frac{\mathrm{d}}{\mathrm{d}(2\beta)} \right] Z(2\beta)$

(vii) $\left\| a(f) a^\dagger(f) e^{-\beta H_B} \right\|_{\mathrm{HS}}^2 < \|f\|^4 \left[ \frac{1}{m^2} \frac{\mathrm{d}^2}{\mathrm{d}(2\beta)^2} - \frac{2}{m} \frac{\mathrm{d}}{\mathrm{d}(2\beta)} + 1 \right] Z(2\beta);$

(viii) $\left\| \left[ a^\dagger(f), e^{-\beta H_B} \right] \right\|_{\mathrm{HS}}^2 < \|f\|^2 \left[ -\frac{1}{m} \frac{\mathrm{d}}{\mathrm{d}(2\beta)} + 1 \right] Z(2\beta);$

(ix) $\left\| \left[ a(f), e^{-\beta H_B} \right] \right\|_{\mathrm{HS}}^2 < \|f\|^2 \left[ -\frac{1}{m} \frac{\mathrm{d}}{\mathrm{d}(2\beta)} + 1 \right] Z(2\beta);$

(x) $\left\| a^\dagger(f) \left[ a^\dagger(f), e^{-\beta H_B} \right] \right\|_{\mathrm{HS}}^2 < \|f\|^4 \left[ \frac{1}{m^2} \frac{\mathrm{d}^2}{\mathrm{d}(2\beta)^2} - \frac{3}{m} \frac{\mathrm{d}}{\mathrm{d}(2\beta)} + 2 \right] Z(2\beta);$

(xi) $\left\| a(f) \left[ a(f), e^{-\beta H_B} \right] \right\|_{\mathrm{HS}}^2 < \|f\|^4 \left[ \frac{1}{m^2} \frac{\mathrm{d}^2}{\mathrm{d}(2\beta)^2} - \frac{3}{m} \frac{\mathrm{d}}{\mathrm{d}(2\beta)} + 2 \right] Z(2\beta);$

(xii) $\left\| a(f) \left[ a^\dagger(f), e^{-\beta H_B} \right] \right\|_{\mathrm{HS}}^2 < \|f\|^4 \left[ \frac{1}{m^2} \frac{\mathrm{d}^2}{\mathrm{d}(2\beta)^2} - \frac{2}{m} \frac{\mathrm{d}}{\mathrm{d}(2\beta)} + 1 \right] Z(2\beta);$

(xiii) $\left\| \left[ a^\dagger(f), a(f) e^{-\beta H_B} \right] \right\|_{\mathrm{HS}}^2 < \|f\|^4 \left[ \frac{1}{m^2} \frac{\mathrm{d}^2}{\mathrm{d}(2\beta)^2} - \frac{2}{m} \frac{\mathrm{d}}{\mathrm{d}(2\beta)} + 1 \right] Z(2\beta);$

(xiv) $\left\| \left[ a^\dagger(f), a^\dagger(f) e^{-\beta H_B} \right] \right\|_{\mathrm{HS}}^2 < \|f\|^4 \left[ \frac{1}{m^2} \frac{\mathrm{d}^2}{\mathrm{d}(2\beta)^2} - \frac{3}{m} \frac{\mathrm{d}}{\mathrm{d}(2\beta)} + 2 \right] Z(2\beta);$

(xv) $\left\| \left[ a(f), a(f) e^{-\beta H_B} \right] \right\|_{\mathrm{HS}}^2 < \|f\|^4 \left[ \frac{1}{m^2} \frac{\mathrm{d}^2}{\mathrm{d}(2\beta)^2} - \frac{3}{m} \frac{\mathrm{d}}{\mathrm{d}(2\beta)} + 2 \right] Z(2\beta);$

(xvi) $\left\| \left[ a^\dagger(f), \left[ a^\dagger(f), e^{-\beta H_B} \right] \right] \right\|_{\mathrm{HS}}^2 < 4 \|f\|^4 \left[ \frac{1}{m^2} \frac{\mathrm{d}^2}{\mathrm{d}(2\beta)^2} - \frac{3}{m} \frac{\mathrm{d}}{\mathrm{d}(2\beta)} + 2 \right] Z(2\beta);$

(xvii) $\left\| \left[ a(f), \left[ a(f), e^{-\beta H_B} \right] \right] \right\|_{\mathrm{HS}}^2 < 4 \|f\|^4 \left[ \frac{1}{m^2} \frac{\mathrm{d}^2}{\mathrm{d}(2\beta)^2} - \frac{3}{m} \frac{\mathrm{d}}{\mathrm{d}(2\beta)} + 2 \right] Z(2\beta);$

(xviii) $\left\| \left[ H_B, a(f) e^{-\beta H_B} \right] \right\|_{\mathrm{HS}}^2 \leq -\frac{\|\omega f\|^2}{m} \frac{\mathrm{d}}{\mathrm{d}(2\beta)} Z(2\beta);$

(xix) $\left\| \left[ H_B, a^\dagger(f) e^{-\beta H_B} \right] \right\|_{\mathrm{HS}}^2 \leq \|\omega f\|^2 \left[ -\frac{1}{m} \frac{\mathrm{d}}{\mathrm{d}(2\beta)} + 1 \right] Z(2\beta);$

(xx) $\left\| \left[ H_B, \left[ a^\dagger(f), e^{-\beta H_B} \right] \right] \right\|_{\mathrm{HS}}^2 < \|\omega f\|^2 \left[ -\frac{1}{m} \frac{\mathrm{d}}{\mathrm{d}(2\beta)} + 1 \right] Z(2\beta);$

*(xxi)* $\left\| \left[ H_{\mathrm{B}}, [a(f), \mathrm{e}^{-\beta H_{\mathrm{B}}}] \right] \right\|_{\mathrm{HS}}^2 < \|\omega f\|^2 \left[ -\frac{1}{m} \frac{\mathrm{d}}{\mathrm{d}(2\beta)} + 1 \right] Z(2\beta)$.

*Proof.* Due to Assumption 5.2, it follows from Proposition B.1 that the function $\beta \to Z(\beta)$ is twice differentiable. Therefore, we can write all Hilbert–Schmidt norms in terms of derivatives of $Z(\beta)$. We proceed item by item starting with (i):

$$\left\| \mathrm{e}^{-\beta H_{\mathrm{B}}} \right\|_{\mathrm{HS}}^2 = \sum_{\boldsymbol{n}} \left\| \mathrm{e}^{-\beta H_{\mathrm{B}}} \ket{\boldsymbol{n}} \right\|^2$$

$$= \sum_{\boldsymbol{n}} \mathrm{e}^{-2\beta \sum_i \omega_i n_i}$$

$$= Z(2\beta). \tag{116}$$

To prove the inequalities, we notice that

$$\sum_j n_j |f_j|^2 \leq \|f\|^2 \sum_j n_j; \tag{117}$$

$$\sum_j n_j \leq \frac{1}{m} \sum_j n_j \omega_j. \tag{118}$$

Combining these two inequalities, we immediately get

$$\sum_j n_j |f_j|^2 \leq \frac{\|f\|^2}{m} \sum_j n_j \omega_j. \tag{119}$$

We use Eq. (119) for the inequality (ii):

$$\left\| a(f) \mathrm{e}^{-\beta H_{\mathrm{B}}} \right\|_{\mathrm{HS}}^2 = \sum_{\boldsymbol{n}} \left\| a(f) \mathrm{e}^{-\beta H_{\mathrm{B}}} \ket{\boldsymbol{n}} \right\|^2$$

$$\leq \frac{\|f\|^2}{m} \sum_{\boldsymbol{n}} \left( \sum_i n_i \omega_i \right) \mathrm{e}^{-2\beta \sum_i n_i \omega_i}$$

$$= \frac{\|f\|^2}{m} \sum_{\boldsymbol{n}} -\frac{\mathrm{d}}{\mathrm{d}(2\beta)} \mathrm{e}^{-2\beta \sum_i n_i \omega_i}$$

$$= -\frac{\|f\|^2}{m} \frac{\mathrm{d}}{\mathrm{d}(2\beta)} Z(2\beta). \tag{120}$$

Likewise, we can proceed for the creation operator to show (iii):

$$\left\| a^\dagger(f) \mathrm{e}^{-\beta H_{\mathrm{B}}} \right\|_{\mathrm{HS}}^2 = \sum_{\boldsymbol{n}} \left\| a^\dagger(f) \mathrm{e}^{-\beta H_{\mathrm{B}}} \ket{\boldsymbol{n}} \right\|^2$$

$$\leq \sum_{\boldsymbol{n}} \frac{\|f\|^2}{m} \left( \sum_i n_i \omega_i \right) \mathrm{e}^{-2\beta \sum_i n_i \omega_i} + \|f\|^2 \, \mathrm{e}^{-2\beta \sum_i n_i \omega_i}$$

$$= \sum_{\boldsymbol{n}} \frac{\|f\|^2}{2m} \left( -\frac{\mathrm{d}}{\mathrm{d}\beta} \right) \mathrm{e}^{-2\beta \sum_i n_i \omega_i} + \|f\|^2 \, \mathrm{e}^{-2\beta \sum_i n_i \omega_i}$$

$$= -\frac{\|f\|^2}{m} \frac{\mathrm{d}}{\mathrm{d}(2\beta)} Z(2\beta) + \|f\|^2 \, Z(2\beta). \tag{121}$$

For the squared annihilation operator, we obtain (iv):

$$\left\|a(f)^2 e^{-\beta H_B}\right\|_{HS}^2 = \sum_{\boldsymbol{n}} \left\|a(f)^2 e^{-\beta H_B} |\boldsymbol{n}\rangle\right\|^2$$

$$= \sum_{\boldsymbol{n}} \left\|\sum_k f_k^* a_k \sum_j f_j^* \sqrt{n_j} e^{-\beta \sum_i n_i \omega_i} |\boldsymbol{n}(n_j \to n_j - 1)\rangle\right\|^2$$

$$= \sum_{\boldsymbol{n}} \left\|\sum_{j \neq k} f_k^* f_j^* \sqrt{n_j} \sqrt{n_k} e^{-\beta \sum_i n_i \omega_i} |\boldsymbol{n}(n_j \to n_j - 1, n_k \to n_k - 1)\rangle\right\|^2$$

$$+ \left\|\sum_j \left(f_j^*\right)^2 \sqrt{n_j} \sqrt{n_j - 1} e^{-\beta \sum_i n_i \omega_i} |\boldsymbol{n}(n_j \to n_j - 2)\rangle\right\|^2$$

$$\leq \|f\|^4 \sum_{\boldsymbol{n}} \left(\sum_{j \neq k} n_j n_k + \sum_j n_j (n_j - 1)\right) e^{-2\beta \sum_i n_i \omega_i}$$

$$< \frac{\|f\|^4}{m^2} \sum_{\boldsymbol{n}} \left(\sum_i n_i \omega_i\right)^2 e^{-2\beta \sum_i n_i \omega_i}$$

$$= \frac{\|f\|^4}{4m^2} \sum_{\boldsymbol{n}} \frac{d^2}{d\beta^2} e^{-2\beta \sum_i n_i \omega_i}$$

$$= \frac{\|f\|^4}{m^2} \frac{d^2}{d(2\beta)^2} Z(2\beta). \tag{122}$$

To compute the term involving the squared creation operator, first recall that the notation $|\boldsymbol{n}(n_j \to n_j + 1)\rangle$ refers to the vector obtained from $|\boldsymbol{n}\rangle = |n_1, n_2, \ldots\rangle$ by replacing the entry $n_j$ with $n_j + 1$, i.e. $|\boldsymbol{n}(n_j \to n_j + 1)\rangle = |n_1, \ldots, n_{j-1}, n_j + 1, n_{j+1}, \ldots\rangle$. Then, the relation (v) is proven as follows:

$$\left\|a^\dagger(f)^2 e^{-\beta H_B}\right\|_{HS}^2 = \sum_{\boldsymbol{n}} \left\|a^\dagger(f)^2 e^{-\beta H_B} |\boldsymbol{n}\rangle\right\|^2$$

$$= \sum_{\boldsymbol{n}} \left\|e^{-\beta \sum_i n_i \omega_i} a^\dagger(f) \sum_j f_j \sqrt{n_j + 1} |\boldsymbol{n}(n_j \to n_j + 1)\rangle\right\|^2$$

$$= \sum_{\boldsymbol{n}} \left\|e^{-\beta \sum_i n_i \omega_i} \left[\sum_{j \neq k} f_j f_k \sqrt{n_j + 1} \sqrt{n_k + 1} |\boldsymbol{n}(n_j \to n_j + 1, n_k \to n_k + 1)\rangle\right.\right.$$

$$\left.\left.+ \sum_j f_j^2 \sqrt{n_j + 1} \sqrt{n_j + 2} |\boldsymbol{n}(n_j \to n_j + 2)\rangle\right]\right\|^2$$

$$= \sum_{\boldsymbol{n}} e^{-2\beta \sum_i n_i \omega_i} \left[\sum_{j \neq k} |f_j|^2 |f_k|^2 (n_j + 1)(n_k + 1) + \sum_j |f_j|^4 (n_j + 1)(n_j + 2)\right]$$

$$< \|f\|^4 \sum_{\bm{n}} \mathrm{e}^{-2\beta \sum_i n_i \omega_i} \left[ \sum_{j \neq k} (n_j + 1)(n_k + 2) + \sum_j (n_j + 1)(n_j + 2) \right]$$

$$< \|f\|^4 \sum_{\bm{n}} \mathrm{e}^{-2\beta \sum_i n_i \omega_i} \left[ \sum_j n_j + 2 \right] \left[ \sum_j n_j + 1 \right]$$

$$= \|f\|^4 \sum_{\bm{n}} \mathrm{e}^{-2\beta \sum_i n_i \omega_i} \left[ \left( \sum_j n_j \right)^2 + 3 \left( \sum_j n_j \right) + 2 \right]$$

$$\leq \|f\|^4 \sum_{\bm{n}} \mathrm{e}^{-2\beta \sum_i n_i \omega_i} \left[ \frac{1}{m^2} \left( \sum_j n_j \omega_j \right)^2 + \frac{3}{m} \left( \sum_j n_j \omega_j \right) + 2 \right]$$

$$= \|f\|^4 \left[ \frac{1}{m^2} \frac{\mathrm{d}^2}{\mathrm{d}(2\beta)^2} - \frac{3}{m} \frac{\mathrm{d}}{\mathrm{d}(2\beta)} + 2 \right] Z(2\beta). \tag{123}$$

Next, we show the inequalities (vi):

$$\left\| a^\dagger(f) a(f) \mathrm{e}^{-\beta H_{\mathrm{B}}} \right\|_{\mathrm{HS}}^2 = \sum_{\bm{n}} \left\| a^\dagger(f) a(f) \mathrm{e}^{-\beta H_{\mathrm{B}}} \ket{\bm{n}} \right\|^2$$

$$= \sum_{\bm{n}} \left\| \sum_j a^\dagger(f) f_j^* \sqrt{n_j} \mathrm{e}^{-\beta \sum_i \omega_i n_i} \ket{\bm{n}(n_j \to n_j - 1)} \right\|^2$$

$$= \sum_{\bm{n}} \left\| \sum_{j \neq k} f_k f_j^* \sqrt{n_k + 1} \sqrt{n_j} \mathrm{e}^{-\beta \sum_i \omega_i n_i} \ket{\bm{n}(n_j \to n_j - 1, n_k \to n_k + 1)} \right.$$

$$\left. + \sum_j |f_j| \, n_j \mathrm{e}^{-\beta \sum_i \omega_i n_i} \ket{\bm{n}} \right\|^2$$

$$< \|f\|^4 \sum_{\bm{n}} \left( \sum_i n_i + 1 \right) \left( \sum_i n_i \right) \mathrm{e}^{-2\beta \sum_i \omega_i n_i}$$

$$\leq \|f\|^4 \sum_{\bm{n}} \mathrm{e}^{-2\beta \sum_i n_i \omega_i} \left[ \frac{1}{m^2} \left( \sum_j n_j \omega_j \right)^2 + \frac{1}{m} \left( \sum_j n_j \omega_j \right) \right]$$

$$= \|f\|^4 \left[ \frac{1}{m^2} \frac{\mathrm{d}^2}{\mathrm{d}(2\beta)^2} - \frac{1}{m} \frac{\mathrm{d}}{\mathrm{d}(2\beta)} \right] Z(2\beta), \tag{124}$$

and (vii):

$$\left\| a(f) a^\dagger(f) \mathrm{e}^{-\beta H_{\mathrm{B}}} \right\|_{\mathrm{HS}}^2 = \sum_{\bm{n}} \left\| a(f) a^\dagger(f) \mathrm{e}^{-\beta H_{\mathrm{B}}} \ket{\bm{n}} \right\|^2$$

$$= \sum_{\bm{n}} \left\| \mathrm{e}^{-\beta \sum_i n_i \omega_i} a(f) \sum_j f_j \sqrt{n_j + 1} \ket{\bm{n}(n_j \to n_j + 1)} \right\|^2$$

$$
= \sum_{\boldsymbol{n}} \left\| \mathrm{e}^{-\beta \sum_i n_i \omega_i} \left[ \sum_{j \neq k} f_j f_k^* \sqrt{n_j + 1} \sqrt{n_k} \, |\boldsymbol{n}(n_j \to n_j + 1, n_k \to n_k - 1)\rangle \right. \right.
$$

$$
\left. \left. + \sum_j |f_j|^2 (n_j + 1) \, |\boldsymbol{n}\rangle \right] \right\|^2
$$

$$
< \|f\|^4 \sum_{\boldsymbol{n}} \mathrm{e}^{-2\beta \sum_i n_i \omega_i} \left[ \sum_j n_j + 1 \right]^2
$$

$$
\leq \|f\|^4 \sum_{\boldsymbol{n}} \mathrm{e}^{-2\beta \sum_i n_i \omega_i} \left[ \frac{1}{m^2} \left( \sum_j n_j \omega_j \right)^2 + \frac{2}{m} \left( \sum_j n_j \omega_j \right) + 1 \right]
$$

$$
= \|f\|^4 \left[ \frac{1}{m^2} \frac{\mathrm{d}^2}{\mathrm{d}(2\beta)^2} - \frac{2}{m} \frac{\mathrm{d}}{\mathrm{d}(2\beta)} + 1 \right] Z(2\beta). \tag{125}
$$

For the commutators of creation and annihilation operators with the Gibbs state, we use Lemma C.1 to prove (viii):

$$
\left\| \left[ a^\dagger(f), \mathrm{e}^{-\beta H_{\mathrm{B}}} \right] \right\|_{\mathrm{HS}}^2 = \sum_{\boldsymbol{n}} \left\| \left[ a^\dagger(f), \mathrm{e}^{-\beta H_{\mathrm{B}}} \right] |\boldsymbol{n}\rangle \right\|^2
$$

$$
= \sum_{\boldsymbol{n}} \left\| \sum_j f_j \sqrt{n_j + 1} \mathrm{e}^{-\beta \sum_i \omega_i n_i} \left( 1 - \mathrm{e}^{-\beta \omega_j} \right) |\boldsymbol{n}(n_j \to n_j + 1)\rangle \right\|^2
$$

$$
< \sum_{\boldsymbol{n}} \frac{\|f\|^2}{m} \left( \sum_i n_i \omega_i \right) \mathrm{e}^{-2\beta \sum_i n_i \omega_i} + \|f\|^2 \mathrm{e}^{-2\beta \sum_i n_i \omega_i}
$$

$$
= -\frac{\|f\|^2}{m} \frac{\mathrm{d}}{\mathrm{d}(2\beta)} Z(2\beta) + \|f\|^2 Z(2\beta), \tag{126}
$$

and (ix):

$$
\left\| \left[ a(f), \mathrm{e}^{-\beta H_{\mathrm{B}}} \right] \right\|_{\mathrm{HS}}^2 = \left\| \left[ a(f), \mathrm{e}^{-\beta H_{\mathrm{B}}} \right]^\dagger \right\|_{\mathrm{HS}}^2
$$

$$
= \left\| - \left[ a^\dagger(f), \mathrm{e}^{-\beta H_{\mathrm{B}}} \right] \right\|_{\mathrm{HS}}^2
$$

$$
< -\frac{\|f\|^2}{m} \frac{\mathrm{d}}{\mathrm{d}(2\beta)} Z(2\beta) + \|f\|^2 Z(2\beta). \tag{127}
$$

Next, we have creation/annihilation operators applied to the commutators. This is (x),

$$
\left\| a^\dagger(f) \left[ a^\dagger(f), \mathrm{e}^{-\beta H_{\mathrm{B}}} \right] \right\|_{\mathrm{HS}}^2 = \sum_{\boldsymbol{n}} \left\| a^\dagger(f) \left[ a^\dagger(f), \mathrm{e}^{-\beta H_{\mathrm{B}}} \right] |\boldsymbol{n}\rangle \right\|^2
$$

$$
= \sum_{\boldsymbol{n}} \left\| a^\dagger(f) \sum_j f_j \sqrt{n_j + 1} \mathrm{e}^{-\beta \sum_i \omega_i n_i} \left( 1 - \mathrm{e}^{-\beta \omega_j} \right) |\boldsymbol{n}(n_j \to n_j + 1)\rangle \right\|^2
$$

$$
= \sum_{\boldsymbol{n}} \left\| \sum_{j \neq k} f_j f_k \sqrt{n_j + 1} \sqrt{n_k + 1} \mathrm{e}^{-\beta \sum_i \omega_i n_i} \left( 1 - \mathrm{e}^{-\beta \omega_j} \right) \right.
$$

$$\times \left| \boldsymbol{n}(n_j \to n_j + 1, n_k \to n_k + 1) \right\rangle$$

$$+ \sum_j f_j^2 \sqrt{n_j + 1} \sqrt{n_j + 2} \mathrm{e}^{-\beta \sum_i \omega_i n_i} \left( 1 - \mathrm{e}^{-\beta \omega_j} \right) \left| \boldsymbol{n}(n_j \to n_j + 2) \right\rangle \Big\|^2$$

$$< \|f\|^4 \sum_{\boldsymbol{n}} \mathrm{e}^{-2\beta \sum_i \omega_i n_i} \left( \sum_j \left( 1 - \mathrm{e}^{-\beta \omega_j} \right)^2 (n_j + 1) \right) \left( \sum_k (n_k + 2) \right)$$

$$< \|f\|^4 \sum_{\boldsymbol{n}} \mathrm{e}^{-2\beta \sum_i \omega_i n_i} \left[ \sum_i n_i + 2 \right] \left[ \sum_i n_i + 1 \right]$$

$$\le \|f\|^4 \left[ \frac{1}{m^2} \frac{\mathrm{d}^2}{\mathrm{d}(2\beta)^2} - \frac{3}{m} \frac{\mathrm{d}}{\mathrm{d}(2\beta)} + 2 \right] Z(2\beta), \tag{128}$$

(xi):

$$\left\| a(f) \left[ a(f), \mathrm{e}^{-\beta H_{\mathrm{B}}} \right] \right\|_{\mathrm{HS}}^2 = \left\| \left[ a(f), \mathrm{e}^{-\beta H_{\mathrm{B}}} \right]^\dagger a^\dagger(f) \right\|_{\mathrm{HS}}^2$$

$$= \left\| \left[ a^\dagger(f), \mathrm{e}^{-\beta H_{\mathrm{B}}} \right]^\dagger a^\dagger(f) \right\|_{\mathrm{HS}}^2$$

$$= \sum_{\boldsymbol{n}} \left\| \left[ a^\dagger(f), \mathrm{e}^{-\beta H_{\mathrm{B}}} \right]^\dagger a^\dagger(f) \left| \boldsymbol{n} \right\rangle \right\|^2$$

$$= \sum_{\boldsymbol{n}} \left\| \sum_j a^\dagger(f) \mathrm{e}^{-\beta H_{\mathrm{B}}} f_j \sqrt{n_j + 1} \left| \boldsymbol{n}(n_j \to n_j + 1) \right\rangle \right.$$

$$\left. - \sum_j \mathrm{e}^{-\beta H_{\mathrm{B}}} a^\dagger(f) f_j \sqrt{n_j + 1} \left| \boldsymbol{n}(n_j \to n_j + 1) \right\rangle \right\|^2$$

$$= \sum_{\boldsymbol{n}} \left\| \sum_{j \ne k} f_k f_j \sqrt{n_k + 1} \sqrt{n_j + 1} \mathrm{e}^{-\beta \sum_i n_i \omega_i} \mathrm{e}^{-\beta \omega_j} \left| \boldsymbol{n}(n_j \to n_j + 1, n_k \to n_k + 1) \right\rangle \right.$$

$$+ \sum_j f_j^2 \sqrt{n_j + 1} \sqrt{n_j + 2} \mathrm{e}^{-\beta \sum_i n_i \omega_i} \mathrm{e}^{-\beta \omega_j} \left| \boldsymbol{n}(n_j \to n_j + 2) \right\rangle$$

$$- \sum_{j \ne k} \mathrm{e}^{-\beta \sum_i n_i \omega_i} \mathrm{e}^{-\beta (\omega_j + \omega_k)} f_k f_j \sqrt{n_j + 1} \sqrt{n_k + 1} \left| \boldsymbol{n}(n_j \to n_j + 1, n_k \to n_k + 1) \right\rangle$$

$$\left. - \sum_j \mathrm{e}^{-\beta \sum_i n_i \omega_i} \mathrm{e}^{-2\beta \omega_j} f_j^2 \sqrt{n_j + 1} \sqrt{n_j + 2} \left| \boldsymbol{n}(n_j \to n_j + 2) \right\rangle \right\|^2$$

$$= \sum_{\boldsymbol{n}} \left\| \sum_{j \ne k} f_k f_j \sqrt{n_k + 1} \sqrt{n_j + 1} \mathrm{e}^{-\beta \sum_i n_i \omega_i} \left( \mathrm{e}^{-\beta \omega_j} - \mathrm{e}^{-\beta(\omega_j + \omega_k)} \right) \right.$$

$$\times \left| \boldsymbol{n}(n_j \to n_j + 1, n_k \to n_k + 1) \right\rangle$$

$$\left. + \sum_j f_j^2 \sqrt{n_j + 1} \sqrt{n_j + 2} \mathrm{e}^{-\beta \sum_i n_i \omega_i} \left( \mathrm{e}^{-\beta \omega_j} - \mathrm{e}^{-2\beta \omega_j} \right) \left| \boldsymbol{n}(n_j \to n_j + 2) \right\rangle \right\|^2$$

$$< \|f\|^4 \sum_{\boldsymbol{n}} \left( \sum_i n_i + 1 \right) \left( \sum_i n_i + 2 \right) e^{-2\beta \sum_i n_i \omega_i}$$

$$\leq \|f\|^4 \sum_{\boldsymbol{n}} \left[ \frac{1}{m^2} \left( \sum_i n_i \omega_i \right)^2 + \frac{3}{m} \left( \sum_i n_i \omega_i \right) + 2 \right] e^{-2\beta \sum_i n_i \omega_i}$$

$$= \|f\|^4 \left[ \frac{1}{m^2} \frac{\mathrm{d}^2}{\mathrm{d}(2\beta)^2} - \frac{3}{m} \frac{\mathrm{d}}{\mathrm{d}(2\beta)} + 2 \right] Z(2\beta), \tag{129}$$

and (xii):

$$\left\| a(f) \left[ a^\dagger(f), e^{-\beta H_{\mathrm{B}}} \right] \right\|_{\mathrm{HS}}^2 = \sum_{\boldsymbol{n}} \left\| a(f) \left[ a^\dagger(f), e^{-\beta H_{\mathrm{B}}} \right] |\boldsymbol{n}\rangle \right\|^2$$

$$= \sum_{\boldsymbol{n}} \left\| a(f) \sum_j f_j \sqrt{n_j + 1} e^{-\beta \sum_i \omega_i n_i} \left( 1 - e^{-\beta \omega_j} \right) |\boldsymbol{n}(n_j \to n_j + 1)\rangle \right\|^2$$

$$= \sum_{\boldsymbol{n}} \left\| \sum_{j \neq k} f_j f_k^* \sqrt{n_j + 1} \sqrt{n_k} e^{-\beta \sum_i \omega_i n_i} \left( 1 - e^{-\beta \omega_j} \right) |\boldsymbol{n}(n_j \to n_j + 1, n_k \to n_k - 1)\rangle \right.$$

$$\left. + \sum_j |f_j|^2 (n_j + 1) e^{-\beta \sum_i \omega_i n_i} \left( 1 - e^{-\beta \omega_j} \right) |\boldsymbol{n}\rangle \right\|^2$$

$$< \|f\|^4 \sum_{\boldsymbol{n}} e^{-2\beta \sum_i \omega_i n_i} \left( \sum_j \left( 1 - e^{-\beta \omega_j} \right)^2 (n_j + 1) \right) \left( \sum_k (n_k + 1) \right)$$

$$< \|f\|^4 \sum_{\boldsymbol{n}} e^{-2\beta \sum_i \omega_i n_i} \left( \sum_i n_i + 1 \right)^2$$

$$\leq \|f\|^4 \left[ \frac{1}{m^2} \frac{\mathrm{d}^2}{\mathrm{d}(2\beta)^2} - \frac{2}{m} \frac{\mathrm{d}}{\mathrm{d}(2\beta)} + 1 \right] Z(2\beta), \tag{130}$$

where we used Lemma C.1 in each computation. We continue with commutators of the form (xiii):

$$\left\| \left[ a^\dagger(f), a(f) e^{-\beta H_{\mathrm{B}}} \right] \right\|_{\mathrm{HS}}^2 = \sum_{\boldsymbol{n}} \left\| a^\dagger(f) a(f) e^{-\beta H_{\mathrm{B}}} |\boldsymbol{n}\rangle - a(f) e^{-\beta H_{\mathrm{B}}} a^\dagger(f) |\boldsymbol{n}\rangle \right\|^2$$

$$= \sum_{\boldsymbol{n}} \left\| \sum_j a^\dagger(f) f_j^* \sqrt{n_j} e^{-\beta \sum_i \omega_i n_i} |\boldsymbol{n}(n_j \to n_j - 1)\rangle \right.$$

$$\left. - \sum_k a(f) f_k \sqrt{n_k + 1} e^{-\beta \sum_i \omega_i n_i} e^{-\beta \omega_k} |\boldsymbol{n}(n_k \to n_k + 1)\rangle \right\|^2$$

$$= \sum_{\boldsymbol{n}} \left\| \sum_{j \neq k} f_j^* f_k \sqrt{n_j} \sqrt{n_k + 1} e^{-\beta \sum_i \omega_i n_i} \left( 1 - e^{-\beta \omega_k} \right) |\boldsymbol{n}(n_j \to n_j - 1, n_k \to n_k + 1)\rangle \right.$$

$$\left. + \sum_j |f_j|^2 n_j e^{-\beta \sum_i \omega_i n_i} |\boldsymbol{n}\rangle - \sum_j |f_j|^2 (n_j + 1) e^{-\beta \sum_i \omega_i n_i} e^{-\beta \omega_j} |\boldsymbol{n}\rangle \right\|^2$$

$$< \|f\|^4 \sum_{\boldsymbol{n}} \left(\sum_i n_i + 1\right)^2 \mathrm{e}^{-2\beta \sum_i \omega_i n_i}$$

$$\leq \|f\|^4 \sum_{\boldsymbol{n}} \mathrm{e}^{-2\beta \sum_i n_i \omega_i} \left[\frac{1}{m^2}\left(\sum_i n_i \omega_i\right)^2 + \frac{2}{m}\left(\sum_i n_i \omega_i\right) + 1\right]$$

$$= \|f\|^4 \left[\frac{1}{m^2}\frac{\mathrm{d}^2}{\mathrm{d}(2\beta)^2} - \frac{2}{m}\frac{\mathrm{d}}{\mathrm{d}(2\beta)} + 1\right] Z(2\beta), \tag{131}$$

(xiv):

$$\left\|\left[a^\dagger(f), a^\dagger(f)\mathrm{e}^{-\beta H_\mathrm{B}}\right]\right\|_\mathrm{HS} = \left\|a^\dagger(f)a^\dagger(f)\mathrm{e}^{-\beta H_\mathrm{B}} - a^\dagger(f)\mathrm{e}^{-\beta H_\mathrm{B}}a^\dagger(f)\right\|_\mathrm{HS}$$

$$= \left\|a^\dagger(f)\left[a^\dagger(f), \mathrm{e}^{-\beta H_\mathrm{B}}\right]\right\|_\mathrm{HS}$$

$$< \|f\|^4 \left[\frac{1}{m^2}\frac{\mathrm{d}^2}{\mathrm{d}(2\beta)^2} - \frac{3}{m}\frac{\mathrm{d}}{\mathrm{d}(2\beta)} + 2\right] Z(2\beta), \tag{132}$$

and (xv):

$$\left\|\left[a(f), a(f)\mathrm{e}^{-\beta H_\mathrm{B}}\right]\right\|_\mathrm{HS} = \left\|a(f)a(f)\mathrm{e}^{-\beta H_\mathrm{B}} - a(f)\mathrm{e}^{-\beta H_\mathrm{B}}a(f)\right\|_\mathrm{HS}$$

$$= \left\|a(f)\left[a(f), \mathrm{e}^{-\beta H_\mathrm{B}}\right]\right\|_\mathrm{HS}$$

$$< \|f\|^4 \left[\frac{1}{m^2}\frac{\mathrm{d}^2}{\mathrm{d}(2\beta)^2} - \frac{3}{m}\frac{\mathrm{d}}{\mathrm{d}(2\beta)} + 2\right] Z(2\beta). \tag{133}$$

Next, we have the double commutators. Starting with (xv), we have

$$\left\|\left[a^\dagger(f), \left[a^\dagger(f), \mathrm{e}^{-\beta H_\mathrm{B}}\right]\right]\right\|_\mathrm{HS}^2$$

$$= \left\|a^\dagger(f)\left[a^\dagger(f), \mathrm{e}^{-\beta H_\mathrm{B}}\right] + \mathrm{e}^{-\beta H_\mathrm{B}}a^\dagger(f)a^\dagger(f) - a^\dagger(f)\mathrm{e}^{-\beta H_\mathrm{B}}a^\dagger(f)\right\|_\mathrm{HS}^2$$

$$= \sum_{\boldsymbol{n}} \left\|\sum_j a^\dagger(f)f_j\sqrt{n_j+1}\mathrm{e}^{-\beta\sum_i \omega_i n_i}\left(1 - \mathrm{e}^{-\beta\omega_j}\right)|\boldsymbol{n}(n_j \to n_j+1)\rangle\right.$$

$$+ \sum_j \mathrm{e}^{-\beta H_\mathrm{B}}a^\dagger(f)f_j\sqrt{n_j+1}|\boldsymbol{n}(n_j \to n_j+1)\rangle$$

$$\left. - \sum_j a^\dagger(f)\mathrm{e}^{-\beta H_\mathrm{B}}f_j\sqrt{n_j+1}|\boldsymbol{n}(n_j \to n_j+1)\rangle\right\|^2$$

$$= \sum_{\boldsymbol{n}} \left\|\sum_{j\neq k} f_j f_k\sqrt{n_j+1}\sqrt{n_k+1}\mathrm{e}^{-\beta\sum_i \omega_i n_i}\left(1 - \mathrm{e}^{-\beta\omega_j}\right)|\boldsymbol{n}(n_j \to n_j+1, n_k \to n_k+1)\rangle\right.$$

$$+ \sum_j f_j^2\sqrt{n_j+1}\sqrt{n_j+2}\mathrm{e}^{-\beta\sum_i \omega_i n_i}\left(1 - \mathrm{e}^{-\beta\omega_j}\right)|\boldsymbol{n}(n_j \to n_j+2)\rangle$$

$$+ \sum_{j\neq k} \mathrm{e}^{-\beta\sum_i \omega_i n_i}\mathrm{e}^{-\beta\omega_j}\mathrm{e}^{-\beta\omega_k}f_j f_k\sqrt{n_j+1}\sqrt{n_k+1}|\boldsymbol{n}(n_j \to n_j+1, n_k \to n_k+1)\rangle$$

$$+ \sum_j \mathrm{e}^{-\beta\sum_i \omega_i n_i}\mathrm{e}^{-\beta\omega_j}\mathrm{e}^{-\beta\omega_j}f_j^2\sqrt{n_j+1}\sqrt{n_j+2}|\boldsymbol{n}(n_j \to n_j+2)\rangle$$

$$-\sum_{j\neq k}\mathrm{e}^{-\beta\sum_i\omega_i n_i}\mathrm{e}^{-\beta\omega_j}f_j f_k\sqrt{n_j+1}\sqrt{n_k+1}\,|\boldsymbol{n}(n_j\to n_j+1,n_k\to n_k+1)\rangle$$

$$\left.-\sum_j\mathrm{e}^{-\beta\sum_i\omega_i n_i}\mathrm{e}^{-\beta\omega_j}f_j^2\sqrt{n_j+1}\sqrt{n_j+2}\,|\boldsymbol{n}(n_j\to n_j+2)\rangle\right\|^2$$

$$=\sum_{\boldsymbol{n}}\left\|\sum_{j\neq k}f_j f_k\sqrt{n_j+1}\sqrt{n_k+1}\mathrm{e}^{-\beta\sum_i\omega_i n_i}\left(1-2\mathrm{e}^{-\beta\omega_j}+\mathrm{e}^{-\beta\omega_j}\mathrm{e}^{-\beta\omega_k}\right)\right.$$

$$\times\,|\boldsymbol{n}(n_j\to n_j+1,n_k\to n_k+1)\rangle$$

$$+\sum_j f_j^2\sqrt{n_j+1}\sqrt{n_j+2}\mathrm{e}^{-\beta\sum_i\omega_i n_i}\left(1-2\mathrm{e}^{-\beta\omega_j}+\mathrm{e}^{-\beta\omega_j}\mathrm{e}^{-\beta\omega_j}\right)$$

$$\left.\times\,|\boldsymbol{n}(n_j\to n_j+2)\rangle\right\|^2$$

$$<\|f\|^4\sum_{\boldsymbol{n}}\sum_{j,k}(n_j+2)(n_k+1)\,\mathrm{e}^{-2\beta\sum_i\omega_i n_i}\left(1-2\mathrm{e}^{-\beta\omega_j}+\mathrm{e}^{-\beta\omega_j}\mathrm{e}^{-\beta\omega_k}\right)^2$$

$$<4\|f\|^4\sum_{\boldsymbol{n}}\sum_{j,k}(n_j+2)(n_k+1)\,\mathrm{e}^{-2\beta\sum_i\omega_i n_i}\left(1-\mathrm{e}^{-\beta\omega_j}\right)^2$$

$$<4\|f\|^4\sum_{\boldsymbol{n}}\left(\sum_i n_i+2\right)\left(\sum_i n_i+1\right)\mathrm{e}^{-2\beta\sum_i\omega_i n_i}$$

$$\leq4\|f\|^4\sum_{\boldsymbol{n}}\left[\frac{1}{m^2}\left(\sum_j n_j\omega_j\right)^2+\frac{3}{m}\left(\sum_j n_j\omega_j\right)+2\right]\mathrm{e}^{-2\beta\sum_i\omega_i n_i}$$

$$=4\|f\|^4\left[\frac{1}{m^2}\frac{\mathrm{d}^2}{\mathrm{d}(2\beta)^2}-\frac{3}{m}\frac{\mathrm{d}}{\mathrm{d}(2\beta)}+2\right]Z(2\beta). \tag{134}$$

Furthermore, (xvii) reads

$$\left\|\left[a(f),\left[a(f),\mathrm{e}^{-\beta H_{\mathrm{B}}}\right]\right]\right\|_{\mathrm{HS}}^2$$

$$=\left\|a^\dagger(f)\left[a^\dagger(f),\mathrm{e}^{-\beta H_{\mathrm{B}}}\right]+\mathrm{e}^{-\beta H_{\mathrm{B}}}a^\dagger(f)a^\dagger(f)-a^\dagger(f)\mathrm{e}^{-\beta H_{\mathrm{B}}}a^\dagger(f)\right\|_{\mathrm{HS}}^2$$

$$=\left\|\left[a^\dagger(f),\left[a^\dagger(f),\mathrm{e}^{-\beta H_{\mathrm{B}}}\right]\right]\right\|_{\mathrm{HS}}^2$$

$$<4\|f\|^4\left[\frac{1}{m^2}\frac{\mathrm{d}^2}{\mathrm{d}(2\beta)^2}-\frac{3}{m}\frac{\mathrm{d}}{\mathrm{d}(2\beta)}+2\right]Z(2\beta). \tag{135}$$

Finally, we compute the commutators involving $H_{\mathrm{B}}$, beginning with (xviii):

$$\left\|\left[H_{\mathrm{B}},a(f)\mathrm{e}^{-\beta H_{\mathrm{B}}}\right]\right\|_{\mathrm{HS}}^2=\sum_{\boldsymbol{n}}\left\|H_{\mathrm{B}}a(f)\mathrm{e}^{-\beta H_{\mathrm{B}}}\,|\boldsymbol{n}\rangle-a(f)\mathrm{e}^{-\beta H_{\mathrm{B}}}H_{\mathrm{B}}\,|\boldsymbol{n}\rangle\right\|^2$$

$$=\sum_{\boldsymbol{n}}\left\|\sum_j\left((n_j-1)\omega_j+\sum_{i\neq j}n_i\omega_i\right)f_j^*\sqrt{n_j}\mathrm{e}^{-\beta\sum_i\omega_i n_i}\,|\boldsymbol{n}(n_j\to n_j-1)\rangle\right.$$

$$
\qquad - \sum_j f_j^* \sqrt{n_j} \mathrm{e}^{-\beta \sum_i \omega_i n_i} \left( \sum_i n_i \omega_i \right) |\boldsymbol{n}(n_j \to n_j - 1)\rangle \Bigg\|^2
$$

$$
= \sum_{\boldsymbol{n}} \left\| \sum_j -\omega_j f_j^* \sqrt{n_j} \mathrm{e}^{-\beta \sum_i \omega_i n_i} |\boldsymbol{n}(n_j \to n_j - 1)\rangle \right\|^2
$$

$$
\leq \frac{\|\omega f\|^2}{m} \sum_{\boldsymbol{n}} \left( \sum_i \omega_i n_i \right) \mathrm{e}^{-2\beta \sum_i \omega_i n_i}
$$

$$
= -\frac{\|\omega f\|^2}{m} \frac{\mathrm{d}}{\mathrm{d}(2\beta)} Z(2\beta). \tag{136}
$$

Likewise (xix) reads

$$
\left\| \left[ H_{\mathrm{B}}, a^\dagger(f) \mathrm{e}^{-\beta H_{\mathrm{B}}} \right] \right\|_{\mathrm{HS}}^2 = \sum_{\boldsymbol{n}} \left\| H_{\mathrm{B}} a^\dagger(f) \mathrm{e}^{-\beta H_{\mathrm{B}}} |\boldsymbol{n}\rangle - a^\dagger(f) \mathrm{e}^{-\beta H_{\mathrm{B}}} H_{\mathrm{B}} |\boldsymbol{n}\rangle \right\|^2
$$

$$
= \sum_{\boldsymbol{n}} \left\| \sum_j \left( (n_j + 1)\omega_j + \sum_{i \neq j} n_i \omega_i \right) f_j \sqrt{n_j + 1} \mathrm{e}^{-\beta \sum_i \omega_i n_i} |\boldsymbol{n}(n_j \to n_j + 1)\rangle \right.
$$

$$
\qquad \left. - \sum_j f_j \sqrt{n_j + 1} \mathrm{e}^{-\beta \sum_i \omega_i n_i} \left( \sum_i n_i \omega_i \right) |\boldsymbol{n}(n_j \to n_j + 1)\rangle \right\|^2
$$

$$
= \sum_{\boldsymbol{n}} \left\| \sum_j \omega_j f_j \sqrt{n_j + 1} \mathrm{e}^{-\beta \sum_i \omega_i n_i} |\boldsymbol{n}(n_j \to n_j + 1)\rangle \right\|^2
$$

$$
\leq \frac{\|\omega f\|^2}{m} \sum_{\boldsymbol{n}} \left( \sum_i \omega_i n_i \right) \mathrm{e}^{-2\beta \sum_i \omega_i n_i} + \|\omega f\|^2 \sum_{\boldsymbol{n}} \mathrm{e}^{-2\beta \sum_i \omega_i n_i}
$$

$$
= -\frac{\|\omega f\|^2}{m} \frac{\mathrm{d}}{\mathrm{d}(2\beta)} Z(2\beta) + \|\omega f\|^2 Z(2\beta). \tag{137}
$$

The last two are the double commutators with $H_{\mathrm{B}}$, i.e. (xx):

$$
\left\| \left[ H_{\mathrm{B}}, \left[ a^\dagger(f), \mathrm{e}^{-\beta H_{\mathrm{B}}} \right] \right] \right\|_{\mathrm{HS}}^2 = \sum_{\boldsymbol{n}} \left\| H_{\mathrm{B}} \left[ a^\dagger(f), \mathrm{e}^{-\beta H_{\mathrm{B}}} \right] |\boldsymbol{n}\rangle - \left[ a^\dagger(f), \mathrm{e}^{-\beta H_{\mathrm{B}}} \right] H_{\mathrm{B}} |\boldsymbol{n}\rangle \right\|^2
$$

$$
= \sum_{\boldsymbol{n}} \left\| \sum_j \left( \left[ \omega_j (n_j + 1) + \sum_{i \neq j} \omega_i n_i \right] f_j \sqrt{n_j + 1} \mathrm{e}^{-\beta \sum_i n_i \omega_i} \left[ 1 - \mathrm{e}^{-\beta \omega_j} \right] \right. \right.
$$

$$
\qquad \left. \left. - \left( \sum_i \omega_i n_i \right) f_j \sqrt{n_j + 1} \mathrm{e}^{-\beta \sum_i n_i \omega_i} \left[ 1 - \mathrm{e}^{-\beta \omega_j} \right] \right) |\boldsymbol{n}(n_j \to n_j + 1)\rangle \right\|^2
$$

$$
= \sum_{\boldsymbol{n}} \left\| \sum_j \omega_j f_j \sqrt{n_j + 1} \mathrm{e}^{-\beta \sum_i n_i \omega_i} \left[ 1 - \mathrm{e}^{-\beta \omega_j} \right] |\boldsymbol{n}(n_j \to n_j + 1)\rangle \right\|
$$

$$
< \frac{\|\omega f\|^2}{m} \sum_{\boldsymbol{n}} \left( \sum_i \omega_i n_i \right) \mathrm{e}^{-2\beta \sum_i \omega_i n_i} + \|\omega f\|^2 \sum_{\boldsymbol{n}} \mathrm{e}^{-2\beta \sum_i \omega_i n_i}
$$

$$= -\frac{\|\omega f\|^2}{m}\frac{\mathrm{d}}{\mathrm{d}(2\beta)}Z(2\beta) + \|\omega f\|^2\, Z(2\beta) \tag{138}$$

and (xxi):

$$\left\|\left[H_{\mathrm{B}},\left[a(f),\mathrm{e}^{-\beta H_{\mathrm{B}}}\right]\right]\right\|_{\mathrm{HS}}^2 = \left\|\left(H_{\mathrm{B}}\left[a(f),\mathrm{e}^{-\beta H_{\mathrm{B}}}\right] - \left[a(f),\mathrm{e}^{-\beta H_{\mathrm{B}}}\right]H_{\mathrm{B}}\right)^\dagger\right\|_{\mathrm{HS}}^2$$

$$= \left\|\left[H_{\mathrm{B}},\left[a^\dagger(f),\mathrm{e}^{-\beta H_{\mathrm{B}}}\right]\right]\right\|_{\mathrm{HS}}^2$$

$$< -\frac{\|\omega f\|^2}{m}\frac{\mathrm{d}}{\mathrm{d}(2\beta)}Z(2\beta) + \|\omega f\|^2\, Z(2\beta). \tag{139}$$

This completes the proof. $\qquad\square$

The remaining norms that we have not computed yet involve commutators with the annihilation operator $a(f)$. We can see from Lemma C.1 (ii) that these terms will have a factor $\left(1 - \mathrm{e}^{+\beta\omega_k}\right)^2$, which increases exponentially in $\beta$; therefore, we cannot simply bound this factor independently of $\beta$ as we have done for commutators with the creation operators $a^\dagger(f)$ in Lemma C.2 (where we used $\left(1 - \mathrm{e}^{-\beta\omega_k}\right)^2 \leq 1$). For this reason, we pursue a different strategy for the commutators containing $a(f)$, which employs the following lemma.

**Lemma C.3.** *The following inequality holds*

$$\left\|a^{\sharp_1}(f)\mathrm{e}^{-\beta H_{\mathrm{B}}}a^{\sharp_2}(f)\right\|_{\mathrm{HS}}^2 \leq \left\|a^{\sharp_1}(f)^\dagger a^{\sharp_1}(f)\mathrm{e}^{-\beta H_{\mathrm{B}}}\right\|_{\mathrm{HS}}\left\|a^{\sharp_2}(f)a^{\sharp_2}(f)^\dagger\mathrm{e}^{-\beta H_{\mathrm{B}}}\right\|_{\mathrm{HS}}, \tag{140}$$

*where $a^\sharp(f)$ denotes either the annihilation or creation operator.*

*Proof.*

$$\left\|a^{\sharp_1}(f)\mathrm{e}^{-\beta H_{\mathrm{B}}}a^{\sharp_2}(f)\right\|_{\mathrm{HS}}^2 = \mathrm{tr}\left(a^{\sharp_1}(f)\mathrm{e}^{-\beta H_{\mathrm{B}}}a^{\sharp_2}(f)a^{\sharp_2}(f)^\dagger\mathrm{e}^{-\beta H_{\mathrm{B}}}a^{\sharp_1}(f)^\dagger\right)$$

$$= \mathrm{tr}\left(a^{\sharp_1}(f)^\dagger a^{\sharp_1}(f)\mathrm{e}^{-\beta H_{\mathrm{B}}}a^{\sharp_2}(f)a^{\sharp_2}(f)^\dagger\mathrm{e}^{-\beta H_{\mathrm{B}}}\right)$$

$$= \left|\left\langle\mathrm{e}^{-\beta H_{\mathrm{B}}}a^{\sharp_1}(f)^\dagger a^{\sharp_1}(f), a^{\sharp_2}(f)a^{\sharp_2}(f)^\dagger\mathrm{e}^{-\beta H_{\mathrm{B}}}\right\rangle_{\mathrm{HS}}\right|$$

$$\leq \left\|\mathrm{e}^{-\beta H_{\mathrm{B}}}a^{\sharp_1}(f)^\dagger a^{\sharp_1}(f)\right\|_{\mathrm{HS}}\left\|a^{\sharp_2}(f)a^{\sharp_2}(f)^\dagger\mathrm{e}^{-\beta H_{\mathrm{B}}}\right\|_{\mathrm{HS}}$$

$$= \left\|a^{\sharp_1}(f)^\dagger a^{\sharp_1}(f)\mathrm{e}^{-\beta H_{\mathrm{B}}}\right\|_{\mathrm{HS}}\left\|a^{\sharp_2}(f)a^{\sharp_2}(f)^\dagger\mathrm{e}^{-\beta H_{\mathrm{B}}}\right\|_{\mathrm{HS}}, \tag{141}$$

where the third line uses that the Hilbert–Schmidt norm is induced by the Hilbert–Schmidt inner product $\langle\cdot,\cdot\rangle_{\mathrm{HS}}$ and the fourth line is the Cauchy–Schwarz inequality. $\qquad\square$

Using Lemma C.3, we can compute the remaining terms. We collect them in the following lemma.

**Lemma C.4.** *Under the same assumptions as in Lemma C.2 we have*

*(i)* $\left\|a^\dagger(f)\left[a(f),\mathrm{e}^{-\beta H_{\mathrm{B}}}\right]\right\|_{\mathrm{HS}}^2 < \|f\|^4\left(\sqrt{\left[\frac{1}{m^2}\frac{\mathrm{d}^2}{\mathrm{d}(2\beta)^2} - \frac{1}{m}\frac{\mathrm{d}}{\mathrm{d}(2\beta)}\right]Z(2\beta)}\right.$

$$\left.+\sqrt{\left[\frac{1}{m^2}\frac{\mathrm{d}^2}{\mathrm{d}(2\beta)^2} - \frac{2}{m}\frac{\mathrm{d}}{\mathrm{d}(2\beta)} + 1\right]Z(2\beta)}\right)^2;$$

*(ii)* $\left\|\left[a(f), a^\dagger(f)\mathrm{e}^{-\beta H_\mathrm{B}}\right]\right\|_\mathrm{HS}^2 < 4\left\|f\right\|^4 \left[\frac{1}{m^2}\frac{\mathrm{d}^2}{\mathrm{d}(2\beta)^2} - \frac{2}{m}\frac{\mathrm{d}}{\mathrm{d}(2\beta)} + 1\right] Z(2\beta);$

*(iii)* $\left\|\left[a(f), \left[a^\dagger(f), \mathrm{e}^{-\beta H_\mathrm{B}}\right]\right]\right\|_\mathrm{HS}^2 < 4\left\|f\right\|^4 \left(\sqrt{\left[\frac{1}{m^2}\frac{\mathrm{d}^2}{\mathrm{d}(2\beta)^2} - \frac{2}{m}\frac{\mathrm{d}}{\mathrm{d}(2\beta)} + 1\right] Z(2\beta)}\right.$

$$\left.+ \sqrt{\left[\frac{1}{m^2}\frac{\mathrm{d}^2}{\mathrm{d}(2\beta)^2} - \frac{1}{m}\frac{\mathrm{d}}{\mathrm{d}(2\beta)}\right] Z(2\beta)}\right)^2;$$

*(iv)* $\left\|\left[a^\dagger(f), \left[a(f), \mathrm{e}^{-\beta H_\mathrm{B}}\right]\right]\right\|_\mathrm{HS}^2 < 4\left\|f\right\|^4 \left(\sqrt{\left[\frac{1}{m^2}\frac{\mathrm{d}^2}{\mathrm{d}(2\beta)^2} - \frac{2}{m}\frac{\mathrm{d}}{\mathrm{d}(2\beta)} + 1\right] Z(2\beta)}\right.$

$$\left.+ \sqrt{\left[\frac{1}{m^2}\frac{\mathrm{d}^2}{\mathrm{d}(2\beta)^2} - \frac{1}{m}\frac{\mathrm{d}}{\mathrm{d}(2\beta)}\right] Z(2\beta)}\right)^2;$$

*Proof.* As before, we compute each term individually. We begin with (i):

$$\left\|a^\dagger(f)\left[a(f), \mathrm{e}^{-\beta H_\mathrm{B}}\right]\right\|_\mathrm{HS}^2 \le \left(\left\|a^\dagger(f)a(f)\mathrm{e}^{-\beta H_\mathrm{B}}\right\|_\mathrm{HS} + \left\|a^\dagger(f)\mathrm{e}^{-\beta H_\mathrm{B}}a(f)\right\|_\mathrm{HS}\right)^2$$

$$\le \left(\left\|a^\dagger(f)a(f)\mathrm{e}^{-\beta H_\mathrm{B}}\right\|_\mathrm{HS} + \sqrt{\|a(f)a^\dagger(f)\mathrm{e}^{-\beta H_\mathrm{B}}\|_\mathrm{HS}\,\|a(f)a^\dagger(f)\mathrm{e}^{-\beta H_\mathrm{B}}\|_\mathrm{HS}}\right)^2$$

$$< \left\|f\right\|^4 \left(\sqrt{\left[\frac{1}{m^2}\frac{\mathrm{d}^2}{\mathrm{d}(2\beta)^2} - \frac{1}{m}\frac{\mathrm{d}}{\mathrm{d}(2\beta)}\right] Z(2\beta)} + \sqrt{\left[\frac{1}{m^2}\frac{\mathrm{d}^2}{\mathrm{d}(2\beta)^2} - \frac{2}{m}\frac{\mathrm{d}}{\mathrm{d}(2\beta)} + 1\right] Z(2\beta)}\right)^2. \tag{142}$$

For (ii), we have

$$\left\|\left[a(f), a^\dagger(f)\mathrm{e}^{-\beta H_\mathrm{B}}\right]\right\|_\mathrm{HS}^2 \le \left(\left\|a(f)a^\dagger(f)\mathrm{e}^{-\beta H_\mathrm{B}}\right\|_\mathrm{HS} + \left\|a^\dagger(f)\mathrm{e}^{-\beta H_\mathrm{B}}a(f)\right\|_\mathrm{HS}\right)^2$$

$$\le \left(\left\|a(f)a^\dagger(f)\mathrm{e}^{-\beta H_\mathrm{B}}\right\|_\mathrm{HS} + \sqrt{\|a(f)a^\dagger(f)\mathrm{e}^{-\beta H_\mathrm{B}}\|_\mathrm{HS}\,\|a(f)a^\dagger(f)\mathrm{e}^{-\beta H_\mathrm{B}}\|_\mathrm{HS}}\right)^2$$

$$< 4\left\|f\right\|^4 \left[\frac{1}{m^2}\frac{\mathrm{d}^2}{\mathrm{d}(2\beta)^2} - \frac{2}{m}\frac{\mathrm{d}}{\mathrm{d}(2\beta)} + 1\right] Z(2\beta), \tag{143}$$

and similarly for (iii):

$$\left\|\left[a(f), \left[a^\dagger(f), \mathrm{e}^{-\beta H_\mathrm{B}}\right]\right]\right\|_\mathrm{HS}^2$$

$$\le \left(\left\|a(f)a^\dagger(f)\mathrm{e}^{-\beta H_\mathrm{B}}\right\|_\mathrm{HS} + \left\|a(f)\mathrm{e}^{-\beta H_\mathrm{B}}a^\dagger(f)\right\|_\mathrm{HS}\right.$$

$$\left.+ \left\|a^\dagger(f)\mathrm{e}^{-\beta H_\mathrm{B}}a(f)\right\|_\mathrm{HS} + \left\|\mathrm{e}^{-\beta H_\mathrm{B}}a^\dagger(f)a(f)\right\|_\mathrm{HS}\right)^2$$

$$\le \left(\left\|a(f)a^\dagger(f)\mathrm{e}^{-\beta H_\mathrm{B}}\right\|_\mathrm{HS} + \sqrt{\|a^\dagger(f)a(f)\mathrm{e}^{-\beta H_\mathrm{B}}\|_\mathrm{HS}\,\|a^\dagger(f)a(f)\mathrm{e}^{-\beta H_\mathrm{B}}\|_\mathrm{HS}}\right.$$

$$\left.+ \sqrt{\|a(f)a^\dagger(f)\mathrm{e}^{-\beta H_\mathrm{B}}\|_\mathrm{HS}\,\|a(f)a^\dagger(f)\mathrm{e}^{-\beta H_\mathrm{B}}\|_\mathrm{HS}} + \left\|a^\dagger(f)a(f)\mathrm{e}^{-\beta H_\mathrm{B}}\right\|_\mathrm{HS}\right)^2$$

$$= 4 \left(\left\|a(f)a^\dagger(f)\mathrm{e}^{-\beta H_\mathrm{B}}\right\|_\mathrm{HS} + \left\|a^\dagger(f)a(f)\mathrm{e}^{-\beta H_\mathrm{B}}\right\|_\mathrm{HS}\right)^2$$

$$< 4\left\|f\right\|^4 \left(\sqrt{\left[\frac{1}{m^2}\frac{\mathrm{d}^2}{\mathrm{d}(2\beta)^2} - \frac{2}{m}\frac{\mathrm{d}}{\mathrm{d}(2\beta)} + 1\right] Z(2\beta)} + \sqrt{\left[\frac{1}{m^2}\frac{\mathrm{d}^2}{\mathrm{d}(2\beta)^2} - \frac{1}{m}\frac{\mathrm{d}}{\mathrm{d}(2\beta)}\right] Z(2\beta)}\right)^2. \tag{144}$$

(iv) reads

$$
\begin{aligned}
\left\| \left[ a^\dagger(f), \left[ a(f), \mathrm{e}^{-\beta H_\mathrm{B}} \right] \right] \right\|_\mathrm{HS}^2 &= \left\| \left( a^\dagger(f) \left[ a(f), \mathrm{e}^{-\beta H_\mathrm{B}} \right] - \left[ a(f), \mathrm{e}^{-\beta H_\mathrm{B}} \right] a^\dagger(f) \right)^\dagger \right\|_\mathrm{HS}^2 \\
&= \left\| - \left[ a^\dagger(f), \mathrm{e}^{-\beta H_\mathrm{B}} \right] a(f) + a(f) \left[ a^\dagger(f), \mathrm{e}^{-\beta H_\mathrm{B}} \right] \right\|_\mathrm{HS}^2 \\
&= \left\| \left[ a(f), \left[ a^\dagger(f), \mathrm{e}^{-\beta H_\mathrm{B}} \right] \right] \right\|_\mathrm{HS}^2 \\
&< 4 \left\| f \right\|^4 \left( \sqrt{ \left[ \frac{1}{m^2} \frac{\mathrm{d}^2}{\mathrm{d}(2\beta)^2} - \frac{2}{m} \frac{\mathrm{d}}{\mathrm{d}(2\beta)} + 1 \right] Z(2\beta) } \right. \\
&\qquad\qquad \left. + \sqrt{ \left[ \frac{1}{m^2} \frac{\mathrm{d}^2}{\mathrm{d}(2\beta)^2} - \frac{1}{m} \frac{\mathrm{d}}{\mathrm{d}(2\beta)} \right] Z(2\beta) } \right)^2 .
\end{aligned}
\tag{145}
$$

This completes the proof. $\qquad\square$

Finally, we can compute the norm of Eq. (115). By using the estimates provided in Lemma C.2 and Lemma C.4, we obtain

$$
\begin{aligned}
\left\| \mathbf{ad}_{U_j H U_j^\dagger} \mathbf{ad}_{U_i H U_i^\dagger} (\rho_S \otimes \rho_B) \right\|_\mathrm{HS}^2 &< \frac{1}{Z(\beta)^2} \left( \left\| \left[ v_j H_\mathrm{S} v_j^\dagger, \left[ v_i H_\mathrm{S} v_i^\dagger, \rho_\mathrm{S} \right] \right] \right\|_\mathrm{HS}^2 \times Z(2\beta) \right. \\
&+ \left\| \left[ v_j B v_j^\dagger, \left[ v_i H_\mathrm{S} v_i^\dagger, \rho_\mathrm{S} \right] \right] + \left[ v_j H_\mathrm{S} v_j^\dagger, \left[ v_i B v_i^\dagger, \rho_\mathrm{S} \right] \right] \right\|_\mathrm{HS}^2 \times \left\| f \right\|^2 \left[ -\frac{1}{m} \frac{\mathrm{d}}{\mathrm{d}(2\beta)} + 1 \right] Z(2\beta) \\
&+ \left\| \left[ v_j B^\dagger v_j^\dagger, \left[ v_i H_\mathrm{S} v_i^\dagger, \rho_\mathrm{S} \right] \right] + \left[ v_j H_\mathrm{S} v_j^\dagger, \left[ v_i B^\dagger v_i^\dagger, \rho_\mathrm{S} \right] \right] \right\|_\mathrm{HS}^2 \times \left( -\frac{\left\| f \right\|^2}{m} \frac{\mathrm{d}}{\mathrm{d}(2\beta)} Z(2\beta) \right) \\
&+ \left\| \left[ v_i H_\mathrm{S} v_i^\dagger, \rho_\mathrm{S} \right] v_j B v_j^\dagger + \left[ v_j H_\mathrm{S} v_j^\dagger, \rho_\mathrm{S} v_i B v_i^\dagger \right] \right\|_\mathrm{HS}^2 \times \left\| f \right\|^2 \left[ -\frac{1}{m} \frac{\mathrm{d}}{\mathrm{d}(2\beta)} + 1 \right] Z(2\beta) \\
&+ \left\| \left[ v_i H_\mathrm{S} v_i^\dagger, \rho_\mathrm{S} \right] v_j B^\dagger v_j^\dagger + \left[ v_j H_\mathrm{S} v_j^\dagger, \rho_\mathrm{S} v_i B^\dagger v_i^\dagger \right] \right\|_\mathrm{HS}^2 \times \left\| f \right\|^2 \left[ -\frac{1}{m} \frac{\mathrm{d}}{\mathrm{d}(2\beta)} + 1 \right] Z(2\beta) \\
&+ \left\| \left[ v_j B v_j^\dagger, \left[ v_i B v_i^\dagger, \rho_\mathrm{S} \right] \right] \right\|_\mathrm{HS}^2 \times \left( \left\| f \right\|^4 \left[ \frac{1}{m^2} \frac{\mathrm{d}^2}{\mathrm{d}(2\beta)^2} - \frac{3}{m} \frac{\mathrm{d}}{\mathrm{d}(2\beta)} + 2 \right] Z(2\beta) \right) \\
&+ \left\| \left[ v_j B^\dagger v_j^\dagger, \left[ v_i B^\dagger v_i^\dagger, \rho_\mathrm{S} \right] \right] \right\|_\mathrm{HS}^2 \times \left( \frac{\left\| f \right\|^4}{m^2} \frac{\mathrm{d}^2}{\mathrm{d}(2\beta)^2} Z(2\beta) \right) \\
&+ \left\| \left[ v_j B^\dagger v_j^\dagger, \left[ v_i B v_i^\dagger, \rho_\mathrm{S} \right] \right] \right\|_\mathrm{HS}^2 \times \left( \left\| f \right\|^4 \left[ \frac{1}{m^2} \frac{\mathrm{d}^2}{\mathrm{d}(2\beta)^2} - \frac{2}{m} \frac{\mathrm{d}}{\mathrm{d}(2\beta)} + 1 \right] Z(2\beta) \right) \\
&+ \left\| \left[ v_j B v_j^\dagger, \left[ v_i B^\dagger v_i^\dagger, \rho_\mathrm{S} \right] \right] \right\|_\mathrm{HS}^2 \times \left( \left\| f \right\|^4 \left[ \frac{1}{m^2} \frac{\mathrm{d}^2}{\mathrm{d}(2\beta)^2} - \frac{1}{m} \frac{\mathrm{d}}{\mathrm{d}(2\beta)} \right] Z(2\beta) \right) \\
&+ \left\| \left[ v_i B v_i^\dagger, \rho_\mathrm{S} \right] v_j B v_j^\dagger \right\|_\mathrm{HS}^2 \times \left( \left\| f \right\|^4 \left[ \frac{1}{m^2} \frac{\mathrm{d}^2}{\mathrm{d}(2\beta)^2} - \frac{3}{m} \frac{\mathrm{d}}{\mathrm{d}(2\beta)} + 2 \right] Z(2\beta) \right) \\
&+ \left\| \left[ v_i B^\dagger v_i^\dagger, \rho_\mathrm{S} \right] v_j B^\dagger v_j^\dagger \right\|_\mathrm{HS}^2 \times \left( \left\| f \right\|^4 \left[ \frac{1}{m^2} \frac{\mathrm{d}^2}{\mathrm{d}(2\beta)^2} - \frac{3}{m} \frac{\mathrm{d}}{\mathrm{d}(2\beta)} + 2 \right] Z(2\beta) \right) \\
&+ \left\| \left[ v_i B v_i^\dagger, \rho_\mathrm{S} \right] v_j B^\dagger v_j^\dagger \right\|_\mathrm{HS}^2 \times \left( 4 \left\| f \right\|^4 \left[ \frac{1}{m^2} \frac{\mathrm{d}^2}{\mathrm{d}(2\beta)^2} - \frac{2}{m} \frac{\mathrm{d}}{\mathrm{d}(2\beta)} + 1 \right] Z(2\beta) \right)
\end{aligned}
$$

$$
+ \left\| \left[ v_i B^\dagger v_i^\dagger, \rho_\mathrm{S} \right] v_j B v_j^\dagger \right\|_{\mathrm{HS}}^2 \times \left( \|f\|^4 \left[ \frac{1}{m^2} \frac{\mathrm{d}^2}{\mathrm{d}(2\beta)^2} - \frac{2}{m} \frac{\mathrm{d}}{\mathrm{d}(2\beta)} + 1 \right] Z(2\beta) \right)
$$

$$
+ \left\| \left[ v_j B v_j^\dagger, \rho_\mathrm{S} v_i B v_i^\dagger \right] \right\|_{\mathrm{HS}}^2 \times \left( \|f\|^4 \left[ \frac{1}{m^2} \frac{\mathrm{d}^2}{\mathrm{d}(2\beta)^2} - \frac{3}{m} \frac{\mathrm{d}}{\mathrm{d}(2\beta)} + 2 \right] Z(2\beta) \right)
$$

$$
+ \left\| \left[ v_j B^\dagger v_j^\dagger, \rho_\mathrm{S} v_i B^\dagger v_i^\dagger \right] \right\|_{\mathrm{HS}}^2 \times \left( \|f\|^4 \left[ \frac{1}{m^2} \frac{\mathrm{d}^2}{\mathrm{d}(2\beta)^2} - \frac{3}{m} \frac{\mathrm{d}}{\mathrm{d}(2\beta)} + 2 \right] Z(2\beta) \right)
$$

$$
+ \left\| \left[ v_j B^\dagger v_j^\dagger, \rho_\mathrm{S} v_i B v_i^\dagger \right] \right\|_{\mathrm{HS}}^2 \times \left( \|f\|^4 \left[ \frac{1}{m^2} \frac{\mathrm{d}^2}{\mathrm{d}(2\beta)^2} - \frac{2}{m} \frac{\mathrm{d}}{\mathrm{d}(2\beta)} + 1 \right] Z(2\beta) \right)
$$

$$
+ \left\| \left[ v_j B v_j^\dagger, \rho_\mathrm{S} v_i B^\dagger v_i^\dagger \right] \right\|_{\mathrm{HS}}^2 \times \|f\|^4 \left( \sqrt{ \left[ \frac{1}{m^2} \frac{\mathrm{d}^2}{\mathrm{d}(2\beta)^2} - \frac{1}{m} \frac{\mathrm{d}}{\mathrm{d}(2\beta)} \right] Z(2\beta) } \right.
$$

$$
\left. + \sqrt{ \left[ \frac{1}{m^2} \frac{\mathrm{d}^2}{\mathrm{d}(2\beta)^2} - \frac{2}{m} \frac{\mathrm{d}}{\mathrm{d}(2\beta)} + 1 \right] Z(2\beta) } \right)^2
$$

$$
+ \left\| \rho_\mathrm{S} v_i B v_i^\dagger v_j B v_j^\dagger \right\|_{\mathrm{HS}}^2 \times 4 \|f\|^4 \left[ \frac{1}{m^2} \frac{\mathrm{d}^2}{\mathrm{d}(2\beta)^2} - \frac{3}{m} \frac{\mathrm{d}}{\mathrm{d}(2\beta)} + 2 \right] Z(2\beta)
$$

$$
+ \left\| \rho_\mathrm{S} v_i B^\dagger v_i^\dagger v_j B^\dagger v_j^\dagger \right\|_{\mathrm{HS}}^2 \times 4 \|f\|^4 \left[ \frac{1}{m^2} \frac{\mathrm{d}^2}{\mathrm{d}(2\beta)^2} - \frac{3}{m} \frac{\mathrm{d}}{\mathrm{d}(2\beta)} + 2 \right] Z(2\beta)
$$

$$
+ \left\| \rho_\mathrm{S} v_i B v_i^\dagger v_j B^\dagger v_j^\dagger \right\|_{\mathrm{HS}}^2 \times 4 \|f\|^4 \left( \sqrt{ \left[ \frac{1}{m^2} \frac{\mathrm{d}^2}{\mathrm{d}(2\beta)^2} - \frac{2}{m} \frac{\mathrm{d}}{\mathrm{d}(2\beta)} + 1 \right] Z(2\beta) } \right.
$$

$$
\left. + \sqrt{ \left[ \frac{1}{m^2} \frac{\mathrm{d}^2}{\mathrm{d}(2\beta)^2} - \frac{1}{m} \frac{\mathrm{d}}{\mathrm{d}(2\beta)} \right] Z(2\beta) } \right)^2
$$

$$
+ \left\| \rho_\mathrm{S} v_i B^\dagger v_i^\dagger v_j B v_j^\dagger \right\|_{\mathrm{HS}}^2 \times 4 \|f\|^4 \left( \sqrt{ \left[ \frac{1}{m^2} \frac{\mathrm{d}^2}{\mathrm{d}(2\beta)^2} - \frac{2}{m} \frac{\mathrm{d}}{\mathrm{d}(2\beta)} + 1 \right] Z(2\beta) } \right.
$$

$$
\left. + \sqrt{ \left[ \frac{1}{m^2} \frac{\mathrm{d}^2}{\mathrm{d}(2\beta)^2} - \frac{1}{m} \frac{\mathrm{d}}{\mathrm{d}(2\beta)} \right] Z(2\beta) } \right)^2
$$

$$
+ \left\| \left[ v_i B v_i^\dagger, \rho_\mathrm{S} \right] \right\|_{\mathrm{HS}}^2 \times \|\omega f\|^2 \left[ -\frac{1}{m} \frac{\mathrm{d}}{\mathrm{d}(2\beta)} + 1 \right] Z(2\beta)
$$

$$
+ \left\| \left[ v_i B^\dagger v_i^\dagger, \rho_\mathrm{S} \right] \right\|_{\mathrm{HS}}^2 \times \left( -\frac{\|\omega f\|^2}{m} \frac{\mathrm{d}}{\mathrm{d}(2\beta)} Z(2\beta) \right)
$$

$$
+ \left\| \rho_\mathrm{S} v_i B v_i^\dagger \right\|_{\mathrm{HS}}^2 \times \|\omega f\|^2 \left[ -\frac{1}{m} \frac{\mathrm{d}}{\mathrm{d}(2\beta)} + 1 \right] Z(2\beta)
$$

$$
+ \left\| \rho_\mathrm{S} v_i B^\dagger v_i^\dagger \right\|_{\mathrm{HS}}^2 \times \|\omega f\|^2 \left[ -\frac{1}{m} \frac{\mathrm{d}}{\mathrm{d}(2\beta)} + 1 \right] Z(2\beta) \right). \tag{146}
$$

These terms only involve up to second-order derivatives in $(2\beta)$ of the partition function $Z(2\beta)$. Since by Proposition B.1, $Z(2\beta)$ is indeed twice differentiable in $(2\beta)$, all the norms are finite. This proves that $\mathbf{ad}_{U_j H U_j^\dagger} \mathbf{ad}_{U_i H U_i^\dagger}(\rho_S \otimes \rho_B) \in \mathcal{L}(\mathcal{H})$ and thus $\rho_S \otimes \rho_B \in \mathrm{Dom}\, \mathbf{ad}_{U_j H U_j^\dagger} \mathbf{ad}_{U_i H U_i^\dagger}$. We summarize this discussion in the following theorem.

**Theorem C.5.** *Consider a Hamiltonian $H$ of the form of Eq. (23) satisfying Assumption 3.1, and let $U_j = v_j \otimes I$, where the set $\{v_j\}_{j=0}^{L-1}$ generates a unitary group that acts irreducibly. Set $\rho = \rho_{\mathrm{S}} \otimes \rho_{\mathrm{B}}$, where $\rho_{\mathrm{S}}$ is an arbitrary qubit density operator and $\rho_{\mathrm{B}}$ is the Gibbs state at inverse temperature $\beta$ associated to $H_{\mathrm{B}}$. Assume $f \in \ell^2$ (Assumption 3.2) and also $\omega f \in \ell^2$ (Assumption 5.1) and define $m := \inf_j \omega_j > 0$. Furthermore, let Assumption 5.2 be satisfied. Then the dynamical decoupling error $\xi_N(t; \rho)$ for pulsing with $\{v_j\}_{j=0}^{L-1}$ can be bounded by*

$$\xi_N(t; \rho) \leq \frac{t^2}{NL^2} \sum_{j=0}^{L-1} \left( \frac{1}{2} \left\| \mathbf{ad}^2_{U_j H U_j^\dagger} \rho \right\| + \sum_{i=0}^{j-1} \left\| \mathbf{ad}_{U_j H U_j^\dagger} \mathbf{ad}_{U_i H U_i^\dagger} \rho \right\| \right), \tag{147}$$

*where the individual terms are bounded by*

$$\left\| \mathbf{ad}_{U_j H U_j^\dagger} \mathbf{ad}_{U_i H U_i^\dagger} (\rho_S \otimes \rho_B) \right\|_{\mathrm{HS}}^2 < \frac{1}{Z(\beta)^2} \left( \left\| \left[ v_j H_{\mathrm{S}} v_j^\dagger, \left[ v_i H_{\mathrm{S}} v_i^\dagger, \rho_{\mathrm{S}} \right] \right] \right\|_{\mathrm{HS}}^2 \times Z(2\beta) \right.$$

$$+ \left( \|f\|^2 \left\| \left[ v_j B^\dagger v_j^\dagger, \left[ v_i H_{\mathrm{S}} v_i^\dagger, \rho_{\mathrm{S}} \right] \right] + \left[ v_j H_{\mathrm{S}} v_j^\dagger, \left[ v_i B^\dagger v_i^\dagger, \rho_{\mathrm{S}} \right] \right] \right\|_{\mathrm{HS}}^2 \right.$$

$$\left. + \|\omega f\|^2 \left\| \left[ v_i B^\dagger v_i^\dagger, \rho_{\mathrm{S}} \right] \right\|_{\mathrm{HS}}^2 \right) \times \left[ -\frac{1}{m} \frac{\mathrm{d}}{\mathrm{d}(2\beta)} \right] Z(2\beta)$$

$$+ \left( \|f\|^2 \left\| \left[ v_j B v_j^\dagger, \left[ v_i H_{\mathrm{S}} v_i^\dagger, \rho_{\mathrm{S}} \right] \right] + \left[ v_j H_{\mathrm{S}} v_j^\dagger, \left[ v_i B v_i^\dagger, \rho_{\mathrm{S}} \right] \right] \right\|_{\mathrm{HS}}^2 \right.$$

$$+ \|f\|^2 \left\| \left[ v_i H_{\mathrm{S}} v_i^\dagger, \rho_{\mathrm{S}} \right] v_j B v_j^\dagger + \left[ v_j H_{\mathrm{S}} v_j^\dagger, \rho_{\mathrm{S}} v_i B v_i^\dagger \right] \right\|_{\mathrm{HS}}^2$$

$$+ \|f\|^2 \left\| \left[ v_i H_{\mathrm{S}} v_i^\dagger, \rho_{\mathrm{S}} \right] v_j B^\dagger v_j^\dagger + \left[ v_j H_{\mathrm{S}} v_j^\dagger, \rho_{\mathrm{S}} v_i B^\dagger v_i^\dagger \right] \right\|_{\mathrm{HS}}^2$$

$$+ \|\omega f\|^2 \left\| \left[ v_i B v_i^\dagger, \rho_{\mathrm{S}} \right] \right\|_{\mathrm{HS}}^2 + \|\omega f\|^2 \left\| \rho_{\mathrm{S}} v_i B v_i^\dagger \right\|_{\mathrm{HS}}^2$$

$$\left. + \|\omega f\|^2 \left\| \rho_{\mathrm{S}} v_i B^\dagger v_i^\dagger \right\|_{\mathrm{HS}}^2 \right) \times \left[ -\frac{1}{m} \frac{\mathrm{d}}{\mathrm{d}(2\beta)} + 1 \right] Z(2\beta)$$

$$+ \left\| \left[ v_j B^\dagger v_j^\dagger, \left[ v_i B^\dagger v_i^\dagger, \rho_{\mathrm{S}} \right] \right] \right\|_{\mathrm{HS}}^2 \times \frac{\|f\|^4}{m^2} \frac{\mathrm{d}^2}{\mathrm{d}(2\beta)^2} Z(2\beta)$$

$$+ \left\| \left[ v_j B v_j^\dagger, \left[ v_i B^\dagger v_i^\dagger, \rho_{\mathrm{S}} \right] \right] \right\|_{\mathrm{HS}}^2 \times \|f\|^4 \left[ \frac{1}{m^2} \frac{\mathrm{d}^2}{\mathrm{d}(2\beta)^2} - \frac{1}{m} \frac{\mathrm{d}}{\mathrm{d}(2\beta)} \right] Z(2\beta)$$

$$+ \left( \left\| \left[ v_j B^\dagger v_j^\dagger, \left[ v_i B v_i^\dagger, \rho_{\mathrm{S}} \right] \right] \right\|_{\mathrm{HS}}^2 + 4 \left\| \left[ v_i B v_i^\dagger, \rho_{\mathrm{S}} \right] v_j B^\dagger v_j^\dagger \right\|_{\mathrm{HS}}^2 \right.$$

$$\left. + \left\| \left[ v_i B^\dagger v_i^\dagger, \rho_{\mathrm{S}} \right] v_j B v_j^\dagger \right\|_{\mathrm{HS}}^2 + \left\| \left[ v_j B^\dagger v_j^\dagger, \rho_{\mathrm{S}} v_i B v_i^\dagger \right] \right\|_{\mathrm{HS}}^2 \right)$$

$$\times \|f\|^4 \left[ \frac{1}{m^2} \frac{\mathrm{d}^2}{\mathrm{d}(2\beta)^2} - \frac{2}{m} \frac{\mathrm{d}}{\mathrm{d}(2\beta)} + 1 \right] Z(2\beta)$$

$$+ \left( \left\| \left[ v_j B v_j^\dagger, \left[ v_i B v_i^\dagger, \rho_{\mathrm{S}} \right] \right] \right\|_{\mathrm{HS}}^2 + \left\| \left[ v_i B v_i^\dagger, \rho_{\mathrm{S}} \right] v_j B v_j^\dagger \right\|_{\mathrm{HS}}^2 \right.$$

$$+ \left\| \left[ v_i B^\dagger v_i^\dagger, \rho_{\mathrm{S}} \right] v_j B^\dagger v_j^\dagger \right\|_{\mathrm{HS}}^2 + \left\| \left[ v_j B v_j^\dagger, \rho_{\mathrm{S}} v_i B v_i^\dagger \right] \right\|_{\mathrm{HS}}^2$$

$$+ \left\| \left[ v_j B^\dagger v_j^\dagger, \rho_{\mathrm{S}} v_i B^\dagger v_i^\dagger \right] \right\|_{\mathrm{HS}}^2 + 4 \left\| \rho_{\mathrm{S}} v_i B v_i^\dagger v_j B v_j^\dagger \right\|_{\mathrm{HS}}^2$$

$$\left. + 4 \left\| \rho_{\mathrm{S}} v_i B^\dagger v_i^\dagger v_j B^\dagger v_j^\dagger \right\|_{\mathrm{HS}}^2 \right) \times \|f\|^4 \left[ \frac{1}{m^2} \frac{\mathrm{d}^2}{\mathrm{d}(2\beta)^2} - \frac{3}{m} \frac{\mathrm{d}}{\mathrm{d}(2\beta)} + 2 \right] Z(2\beta)$$

$$+ \left( \left\| \left[ v_j B v_j^\dagger, \rho_S v_i B^\dagger v_i^\dagger \right] \right\|_{\mathrm{HS}}^2 + 4 \left\| \rho_S v_i B v_i^\dagger v_j B^\dagger v_j^\dagger \right\|_{\mathrm{HS}}^2 + 4 \left\| \rho_S v_i B^\dagger v_i^\dagger v_j B v_j^\dagger \right\|_{\mathrm{HS}}^2 \right)$$

$$\times \|f\|^4 \left( \sqrt{ \left[ \frac{1}{m^2} \frac{\mathrm{d}^2}{\mathrm{d}(2\beta)^2} - \frac{1}{m} \frac{\mathrm{d}}{\mathrm{d}(2\beta)} \right] Z(2\beta) } \right.$$

$$\left. + \sqrt{ \left[ \frac{1}{m^2} \frac{\mathrm{d}^2}{\mathrm{d}(2\beta)^2} - \frac{2}{m} \frac{\mathrm{d}}{\mathrm{d}(2\beta)} + 1 \right] Z(2\beta) } \right)^2 \tag{148}$$

*Proof.* Eq. (147) is a direct consequence of Corollary 4.3 from the main text by setting $H_j = U_j H U_j^\dagger$ and using the following triangle inequality

$$\left\| \sum_{i=0}^{j-1} \mathbf{ad}_{U_j H U_j^\dagger} \mathbf{ad}_{U_i H U_i^\dagger} \rho \right\| \leq \sum_{i=0}^{j-1} \left\| \mathbf{ad}_{U_j H U_j^\dagger} \mathbf{ad}_{U_i H U_i^\dagger} \rho \right\|. \tag{149}$$

Eq. (148) follows from inserting Lemma C.2 and Lemma C.4 into Eq. (115). This leads to Eq. (146), which can be rearranged to give Eq. (148). Our Assumptions ensure that all quantities appearing in the bound are well-defined so the bound stays finite. $\square$

**Corollary C.6** (Loose bound). *Under the same assumptions and notation as in Theorem C.5, Eq. (148) can be further bounded by*

$$\left\| \mathbf{ad}_{U_j H U_j^\dagger} \mathbf{ad}_{U_i H U_i^\dagger} (\rho_S \otimes \rho_B) \right\|_{\mathrm{HS}}^2 < \frac{16}{Z(\beta)^2} \max\{\|H_S\|_{\mathrm{HS}}^4, \|B\|_{\mathrm{HS}}^4\} \times \left( Z(2\beta) \right.$$

$$+ 4\left(\|f\|^2 + \|\omega f\|^2\right) \left[ -\frac{1}{m} \frac{\mathrm{d}}{\mathrm{d}(2\beta)} + 1 \right] Z(2\beta)$$

$$\left. + 58\|f\|^4 \left[ \frac{1}{m^2} \frac{\mathrm{d}^2}{\mathrm{d}(2\beta)^2} - \frac{3}{m} \frac{\mathrm{d}}{\mathrm{d}(2\beta)} + 2 \right] Z(2\beta) \right). \tag{150}$$

*Proof.* All quantities from the system part involving $H_S, B$ and $\rho_S$ can be bounded by using the triangle inequality as well as unitary equivalence and submultiplicativity of the Hilbert–Schmidt norm. Furthermore, $\|\rho_S\|_{\mathrm{HS}} \leq 1$, so that each norm on the system is upper bounded by $4\max\{\|H_S\|_{\mathrm{HS}}^2, \|B\|_{\mathrm{HS}}^2\}$. Squaring this quantity yields the part of the loose bound due to the system.

For the norms on the environmental part, we use $\left[ -\frac{1}{m} \frac{\mathrm{d}}{\mathrm{d}(2\beta)} \right] Z(2\beta) \leq \left[ -\frac{1}{m} \frac{\mathrm{d}}{\mathrm{d}(2\beta)} + 1 \right] Z(2\beta)$ and $\left[ \frac{1}{m^2} \frac{\mathrm{d}^2}{\mathrm{d}(2\beta)^2} - \frac{a}{m} \frac{\mathrm{d}}{\mathrm{d}(2\beta)} + b \right] Z(2\beta) \leq \left[ \frac{1}{m^2} \frac{\mathrm{d}^2}{\mathrm{d}(2\beta)^2} - \frac{3}{m} \frac{\mathrm{d}}{\mathrm{d}(2\beta)} + 2 \right] Z(2\beta)$ for $a \leq 3$ and $b \leq 2$. Furthermore, we bound

$$\left( \sqrt{ \left[ \frac{1}{m^2} \frac{\mathrm{d}^2}{\mathrm{d}(2\beta)^2} - \frac{1}{m} \frac{\mathrm{d}}{\mathrm{d}(2\beta)} \right] Z(2\beta) } + \sqrt{ \left[ \frac{1}{m^2} \frac{\mathrm{d}^2}{\mathrm{d}(2\beta)^2} - \frac{2}{m} \frac{\mathrm{d}}{\mathrm{d}(2\beta)} + 1 \right] Z(2\beta) } \right)^2$$

$$< 4 \left[ \frac{1}{m^2} \frac{\mathrm{d}^2}{\mathrm{d}(2\beta)^2} - \frac{3}{m} \frac{\mathrm{d}}{\mathrm{d}(2\beta)} + 2 \right] Z(2\beta), \tag{151}$$

thus completing the proof. $\square$

This is precisely the looser bound presented in Theorem 5.3 in the main text.

## C.2 Reduction to the single-mode case

In the case of a single mode, the calculations presented above simplify; moreover, tighter bounds can be obtained. Since we consider three single-mode examples in the main part of the paper, it is worthwhile considering this case separately. For a single-mode boson bath, i.e. $H_B = \omega a^\dagger a$, we have $Z(\beta) = \left(1 - e^{-\beta\omega}\right)^{-1}$ and $m = \omega$. Furthermore, without loss of generality we can set $|f| = 1$.

We will reduce the single-mode case from the general case. Specifically, we will reprise the calculations from Lemma C.2 up to the point where we only used equalities or non-strict inequalities, and continue calculating the single-mode norms from there. Instead, we do not use the steps that involve strict inequalities and use tighter estimates. This yields the following:

$$
\begin{aligned}
\|\rho_B\|_{\text{HS}}^2 &= \left(1 - e^{-\beta\omega}\right)^2 Z(2\beta) \\
&= \left(1 - e^{-\beta\omega}\right)^2 \left(1 - e^{-2\beta\omega}\right)^{-1} \\
&= \tanh\left(\frac{\beta\omega}{2}\right),
\end{aligned}
\tag{152}
$$

$$
\begin{aligned}
\|aa\rho_B\|_{\text{HS}}^2 &\leq \left(1 - e^{-\beta\omega}\right)^2 \|f\|^4 \sum_n n\,(n-1)\,e^{-2\beta\omega n} \\
&= \left(1 - e^{-\beta\omega}\right)^2 \sum_n \left(\frac{1}{\omega}n^2\omega^2 - \frac{1}{\omega}n\omega\right) e^{-2\beta\omega n} \\
&= \left(1 - e^{-\beta\omega}\right)^2 \left[\frac{1}{\omega^2}\frac{d^2}{d(2\beta)^2} + \frac{1}{\omega}\frac{d}{d(2\beta)}\right] Z(2\beta) \\
&= \frac{2}{\left(e^{\beta\omega} - 1\right)\left(e^{\beta\omega} + 1\right)^3},
\end{aligned}
\tag{153}
$$

$$
\begin{aligned}
\left\|a^\dagger a^\dagger \rho_B\right\|_{\text{HS}}^2 &= \left(1 - e^{-\beta\omega}\right)^2 \sum_n e^{-2\beta n\omega}\,(n+1)\,(n+2) \\
&= \left(1 - e^{-\beta\omega}\right)^2 \sum_n \left(\frac{1}{\omega^2}n^2\omega^2 + \frac{3}{\omega}n\omega + 2\right) e^{-2\beta\omega n} \\
&= \left(1 - e^{-\beta\omega}\right)^2 \left[\frac{1}{\omega^2}\frac{d^2}{d(2\beta)^2} - \frac{3}{\omega}\frac{d}{d(2\beta)} + 2\right] Z(2\beta) \\
&= \frac{2e^{4\beta\Omega}}{\left(e^{\beta\Omega} - 1\right)\left(e^{\beta\Omega} + 1\right)^3},
\end{aligned}
\tag{154}
$$

$$\left\| aa^\dagger \rho_{\mathrm{B}} \right\|_{\mathrm{HS}}^2 = \left(1 - \mathrm{e}^{-\beta\omega}\right)^2 \sum_n \left\| \mathrm{e}^{-\beta n \omega} \left[ |f|^2 (n+1) |n\rangle \right] \right\|$$

$$= \left(1 - \mathrm{e}^{-\beta\omega}\right)^2 \sum_n \left( \frac{1}{\omega^2} n^2 \omega^2 + \frac{2}{\omega} n\omega + 1 \right)^2 \mathrm{e}^{-2\beta\omega n}$$

$$= \left(1 - \mathrm{e}^{-\beta\omega}\right)^2 \left[ \frac{1}{\omega^2} \frac{\mathrm{d}^2}{\mathrm{d}(2\beta)^2} - \frac{2}{\omega} \frac{\mathrm{d}}{\mathrm{d}(2\beta)} + 1 \right] Z(2\beta)$$

$$= \frac{\mathrm{e}^{2\beta\omega} \left(1 + \mathrm{e}^{2\beta\omega}\right)}{\left(\mathrm{e}^{\beta\omega} - 1\right) \left(\mathrm{e}^{\beta\omega} + 1\right)^3}, \tag{155}$$

$$\left\| \left[ a^\dagger, \rho_{\mathrm{B}} \right] \right\|_{\mathrm{HS}}^2 = \left(1 - \mathrm{e}^{-\beta\omega}\right)^2 \sum_n \left\| f\sqrt{n+1} \mathrm{e}^{-\beta\omega n} \left(1 - \mathrm{e}^{-\beta\omega}\right) |n+1\rangle \right\|^2$$

$$= \left(1 - \mathrm{e}^{-\beta\omega}\right)^4 \sum_n (n+1) \, \mathrm{e}^{-2\beta\omega n}$$

$$= \left(1 - \mathrm{e}^{-\beta\omega}\right)^4 \left[ -\frac{1}{\omega} \frac{\mathrm{d}}{\mathrm{d}(2\beta)} Z(2\beta) + Z(2\beta) \right]$$

$$= \left(1 - \mathrm{e}^{-\beta\omega}\right)^4 \left[ -\frac{1}{\omega} \frac{\mathrm{d}}{\mathrm{d}(2\beta)} \left(1 - \mathrm{e}^{-2\beta\omega}\right)^{-1} + \left(1 - \mathrm{e}^{-2\beta\omega}\right)^{-1} \right]$$

$$= \tanh\left( \frac{\beta\omega}{2} \right), \tag{156}$$

$$\left\| \left[ a(f), \mathrm{e}^{-\beta H_{\mathrm{B}}} \right] \right\|_{\mathrm{HS}}^2 = \left\| - \left[ a^\dagger(f), \mathrm{e}^{-\beta H_{\mathrm{B}}} \right] \right\|_{\mathrm{HS}}^2$$

$$= \tanh\left( \frac{\beta\omega}{2} \right), \tag{157}$$

$$\left\| a^\dagger \left[ a^\dagger, \rho_{\mathrm{B}} \right] \right\|_{\mathrm{HS}}^2 = \left(1 - \mathrm{e}^{-\beta\omega}\right)^2 \sum_n \left\| f^2 \sqrt{n+1} \sqrt{n+2} \, \mathrm{e}^{-\beta\omega n} \left(1 - \mathrm{e}^{-\beta\omega}\right) |n+2\rangle \right\|^2$$

$$= \left(1 - \mathrm{e}^{-\beta\omega}\right)^4 \sum_n \left(n^2 + 3n + 2\right) \mathrm{e}^{-2\beta\omega n}$$

$$= \left(1 - \mathrm{e}^{-\beta\omega}\right)^4 \left[ \frac{1}{\omega^2} \frac{\mathrm{d}^2}{\mathrm{d}(2\beta)^2} Z(2\beta) - \frac{3}{\omega} \frac{\mathrm{d}}{\mathrm{d}(2\beta)} Z(2\beta) + 2 Z(2\beta) \right]$$

$$= \left(1 - \mathrm{e}^{-\beta\omega}\right)^4 \left[ \frac{1}{\omega^2} \frac{\mathrm{d}^2}{\mathrm{d}(2\beta)^2} \left(1 - \mathrm{e}^{-2\beta\omega}\right)^{-1} - \frac{3}{\omega} \frac{\mathrm{d}}{\mathrm{d}(2\beta)} \left(1 - \mathrm{e}^{-2\beta\omega}\right)^{-1} \right.$$

$$\left. + 2 \left(1 - \mathrm{e}^{-2\beta\omega}\right)^{-1} \right]$$

$$= \frac{2\mathrm{e}^{2\beta\omega} \left(\mathrm{e}^{\beta\omega} - 1\right)}{\left(\mathrm{e}^{\beta\omega} + 1\right)^3}, \tag{158}$$

$$
\left\| a \left[ a, \mathrm{e}^{-\beta H_{\mathrm{B}}} \right] \right\|_{\mathrm{HS}}^2 = \left( 1 - \mathrm{e}^{-\beta\omega} \right)^2 \sum_n \left\| f^2 \sqrt{n+1}\sqrt{n+2}\,\mathrm{e}^{-\beta n\omega} \left( \mathrm{e}^{-\beta\omega} - \mathrm{e}^{-2\beta\omega} \right) |n+2\rangle \right\|^2
$$

$$
= \left( 1 - \mathrm{e}^{-\beta\omega} \right)^4 \mathrm{e}^{-2\beta\omega} \sum_n \left( n^2 + 3n + 2 \right) \mathrm{e}^{-2\beta\omega n}
$$

$$
= \mathrm{e}^{-2\beta\omega} \left\| a^\dagger \left[ a^\dagger, \rho_{\mathrm{B}} \right] \right\|_{\mathrm{HS}}^2
$$

$$
= \frac{2 \left( \mathrm{e}^{\beta\omega} - 1 \right)}{\left( \mathrm{e}^{\beta\omega} + 1 \right)^3}, \tag{159}
$$

$$
\left\| a \left[ a^\dagger, \rho_{\mathrm{B}} \right] \right\|_{\mathrm{HS}}^2 = \left( 1 - \mathrm{e}^{-\beta\omega} \right)^2 \sum_n \left\| |f|^2 \left( n+1 \right) \mathrm{e}^{-\beta\omega n} \left( 1 - \mathrm{e}^{-\beta\omega} \right) |n\rangle \right\|^2
$$

$$
= \left( 1 - \mathrm{e}^{-\beta\omega} \right)^4 \sum_n \left( n^2 + 2n + 1 \right) \mathrm{e}^{-2\beta\omega n}
$$

$$
= \left( 1 - \mathrm{e}^{-\beta\omega} \right)^4 \left[ \frac{1}{\omega^2} \frac{\mathrm{d}^2}{\mathrm{d}(2\beta)^2} Z(2\beta) - \frac{2}{\omega} \frac{\mathrm{d}}{\mathrm{d}(2\beta)} Z(2\beta) + Z(2\beta) \right]
$$

$$
= \left( 1 - \mathrm{e}^{-\beta\omega} \right)^4 \left[ \frac{1}{\omega^2} \frac{\mathrm{d}^2}{\mathrm{d}(2\beta)^2} \left( 1 - \mathrm{e}^{-2\beta\omega} \right)^{-1} - \frac{2}{\omega} \frac{\mathrm{d}}{\mathrm{d}(2\beta)} \left( 1 - \mathrm{e}^{-2\beta\omega} \right)^{-1} \right.
$$

$$
\left. + \left( 1 - \mathrm{e}^{-2\beta\omega} \right)^{-1} \right]
$$

$$
= 4 \sinh^4 \left( \frac{\beta\omega}{2} \right) \coth(\beta\omega) \mathrm{csch}^2(\beta\omega), \tag{160}
$$

$$
\left\| \left[ a^\dagger, a\rho_{\mathrm{B}} \right] \right\|_{\mathrm{HS}}^2 = \left( 1 - \mathrm{e}^{-\beta\omega} \right)^2 \sum_n \left\| |f|^2 n\mathrm{e}^{-\beta\omega n} |n\rangle - |f|^2 \left( n+1 \right) \mathrm{e}^{-\beta\omega n} \mathrm{e}^{-\beta\omega} |n\rangle \right\|^2
$$

$$
= \left( 1 - \mathrm{e}^{-\beta\omega} \right)^2 \sum_n \left\| \left( n \left( 1 - \mathrm{e}^{-\beta\omega} \right) - \mathrm{e}^{-\beta\omega} \right) \mathrm{e}^{-\beta\omega n} |n\rangle \right\|^2
$$

$$
= \left( 1 - \mathrm{e}^{-\beta\omega} \right)^4 \frac{1}{\omega^2} \frac{\mathrm{d}^2}{\mathrm{d}(2\beta)^2} Z(2\beta) + 2 \left( 1 - \mathrm{e}^{-\beta\omega} \right)^3 \mathrm{e}^{-\beta\omega} \frac{1}{\omega} \frac{\mathrm{d}}{\mathrm{d}(2\beta)} Z(2\beta)
$$

$$
+ \left( 1 - \mathrm{e}^{-\beta\omega} \right)^2 \mathrm{e}^{-2\beta\omega} Z(2\beta)
$$

$$
= \left[ \left( 1 - \mathrm{e}^{-\beta\omega} \right)^4 \frac{1}{\omega^2} \frac{\mathrm{d}^2}{\mathrm{d}(2\beta)^2} + 2 \left( 1 - \mathrm{e}^{-\beta\omega} \right)^3 \mathrm{e}^{-\beta\omega} \frac{1}{\omega} \frac{\mathrm{d}}{\mathrm{d}(2\beta)} \right.
$$

$$
\left. + \left( 1 - \mathrm{e}^{-\beta\omega} \right)^2 \mathrm{e}^{-2\beta\omega} \right] \left( 1 - \mathrm{e}^{-2\beta\omega} \right)^{-1}
$$

$$
= \frac{2 \left( \mathrm{e}^{\beta\omega} - 1 \right)}{\left( \mathrm{e}^{\beta\omega} + 1 \right)^3}, \tag{161}
$$

$$\left\| \left[ a^\dagger, \left[ a^\dagger, \rho_\mathrm{B} \right] \right] \right\|_\mathrm{HS}^2 = \left( 1 - \mathrm{e}^{-\beta\omega} \right)^2 \sum_n \left\| f^2 \sqrt{n+1}\sqrt{n+2}\,\mathrm{e}^{-\beta\omega n} \right.$$

$$\left. \times \left( 1 - 2\mathrm{e}^{-\beta\omega} + \mathrm{e}^{-\beta\omega}\mathrm{e}^{-\beta\omega} \right) |n+2\rangle \right\|^2$$

$$= \left( 1 - \mathrm{e}^{-\beta\omega} \right)^6 \sum_n \left( n^2 + 3n + 2 \right) \mathrm{e}^{-2\beta\omega n}$$

$$= \left( 1 - \mathrm{e}^{-\beta\omega} \right)^2 \left\| a^\dagger \left[ a^\dagger, \rho_\mathrm{B} \right] \right\|_\mathrm{HS}^2$$

$$= 2 \tanh^3 \left( \frac{\beta\omega}{2} \right), \tag{162}$$

$$\left\| [a, [a, \rho_\mathrm{B}]] \right\|_\mathrm{HS}^2 = \left\| \left[ a^\dagger, \left[ a^\dagger, \rho_\mathrm{B} \right] \right] \right\|_\mathrm{HS}^2$$

$$= 2 \tanh^3 \left( \frac{\beta\omega}{2} \right), \tag{163}$$

$$\left\| \left[ a^\dagger a, a\rho_\mathrm{B} \right] \right\|_\mathrm{HS}^2 = \left\| \left[ \frac{1}{\omega} H_\mathrm{B}, a \left( 1 - \mathrm{e}^{-\beta\omega} \right) \mathrm{e}^{-\beta H_\mathrm{B}} \right] \right\|_\mathrm{HS}^2$$

$$= -\frac{1}{\omega^2} \left( 1 - \mathrm{e}^{-\beta\omega} \right)^2 \frac{\|\omega f\|^2}{m} \frac{\mathrm{d}}{\mathrm{d}(2\beta)} Z(2\beta)$$

$$= -\left( 1 - \mathrm{e}^{-\beta\omega} \right)^2 \frac{1}{\omega} \frac{\mathrm{d}}{\mathrm{d}(2\beta)} Z(2\beta)$$

$$= \frac{1}{\left( \mathrm{e}^{\beta\omega} + 1 \right)^2}, \tag{164}$$

$$\left\| \left[ a^\dagger a, a^\dagger \rho_\mathrm{B} \right] \right\|_\mathrm{HS}^2 = \left\| \left[ \frac{1}{\omega} H_\mathrm{B}, a^\dagger(f) \left( 1 - \mathrm{e}^{-\beta\omega} \right) \mathrm{e}^{-\beta H_\mathrm{B}} \right] \right\|_\mathrm{HS}^2$$

$$= -\frac{1}{\omega^2} \left( 1 - \mathrm{e}^{-\beta\omega} \right)^2 \frac{\|\omega f\|^2}{m} \frac{\mathrm{d}}{\mathrm{d}(2\beta)} Z(2\beta) + \frac{1}{\omega^2} \left( 1 - \mathrm{e}^{-\beta\omega} \right)^2 \|\omega f\|^2 Z(2\beta)$$

$$= -\left( 1 - \mathrm{e}^{-\beta\omega} \right)^2 \frac{1}{\omega} \frac{\mathrm{d}}{\mathrm{d}(2\beta)} \left( 1 - \mathrm{e}^{-2\beta\omega} \right)^{-1} + \left( 1 - \mathrm{e}^{-\beta\omega} \right)^2 \left( 1 - \mathrm{e}^{-2\beta\omega} \right)^{-1}$$

$$= \frac{\mathrm{e}^{2\beta\omega}}{\left( \mathrm{e}^{\beta\omega} + 1 \right)^2}, \tag{165}$$

$$
\begin{aligned}
\left\| \left[ a^\dagger a, \left[ a^\dagger, \rho_B \right] \right] \right\|_{HS}^2 &= \left\| \left[ \frac{1}{\omega} H_B, \left[ a^\dagger(f), \left( 1 - e^{-\beta\omega} \right) e^{-\beta H_B} \right] \right] \right\|_{HS}^2 \\
&= \frac{1}{\omega^2} \left( 1 - e^{-\beta\omega} \right)^2 \sum_n \left\| \omega f \sqrt{n+1} e^{-\beta n \omega} \left[ 1 - e^{-\beta\omega} \right] |n+1\rangle \right\| \\
&= - \left( 1 - e^{-\beta\omega} \right)^4 \frac{1}{\omega} \frac{d}{d(2\beta)} Z(2\beta) + \left( 1 - e^{-\beta\omega} \right)^4 Z(2\beta) \\
&= - \left( 1 - e^{-\beta\omega} \right)^4 \frac{1}{\omega} \frac{d}{d(2\beta)} \left( 1 - e^{-2\beta\omega} \right)^{-1} + \left( 1 - e^{-\beta\omega} \right)^4 \left( 1 - e^{-2\beta\omega} \right)^{-1} \\
&= \tanh^2 \left( \frac{\beta\omega}{2} \right),
\end{aligned}
\tag{166}
$$

$$
\begin{aligned}
\left\| \left[ a^\dagger a, [a, \rho_B] \right] \right\|_{HS}^2 &= \left\| \left[ a^\dagger a, \left[ a^\dagger, \rho_B \right] \right] \right\|_{HS}^2 \\
&= \tanh^2 \left( \frac{\beta\omega}{2} \right).
\end{aligned}
\tag{167}
$$

The terms involving an annihilation operator $a$ on the right must be computed individually, as in the general case (cf. Lemma C.4) we computed them using a triangle inequality in the arbitrary mode case (which is not tight). This gives the following:

$$
\begin{aligned}
\left\| a^\dagger [a, \rho_B] \right\|_{HS}^2 &= \sum_n \left\| a^\dagger [a, \rho_B] |n\rangle \right\|^2 \\
&= \left( 1 - e^{-\beta\omega} \right)^2 \sum_n \left\| n e^{-\beta\omega n} \left( 1 - e^{+\beta\omega} \right) |n\rangle \right\|^2 \\
&= \left( 1 - e^{-\beta\omega} \right)^2 \left( 1 - e^{+\beta\omega} \right)^2 \left[ \frac{1}{\omega^2} \frac{d^2}{d(2\beta)^2} Z(2\beta) \right] \\
&= 16 \sinh^4 \left( \frac{\beta\omega}{2} \right) \left[ \frac{1}{\omega^2} \frac{d^2}{d(2\beta)^2} \left( 1 - e^{-2\beta\omega} \right)^{-1} \right] \\
&= 4 \sinh^4 \left( \frac{\beta\omega}{2} \right) \coth(\beta\omega) \operatorname{csch}^2(\beta\omega),
\end{aligned}
\tag{168}
$$

$$\left\| \left[ a, a^\dagger \rho_{\mathrm{B}} \right] \right\|_{\mathrm{HS}}^2 = \sum_n \left\| \left[ a, a^\dagger \rho_{\mathrm{B}} \right] |n\rangle \right\|^2$$

$$= \left( 1 - \mathrm{e}^{-\beta\omega} \right)^2 \sum_n \left\| (n+1) \, \mathrm{e}^{-\beta\omega n} |n\rangle - n \mathrm{e}^{-\beta\omega(n-1)} |n\rangle \right\|^2$$

$$= \left( 1 - \mathrm{e}^{-\beta\omega} \right)^2 \sum_n \left\| \left[ n \left( 1 - \mathrm{e}^{+\beta\omega} \right) \mathrm{e}^{-\beta\omega n} + \mathrm{e}^{-\beta\omega n} \right] |n\rangle \right\|^2$$

$$= \left( 1 - \mathrm{e}^{-\beta\omega} \right)^2 \left[ \left( 1 - \mathrm{e}^{+\beta\omega} \right)^2 \frac{1}{\omega^2} \frac{\mathrm{d}^2}{\mathrm{d}(2\beta)^2} Z(2\beta) \right.$$

$$\left. - 2 \left( 1 - \mathrm{e}^{+\beta\omega} \right) \frac{1}{\omega} \frac{\mathrm{d}}{\mathrm{d}(2\beta)} Z(2\beta) + Z(2\beta) \right]$$

$$= \left( 1 - \mathrm{e}^{-\beta\omega} \right)^2 \left[ \left( 1 - \mathrm{e}^{+\beta\omega} \right)^2 \frac{1}{\omega^2} \frac{\mathrm{d}^2}{\mathrm{d}(2\beta)^2} - 2 \left( 1 - \mathrm{e}^{+\beta\omega} \right) \frac{1}{\omega} \frac{\mathrm{d}}{\mathrm{d}(2\beta)} + 1 \right]$$

$$\times \left( 1 - \mathrm{e}^{-2\beta\omega} \right)^{-1}$$

$$= \frac{2\mathrm{e}^{2\beta\omega} \left( \mathrm{e}^{\beta\omega} - 1 \right)}{\left( \mathrm{e}^{\beta\omega} + 1 \right)^3}, \tag{169}$$

$$\left\| \left[ a, \left[ a^\dagger, \rho_{\mathrm{B}} \right] \right] \right\|_{\mathrm{HS}}^2 = \sum_n \left\| a \left[ a^\dagger, \rho_{\mathrm{B}} \right] |n\rangle - \left[ a^\dagger, \rho_{\mathrm{B}} \right] a |n\rangle \right\|^2$$

$$= \sum_n \left\| \left( 1 - \mathrm{e}^{-\beta\omega} \right) (n+1) \, \mathrm{e}^{-\beta\omega n} \left( 1 - \mathrm{e}^{-\beta\omega} \right) |n\rangle - \sqrt{n} \left( a^\dagger \rho_{\mathrm{B}} - \rho_{\mathrm{B}} a^\dagger \right) |n-1\rangle \right\|^2$$

$$= \left( 1 - \mathrm{e}^{-\beta\omega} \right)^2 \sum_n \left\| (n+1) \, \mathrm{e}^{-\beta\omega n} \left( 1 - \mathrm{e}^{-\beta\omega} \right) |n\rangle + n \mathrm{e}^{-\beta\omega n} \left( 1 - \mathrm{e}^{+\beta\omega} \right) |n\rangle \right\|^2$$

$$= \left( 1 - \mathrm{e}^{-\beta\omega} \right)^2 \sum_n \left\| \left[ n \left( 2 - \mathrm{e}^{-\beta\omega} - \mathrm{e}^{+\beta\omega} \right) \mathrm{e}^{-\beta n \omega} + \mathrm{e}^{-\beta n \omega} \left( 1 - \mathrm{e}^{-\beta\omega} \right) \right] |n\rangle \right\|^2$$

$$= \left( 1 - \mathrm{e}^{-\beta\omega} \right)^2 \left[ \frac{4}{\omega^2} \left( 1 - \cosh\left(\beta\omega\right) \right)^2 \frac{\mathrm{d}^2}{\mathrm{d}(2\beta)^2} Z(2\beta) \right.$$

$$\left. - \frac{4}{\omega} \left( 1 - \cosh\left(\beta\omega\right) \right) \left( 1 - \mathrm{e}^{-\beta\omega} \right) \frac{\mathrm{d}}{\mathrm{d}(2\beta)} Z(2\beta) + \left( 1 - \mathrm{e}^{-\beta\omega} \right)^2 Z(2\beta) \right]$$

$$= \left( 1 - \mathrm{e}^{-\beta\omega} \right)^2 \left[ \frac{1}{\omega^2} \left( 1 - \cosh\left(\beta\omega\right) \right)^2 \frac{\mathrm{d}^2}{\mathrm{d}\beta^2} + \frac{1}{\omega} e^{-2\beta\omega} \left( e^{\beta\omega} - 1 \right)^3 \frac{\mathrm{d}}{\mathrm{d}\beta} + \left( 1 - \mathrm{e}^{-\beta\omega} \right)^2 \right]$$

$$\times \left( 1 - \mathrm{e}^{-2\beta\omega} \right)^{-1}$$

$$= 2 \tanh^3 \left( \frac{\beta\omega}{2} \right), \tag{170}$$

$$\left\| \left[ a^\dagger, \left[ a, \rho_{\mathrm{B}} \right] \right] \right\|_{\mathrm{HS}}^2 = \left\| \left[ a, \left[ a^\dagger, \rho_{\mathrm{B}} \right] \right] \right\|_{\mathrm{HS}}^2$$

$$= 2 \tanh^3 \left( \frac{\beta\omega}{2} \right), \tag{171}$$

$$\left\| \left[ a^\dagger, a^\dagger \rho_{\mathrm{B}} \right] \right\|_{\mathrm{HS}}^2 = \left\| a^\dagger \left[ a^\dagger, \rho_{\mathrm{B}} \right] \right\|_{\mathrm{HS}}^2$$
$$= \frac{2 \mathrm{e}^{2\beta\omega} \left( \mathrm{e}^{\beta\omega} - 1 \right)}{\left( \mathrm{e}^{\beta\omega} + 1 \right)^3} \tag{172}$$

$$\| [a, a\rho_{\mathrm{B}}] \|_{\mathrm{HS}}^2 = \| a\, [a, \rho_{\mathrm{B}}] \|_{\mathrm{HS}}^2$$
$$= \frac{2 \left( \mathrm{e}^{\beta\omega} - 1 \right)}{\left( \mathrm{e}^{\beta\omega} + 1 \right)^3}. \tag{173}$$

## C.3   Examples

The discussion in the previous subsections considers the most general case for a Hamiltonian of the form of Eq. (23). Let us apply this to specific models, particularly the examples considered in the main text. For this, we can plug in the corresponding system operators into the bound presented in Theorem C.5. In the single-mode case, the bounds can be tightened using the norms collected in Section C.2.

**Pure dephasing model**.

The first example is the dephasing model presented in Section 2. The Hamiltonian is given in Eq. (2) and reads
$$H = \frac{\omega_{\mathrm{S}}}{2} \sigma_z + \omega_{\mathrm{B}} a^\dagger a + f \sigma_z (a + a^\dagger). \tag{174}$$

As the decoupling set, we choose $\mathscr{V} = \{I, \sigma_x\}$ and $\rho_{\mathrm{S}} = |+\rangle \langle +|$. Then, our bound becomes

$$\xi_N(t; \rho) \leq \frac{t^2}{4N} |f| \left[ \sqrt{2} \sqrt{\frac{\omega_{\mathrm{B}}^2}{\left( \mathrm{e}^{\beta\omega_{\mathrm{B}}} + 1 \right)^2}} + \sqrt{\frac{\omega_{\mathrm{B}}^2 \mathrm{e}^{\beta\omega_{\mathrm{B}}}}{\cosh(\beta\omega_{\mathrm{B}}) + 1}} + 2 \sqrt{\omega_{\mathrm{B}}^2 \tanh^2 \left( \frac{\beta\omega_{\mathrm{B}}}{2} \right)} \right.$$

$$\left. + 2\sqrt{2} \omega_{\mathrm{S}} \left( \sqrt{\frac{1}{\left( \mathrm{e}^{\beta\omega_{\mathrm{B}}} + 1 \right)^2}} + \sqrt{\frac{\mathrm{e}^{2\beta\omega_{\mathrm{B}}}}{\left( \mathrm{e}^{\beta\omega_{\mathrm{B}}} + 1 \right)^2}} + \sqrt{\tanh^2 \left( \frac{\beta\omega_{\mathrm{B}}}{2} \right)} \right) \right]$$

$$+ \frac{t^2}{4N} |f|^2 \left[ 6 \sqrt{\frac{\mathrm{e}^{\beta\omega_{\mathrm{B}}} - 1}{\left( \mathrm{e}^{\beta\omega_{\mathrm{B}}} + 1 \right)^3}} + 4\sqrt{2} \sqrt{\tanh^3 \left( \frac{\beta\omega_{\mathrm{B}}}{2} \right)} \right.$$

$$+ 2 \sqrt{\frac{\cosh(\beta\omega_{\mathrm{B}})(\coth(\beta\omega_{\mathrm{B}}) - 1)}{\cosh(\beta\omega_{\mathrm{B}}) + 1}} + 2 \sqrt{\frac{\cosh(\beta\omega_{\mathrm{B}})(\coth(\beta\omega_{\mathrm{B}}) + 1)}{\cosh(\beta\omega_{\mathrm{B}}) + 1}}$$

$$+ 2 \sqrt{\frac{\mathrm{e}^{-2\beta\omega_{\mathrm{B}}} \operatorname{csch}(\beta\omega_{\mathrm{B}})}{\cosh(\beta\omega_{\mathrm{B}}) + 1}} + 2 \sqrt{\frac{\mathrm{e}^{2\beta\omega_{\mathrm{B}}} \operatorname{csch}(\beta\omega_{\mathrm{B}})}{\cosh(\beta\omega_{\mathrm{B}}) + 1}}$$

$$+ \sqrt{2} \sqrt{\left( \sinh \left( \frac{3\beta\omega_{\mathrm{B}}}{2} \right) - \sinh \left( \frac{\beta\omega_{\mathrm{B}}}{2} \right) \right) \operatorname{sech}^3 \left( \frac{\beta\omega_{\mathrm{B}}}{2} \right)}$$

$$\left. + 3 \sqrt{\mathrm{e}^{\beta\omega_{\mathrm{B}}} \tanh \left( \frac{\beta\omega_{\mathrm{B}}}{2} \right) \operatorname{sech}^2 \left( \frac{\beta\omega_{\mathrm{B}}}{2} \right)} \right]$$

$$+ \frac{t^2}{4\sqrt{2}N} \omega_{\mathrm{S}}^2 \sqrt{\tanh \left( \frac{\beta\omega_{\mathrm{B}}}{2} \right)}. \tag{175}$$

Using Corollary C.6, we can obtain a looser, but simpler bound, which is given in Eq. (13) of the main text.

**Jaynes–Cummings model**

The second example is the Jaynes–Cummings model from Section 6.1:

$$H = \frac{\omega_S}{2}\sigma_z + \omega_B a^\dagger a + f(\sigma^+ a + \sigma^- a^\dagger). \tag{176}$$

As the decoupling set, we choose the full Pauli group $\mathscr{V} = \{I, \sigma_x, \sigma_y, \sigma_z\}$. For $\rho_S = |0\rangle\langle 0|$, our bound reads

$$
\begin{aligned}
\xi_N(t; |0\rangle\langle 0| \otimes \rho_B) \leq{} & \frac{t^2}{4N}|f|\Bigg[2\Bigg\{|\omega_S|\Bigg(\sqrt{\frac{1}{(e^{\beta\omega_B}+1)^2}} + \sqrt{\frac{e^{2\beta\omega_B}}{(e^{\beta\omega_B}+1)^2}} + \sqrt{\tanh^2\left(\frac{\beta\omega_B}{2}\right)}\Bigg) \\
& + \sqrt{\frac{\omega_B^2}{(e^{\beta\omega_B}+1)^2}} + \sqrt{\frac{\omega_B^2 e^{2\beta\omega_B}}{(e^{\beta\omega_B}+1)^2}} + \sqrt{\omega_B^2\tanh^2\left(\frac{\beta\omega_B}{2}\right)}\Bigg\} \\
& + |f|\Bigg(2\sqrt{\frac{1}{(e^{\beta\omega_B}-1)(e^{\beta\omega_B}+1)^3}} + 2\sqrt{2}\sqrt{\frac{e^{\beta\omega_B}-1}{(e^{\beta\omega_B}+1)^3}} \\
& + \sqrt{\frac{e^{\beta\omega_B}-1}{(e^{\beta\omega_B}+1)^3}} + \sqrt{\frac{e^{2\beta\omega_B}(e^{\beta\omega_B}-1)}{(e^{\beta\omega_B}+1)^3}} + 2\sqrt{2}\sqrt{\tanh^3\left(\frac{\beta\omega_B}{2}\right)} \\
& + \sqrt{\tanh^3\left(\frac{\beta\omega_B}{2}\right)} + \tanh\left(\frac{\beta\omega_B}{2}\right) + \sqrt{\frac{\cosh(\beta\omega_B)(\coth(\beta\omega_B)-1)}{\cosh(\beta\omega_B)+1}} \\
& + \sqrt{\frac{e^{2\beta\omega_B}\operatorname{csch}(\beta\omega_B)}{\cosh(\beta\omega_B)+1}} + 4\sqrt{e^{\beta\omega_B}\sinh^4\left(\frac{\beta\omega_B}{2}\right)\operatorname{csch}^3(\beta\omega_B)} \\
& + \sqrt{\frac{(e^{2\beta\omega_B}+1)\operatorname{csch}(\beta\omega_B)}{\sinh(2\beta\omega_B)\operatorname{csch}(\beta\omega_B)+2}}\Bigg)\Bigg],
\end{aligned}
\tag{177}
$$

and for $\rho_S = |+\rangle\langle +|$, we obtain

$$
\begin{aligned}
\xi_N(t; |+\rangle\langle +| \otimes \rho_B) \leq{} & \frac{\sqrt{2}t^2}{4N}\omega_S^2\sqrt{\tanh\left(\frac{\beta\omega_B}{2}\right)} + 2|f|\Bigg[4\sqrt{2}\omega_S\sqrt{\tanh^2\left(\frac{\beta\omega_B}{2}\right)} \\
& + 4\sqrt{2}\sqrt{\omega_B^2\tanh^2\left(\frac{\beta\omega_B}{2}\right)} + 8|f|\sqrt{\tanh^3\left(\frac{\beta\omega_B}{2}\right)} \\
& + \left(1+\sqrt{3}\right)|f|\sqrt{\left(\sinh\left(\frac{3\beta\omega_B}{2}\right)-\sinh\left(\frac{\beta\omega_B}{2}\right)\right)\operatorname{sech}^3\left(\frac{\beta\omega_B}{2}\right)}\Bigg] \\
& + \frac{t^2}{16N}|f|\Bigg[4\sqrt{2}\Bigg\{\omega_S\Bigg(\sqrt{\frac{1}{(e^{\beta\omega_B}+1)^2}} + \sqrt{\frac{e^{2\beta\omega_B}}{(e^{\beta\omega_B}+1)^2}}\Bigg) \\
& + \sqrt{\frac{\omega_B^2}{(e^{\beta\omega_B}+1)^2}} + \sqrt{\frac{\omega_B^2 e^{2\beta\omega_B}}{(e^{\beta\omega_B}+1)^2}}\Bigg\} \\
& + |f|\Bigg(8\sqrt{2}\sqrt{\frac{1}{(e^{\beta\omega_B}-1)(e^{\beta\omega_B}+1)^3}} + 2\sqrt{6}\sqrt{\frac{e^{\beta\omega_B}-1}{(e^{\beta\omega_B}+1)^3}}
\end{aligned}
$$

$$+ 10\sqrt{2}\sqrt{\frac{e^{\beta\omega_B} - 1}{(e^{\beta\omega_B} + 1)^3}} + 4\sqrt{2}\sqrt{\frac{\cosh(\beta\omega_B)(\coth(\beta\omega_B) - 1)}{\cosh(\beta\omega_B) + 1}}$$

$$+ 4\sqrt{2}\sqrt{\frac{e^{2\beta\omega_B}\operatorname{csch}(\beta\omega_B)}{\cosh(\beta\omega_B) + 1}} + 4\sqrt{\frac{(e^{2\beta\omega_B} + 1)\operatorname{csch}(\beta\omega_B)}{\cosh(\beta\omega_B) + 1}}$$

$$+ 20\sqrt{e^{\beta\omega_B}\sinh^4\left(\frac{\beta\omega_B}{2}\right)\operatorname{csch}^3(\beta\omega_B)}$$

$$\left. + \sqrt{6}\sqrt{e^{\beta\omega_B}\tanh\left(\frac{\beta\omega_B}{2}\right)\operatorname{sech}^2\left(\frac{\beta\omega_B}{2}\right)}\right)\right] \tag{178}$$

A simpler loose bound independent of the system input state $\rho_S$ is obtained via Corollary C.6 and is given in Eq. (63) in the main text.

**Quantum Rabi model**

Let us also look at the quantum Rabi model from Section 6.2

$$H = \frac{\omega_S}{2}\sigma_z + \omega_B a^\dagger a + f\sigma_x(a + a^\dagger). \tag{179}$$

Again, the decoupling set will be $\mathcal{V} = \{I, \sigma_x, \sigma_y, \sigma_z\}$ and we consider the initial system input state $\rho_S = |0\rangle\langle0|$. Then, our bound becomes

$$\xi_N(t; |0\rangle\langle0| \otimes \rho_B) \le \frac{t^2}{2N}\left[|f|\left(\sqrt{2}\sqrt{\frac{\omega_B^2}{(e^{\beta\omega_B} + 1)^2}} + \sqrt{\frac{\omega_B^2 e^{\beta\omega_B}}{\cosh(\beta\omega_B) + 1}}\right.\right.$$

$$+ 2\sqrt{\omega_B^2\tanh^2\left(\frac{\beta\omega_B}{2}\right)} + \omega_S\left(\sqrt{2}\sqrt{\frac{1}{(e^{\beta\omega_B} + 1)^2}}\right.$$

$$\left.\left.+ \sqrt{\frac{e^{\beta\omega_B}}{\cosh(\beta\omega_B) + 1}} + 2\sqrt{\tanh^2\left(\frac{\beta\omega_B}{2}\right)}\right)\right)$$

$$+ |f|\left(6\sqrt{\frac{e^{\beta\omega_B} - 1}{(e^{\beta\omega_B} + 1)^3}} + 4\sqrt{2}\sqrt{\tanh^3\left(\frac{\beta\omega_B}{2}\right)}\right.$$

$$+ 2\sqrt{\frac{\cosh(\beta\omega_B)(\coth(\beta\omega_B) - 1)}{\cosh(\beta\omega_B) + 1}}$$

$$+ 2\sqrt{\frac{e^{-2\beta\omega_B}\operatorname{csch}(\beta\omega_B)}{\cosh(\beta\omega_B) + 1}} + 2\sqrt{\frac{e^{2\beta\omega_B}\operatorname{csch}(\beta\omega_B)}{\cosh(\beta\omega_B) + 1}}$$

$$+ \sqrt{2}\sqrt{\frac{(e^{2\beta\omega_B} + 1)\operatorname{csch}(\beta\omega_B)}{\cosh(\beta\omega_B) + 1}}$$

$$+ \sqrt{2}\sqrt{\left(\sinh\left(\frac{3\beta\omega_B}{2}\right) - \sinh\left(\frac{\beta\omega_B}{2}\right)\right)\operatorname{sech}^3\left(\frac{\beta\omega_B}{2}\right)}$$

$$\left.\left.+ 3\sqrt{e^{\beta\omega_B}\tanh\left(\frac{\beta\omega_B}{2}\right)\operatorname{sech}^2\left(\frac{\beta\omega_B}{2}\right)}\right)\right]. \tag{180}$$

Furthermore, the loose bound independent from the system input state $\rho_{\mathrm{S}}$ due to Corollary C.6 is given in Eq. (67) of the main text, which is exactly a factor of 10 larger than the loose bound for the dephasing model (13).

**Qubit coupled to infinitely many modes**

Lastly, we show the derivations for the qubit coupled to infinitely many modes from Section 6.3. The Hamiltonian is given in Eq. (69) and reads

$$H = \frac{\omega_{\mathrm{S}}}{2}\sigma_z + \sum_{k=1}^{\infty} \omega_k a_k^\dagger a_k + \sum_{k=1}^{\infty} f_k \left(\sigma_+ a_k + \sigma_- a_k^\dagger\right). \tag{181}$$

To apply our bound, we have to compute the derivatives of the grand canonical partition function

$$Z(\beta) = \mathrm{e}^{-\sum_{i=1}^{\infty} \ln[1-\exp(-\beta\omega_i)]}. \tag{182}$$

It is straightforward to see that its first derivative computes to

$$-\frac{\mathrm{d}}{\mathrm{d}(2\beta)}Z(2\beta) = \sum_{i=1}^{\infty} \frac{\mathrm{e}^{-2\beta\omega_i}\omega_i}{(1-\mathrm{e}^{-2\beta\omega_i})\prod_{j=1}^{\infty}\left(1-\mathrm{e}^{-2\beta\omega_j}\right)}. \tag{183}$$

To find the second derivative, we again differentiate using the quotient rule. This gives

$$\begin{aligned}
\frac{\mathrm{d}^2}{\mathrm{d}(2\beta)^2}Z(2\beta) &= \sum_{i=1}^{\infty}\left(\left(1-\mathrm{e}^{-2\beta\omega_i}\right)^{-2}\prod_{j=1}^{\infty}\left(1-\mathrm{e}^{-2\beta\omega_j}\right)^{-2}\right) \\
&\quad\times\left(\mathrm{e}^{-2\beta\omega_i}\omega_i^2\left(1-\mathrm{e}^{-2\beta\omega_i}\right)\prod_{j=1}^{\infty}\left(1-\mathrm{e}^{-2\beta\omega_j}\right)\right. \\
&\quad\quad+\mathrm{e}^{-2\beta\omega_i}\omega_i\left[\omega_i\mathrm{e}^{-2\beta\omega_i}\prod_{j=1}^{\infty}\left(1-\mathrm{e}^{-2\beta\omega_j}\right)\right. \\
&\quad\quad\quad\left.\left.+\left(1-\mathrm{e}^{-2\beta\omega_i}\right)\sum_{j=1}^{\infty}\omega_j\mathrm{e}^{-2\beta\omega_j}\prod_{k\neq j}\left(1-\mathrm{e}^{-2\beta\omega_k}\right)\right]\right) \\
&= \sum_{i=1}^{\infty}\frac{\mathrm{e}^{-2\beta\omega_i}\omega_i^2}{(1-\mathrm{e}^{-2\beta\omega_i})\prod_{j=1}^{\infty}\left(1-\mathrm{e}^{-2\beta\omega_j}\right)}+\frac{\mathrm{e}^{-4\beta\omega_i}\omega_i^2}{(1-\mathrm{e}^{-2\beta\omega_i})^2\prod_{j=1}^{\infty}\left(1-\mathrm{e}^{-2\beta\omega_j}\right)} \\
&\quad+\frac{\mathrm{e}^{-2\beta\omega_i}\omega_i\sum_{j=1}^{\infty}\omega_j\mathrm{e}^{-2\beta\omega_j}\prod_{k\neq j}\left(1-\mathrm{e}^{-2\beta\omega_k}\right)}{(1-\mathrm{e}^{-2\beta\omega_i})\prod_{j=1}^{\infty}\left(1-\mathrm{e}^{-2\beta\omega_j}\right)^2} \\
&= \sum_{i=1}^{\infty}\frac{\omega_i^2}{\prod_{j=1}^{\infty}\left(1-\mathrm{e}^{-2\beta\omega_j}\right)}\left[\frac{1}{2}\left(\coth\left(\beta\omega_i\right)-1\right)+\frac{1}{(\mathrm{e}^{2\beta\omega_i}-1)^2}\right] \\
&\quad+\frac{\omega_i\left(\coth\left(\beta\omega_i\right)-1\right)\sum_{j=1}^{\infty}\frac{\omega_j}{\mathrm{e}^{2\beta\omega_j}-1}}{2\prod_{j=1}^{\infty}\left(1-\mathrm{e}^{-2\beta\omega_j}\right)} \\
&= \sum_{i=1}^{\infty}\frac{\omega_i^2\mathrm{csch}^2\left(\beta\omega_i\right)+\omega_i\left(\coth\left(\beta\omega_i\right)-1\right)\sum_{j=1}^{\infty}n_j\omega_j\left(\coth\left(\beta\omega_j\right)-1\right)}{4\prod_{j=1}^{\infty}\left(1-\mathrm{e}^{-2\beta\omega_j}\right)}. 
\end{aligned} \tag{184}$$

For the explicit choice of parameters in the toy model considered in the main text $(\omega_k)_{k\in\mathbb{N}} = (k)_{k\in\mathbb{N}}$, $(f_k)_{k\in\mathbb{N}} = \left(\frac{f}{k^2}\right)_{k\in\mathbb{N}}$ with $f \in \mathbb{R}$, we rewrite the partition function as

$$
\begin{aligned}
Z(\beta) &= \frac{1}{(q_1; q_1)_\infty} \\
&= \exp\left[-\frac{1}{24}\ln\left(\frac{p_1(1-p_1)^4}{16q_1}\right) - \frac{1}{2}\ln\left(\frac{2\mathcal{K}(p_1)}{\pi}\right)\right] \\
&= \frac{\sqrt{\pi}}{2^{1/3}\sqrt{\mathcal{K}(p_1)}\left((p_1-1)^4 p_1 q_{-1}\right)^{1/24}}.
\end{aligned}
\tag{185}
$$

Here, $q_x = \mathrm{e}^{-x\beta}$ and, for a fixed $\beta > 0$ and $x \in \mathbb{R}$, $p_x$ is the unique parameter that corresponds to the nome $q_x$ in an elliptic function. This can be computed numerically, for instance, with the command `EllipticNomeQ` in `Mathematica`. Furthermore, the function $\mathcal{K}$ computes the complete elliptic integral of the first kind [74, Chapter 22.7] (which can easily be computed numerically, e.g. with the command `EllipticK` in `Mathematica`). This expression can be differentiated with respect to $\beta$. In particular,

$$
-\frac{\mathrm{d}}{\mathrm{d}(2\beta)}Z(2\beta) = \frac{\pi^2 + 4\mathcal{K}(p_2)\big(5\mathcal{K}(p_2) - 6\mathcal{E}(p_2)\big) - 4\mathcal{K}(p_2)^2 p_2}{24 \cdot 2^{1/3}\pi^{3/2}\sqrt{\mathcal{K}(p_2)}\big((p_2-1)^4 p_2 q_{-2}\big)^{1/24}}
\tag{186}
$$

and

$$
\begin{aligned}
\frac{\mathrm{d}^2}{\mathrm{d}(2\beta)^2}Z(2\beta) = {} & \frac{1}{576 \cdot 2^{1/3}\pi^{7/2}\sqrt{\mathcal{K}(p_2)}\big((p_2-1)^4 p_2 q_{-2}\big)^{1/24}} \\
& \times \Big[\pi^4 - 8\mathcal{K}(p_2)^2 p_2\big(\pi^2 + 24\mathcal{E}(p_2)\mathcal{K}(p_2) - 76\mathcal{K}(p_2)^2 - 2\mathcal{K}(p_2)^2 p_2\big) \\
& \quad + 8\mathcal{K}(p_2)\big(5\pi^2\mathcal{K}(p_2) - 72\mathcal{E}(p_2)^2\mathcal{K}(p_2) \\
& \quad - 46\mathcal{K}(p_2)^3 - 6\mathcal{E}(p_2)(\pi^2 - 20\mathcal{K}(p_2)^2)\big)\Big],
\end{aligned}
\tag{187}
$$

where the function $\mathcal{E}$ computes the complete elliptic integral of the second kind [74, Chapter 22.73] (which can easily be computed numerically, e.g. with the command `EllipticE` in `Mathematica`).

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
