# Peer review of "Efficiency of Dynamical Decoupling for (Almost) Any Spin–Boson Model"

_SciPost Physics, doi:SciPost Phys. 19, 035 (2025)_

## Round 1 · Referee Report · Anonymous (Referee 1) · 2025-5-9

Report
Efficiency of Dynamical Decoupling for (Almost) Any Spin–Boson Model by Alexander Hahn, Daniel Burgarth, Davide Lonigro
The manuscript investigates dynamical decoupling schemes to suppress dissipative environmental influence in the dynamics of quantum systems. Specifically, a two-level system coupled to a bath is studied. The authors derive analytically a bound on the convergence speed with the number of decoupling pulses within a given time and compare their bound with numerical results. As important technical steps several known results for pure states have here been extended to mixed states.
The results are interesting and the manuscript well written. Before a final recommendation to publish I would ask the authors to address the following points:
-
Discussing Hamiltonian (1) one might want to refer also to -- U. Weiss, Quantum Dissipative Systems, 4th ed. (World Scientific, Singapore, 2012). -- A. J. Leggett, S. Chakravarty, A. T. Dorsey, M. P. A. Fisher, A. Garg, and W. Zwerger, Rev. Mod. Phys. 59, 1 (1987).
-
Eq. (19) means that the spectral density can not go analytically down to zero (even with value zero at w=0). That is unusual given that typically Ohmic spectra are investigated. This unusual choice should be mentioned / discussed a bit clearer.
-
page 10: Remark: '...the general case can always be recovered via a shift of the modes ωk .' Excitation energies can only be shifted as long one never touches zero.
-
For proof of (35) is a recent reference given. The fact (35) is, however, already the base of the (discrete) path integral formalism. Thus, I would expect earlier investigations.
-
Assumption 3.1(1) is different from the one stated in Table 1. Both seem very different. Which one is correct ?
Recommendation
Ask for minor revision

---

## Round 1 · Referee Report · Anonymous (Referee 2) · 2025-5-13

Strengths
1) Mathematical rigor 2) Quite general applicability 3) Compresensive presentation
Weaknesses
1) Physics behind the estimates only partially elucidated
Report
dynamic decoupling for models with bosonic baths. The
models are fairly general for which the bounds apply.
This is remarkable because the bosonic Hamiltonians are
unbounded by construction. Thus, I am in favor of publication
as SciPost Core or SciPost Physics.
Still, there are a number of points which can be improved.
In the Introduction, I find it appropriate to mention
also the previous estimates obtained for finite Hamiltonians,
in particular for non-equidistant pulses, see e.g.
author={G. S. Uhrig and D. A. Lidar}, title = {Rigorous
Bounds for Optimized Dynamical Decoupling},
journal= Phys. Rev. A, volume=82, pages=012301, year = 2010
or
author = {Y. Xia and G. S. Uhrig and D. A. Lidar}, title = {
Rigorous performance bounds for quadratic and nested dynamical decoupling},
journal = Phys. Rev. A, volume = 84, pages = 062332, year = 2011.
Similarly, the authors need not criticize the filter function
approach as strongly as they do. It is a very useful tool
to design experiments, see e.g.
author= {M. J. Biercuk and H. Uys}, title =
{Dynamical decoupling sequence construction as a filter-design problem},
journal=J. Phys. B, volume=44, pages= 154002, year=2011
even if does not provide mathematically rigorous statements.
The authors should discuss the physics behind their formulae wherever
this is possible. In particular, the existence of non-existence
of a UV cutoff is crucial, see e.g. the condition Eq. (56).
A Lorentzian resonance would not comply with this condition, but
still the dynamic decoupling works. This can indicate directions
for improvement of the bounds.
For the qualitative understanding why the Liouvillean is bounded
even for an unbounded Hamiltonian a very simple example
would be very helpful. At least, this crucial point should be exposed
clearly for the general reader.
Generally, the bounds are still quite loose (generically two
orders of magnitude) which is not uncommon in mathematical physics.
The authors should at least discuss why this is so?
Which effects are so strongly simplified in the estimates?
Would another norm be advantageous?
In all estimates presented, the error scales like 1/N. This property
is inherited from the Trotter formular. The authors also highlight
that the properties of the Trotter formulae are important by themselves.
Wouldn't it be an intriguing to improve the scaling to 1/N^2 or even
higher? This should be possible with non-equistant pieces of Hamiltonians
as in iterated universal dynamic decoupling, see
author = {G. S. Uhrig}, title = {Exact Results on Dynamical
Decoupling by $\pi$-Pulses in Quantum Information Processes},
journal= New J. Phys., volume=10, pages=083024, year = 2008
Of course, I am not demanding to include such a derivation
in the present manuscript. But it could be briefly mentioned and
discussed in the outlook.
It seems that on page 28 there is an error in the second mentioned
inequality, please have a look.
Requested changes
Each paragraph of the report given above suggests a certain change.
Recommendation
Ask for minor revision

---

## Round 2 · Author Response

We thank the referees for their positive feedback and their insightful suggestions to improve the manuscript and its presentation. In the following, we address each point mentioned by the referees following the same paragraph structure.

---

## Round 2 · List of Changes

Warnings issued while processing user-supplied markup:

  • Inconsistency: plain/Markdown and reStructuredText syntaxes are mixed. Markdown will be used.
    Add "#coerce:reST" or "#coerce:plain" as the first line of your text to force reStructuredText or no markup.
    You may also contact the helpdesk if the formatting is incorrect and you are unable to edit your text.

Referee 1

As suggested by the referee, we included the two references on error estimates for dynamical decoupling of finite-dimensional Hamiltonians with non-equidistant pulses in the introduction.

We agree with the referee that the filter function approach is a useful approach in practice to design suitable dynamical decoupling sequences. We purposely emphasized the weaknesses of this approach in the paper to contrast it with the Trotterization approach that we employ in the paper. Both methods have their own advantages and disadvantages. While Trotterization provides rigorous error estimates, it does not help with comparing different decoupling sequences or engineering optimal decoupling sequences for a given noise profile. This is where the filter function approach is very valuable. Nevertheless, from a purely mathematical perspective, the filter function approach is not well-defined in many cases. As it can still give valuable insights in certain regimes, we added two sentences to the introduction highlighting the usefulness of this approach: - “On the other hand, filter functions are a useful tool to compare different or design optimal decoupling strategies in the perturbative regime [19].” - “Furthermore, they might help to identify the perturbative regime, in which the filter function approach is favourable.”

Indeed, a physical discussion of Assumption 5.1 would be valuable. We added such a discussion below the statement of the assumption, in which we also refer to the concluding remarks where we point out an avenue of how to potentially relax this assumption.

In general, the Liouvillian will be an unbounded operator on Liouville space if the corresponding Hamiltonian is unbounded. In fact, the situation is even worse: It can happen that the Liouvillian is doubly unbounded even if the Hamiltonian is only semi-unbounded. The reason why the quantities in our error bounds remain finite is because the Gibbs state acts as a regularizer. Under our assumptions, at least the first four moments of the Liouvillian and its rotated versions in the Gibbs state are well-defined and finite. To clarify this point, we added a comment at the end of Section 4.1.

It is true that our bounds are quite loose in their absolute value when compared to the numerical simulations performed in this paper. Indeed, this shows that there is potential to improve our results. This is not surprising given the proof method we use, and we also expect a similar behaviour for other norms. We now explain where in the proof the overestimation of the error comes from, see the newly added paragraph at the beginning of Section 6. Furthermore, we added Remark 4.4 at the end of Section 4 with an abstract result on the somehow more natural trace norm error that also motivates using the Hilbert—Schmidt norm for the explicit decoupling bounds. In addition, we found a small inconsistency in the normalization we used for the Trotter error bound and the decoupling error bound. This led to a slightly larger pre-factor in Theorem 5.3 and our model-specific bounds than necessary. More concretely, the bound was missing a pre-factor of $1/L^2$ coming from Trotterizing between operators $(1/L ad_{H_j})$ instead of just $ad_{H_j}$. We corrected this inconsistency in all affected analytical expressions. Furthermore, we corrected this in the plots that compare the bounds with a numerical simulation. Here, the simulation results were correct but the shown bounds were unnecessarily large. To make the changes clearer, we added further explanation to the proof outline of Theorem 5.3. Through these changes, the bounds became tighter than before.

We thank the referee for pointing out another interesting avenue for extending our results. We added this as a separate point (point (v)) in the Concluding remarks, where we discuss potential directions for generalizing and improving our results. Here, we also comment on a possible way how to obtain the result mentioned by the referee.

Indeed, there was a typo in the inequality. This is corrected now.

Referee 2

  1. We included the mentioned references in the introduction after Equation (1). We thank the referee for pointing out these references.

  2. It is true that the assumption in Equation (19) seems to be unnatural when considering Ohmic spectral densities. However, this is not a problem. In fact, as discussed in the paragraph between Equation (22) and (23), the assumption in Equation (19) is equivalent to $m>-\infty$ after shifting the zero-point energy. W.l.o.g., this can always be done (also see the reply to the third comment by the referee) so that Ohmic spectra can be tackled through our method. For clarity, we added a comment on Ohmic spectra in this paragraph.

  3. The bath Hamiltonian is bounded from below, and by assumption, we are working in a model with discrete bath modes. This means that there can only be finitely many negative $\omega_k$. Therefore, shifting the excitation energies is always possible w.l.o.g. We are grateful to the referee for pointing out that this has not been clearly presented and added a comment between Equation (22) and (23).

  4. We thank the referee for pointing out the existence of earlier work on the statement of Theorem 4.1. Indeed, this statement was first proved by Kato in 1978 in the context of the path integral. Here, one is interested in the Trotter splitting between kinetic and potential energy, i.e., in Trotterizing between two Hamiltonians. In dynamical decoupling, one naturally requires Trotterization between more than two components. This is why, to the best of our knowledge, a generalization of Kato’s result to more than two terms has only recently been obtained in the context of dynamical decoupling (in 2018 in the reference we cited). Nevertheless, it is definitely worth citing Kato’s original work here, and we included this reference.

  5. We thank the referee for pointing out that the presentation of Assumption 3(i) might have been confusing. In fact, both assumptions are equivalent after shifting the zero-point energy, which is always possible (again, see our reply to comment 3 by the referee). We made this clear now in the caption of Table 1, also indicating how our error bounds would change through the energy shift.

---

## Editorial Decision

published